# Remote tuning of single-atom Fe-N$_5$ sites via high-coordination defects for enhanced Fenton-like water decontamination

Sijia Jin[1,2], Wenxian Tan[2], Yilin Huang[3], Yi Wang[2], Zhiqiao He[2], Haiyan Zhang[1], Shuang Song [2], Yaqi Cai [1,4] & Tao Zeng [1,2,5] ✉

Fe-N$_5$ single-atom catalysts (SACs) hold great promise for water decontamination, however, the fundamental relationship between their high coordination shell environment and catalytic performance in Fenton-like reactions remains poorly understood. Here, we precisely regulate the high coordination shell defects of a model SAC with well-defined axial Fe-N$_5$ configurations to elucidate the impact of remote interactions on peroxymonosulfate (PMS) activation. Experimental and theoretical studies confirm that remote modulation of Fe-N$_5$ sites through high coordination shell defects profoundly enhance Fenton-like catalytic activity, enabling FeN$_5$-SD$_2$ to achieve a turnover frequency (TOF) value of 0.338 min$^{-1}$, surpassing state-of-the-art SACs. Our findings reveal a critical volcano-type correlation between defect content and catalytic efficiency, where coordinated modulation of Fe $d$-band center positioning and PMS adsorption energetics governs reaction dynamics. Only the FeN$_5$-SD$_2$ configuration with an optimal level of defects density and moderate adsorption energy enables sufficient O-O bond elongation in PMS to lower the energy barrier for selective singlet oxygen ($^1O_2$) evolution. This study unveils the mechanistic role of higher coordination shell defects in regulating Fe-N$_5$ active sites and introduces a well-defined model to investigate the structure–property correlations of higher coordination shells in SACs for Fenton-like reactions.

Fenton-like chemistry leveraging solid oxyanions, particularly peroxymonosulfate (PMS) activation, represent a promising avenue as alternatives to liquid-phase hydrogen peroxide (H$_2$O$_2$)-based reactions due to their enhanced efficacy and versatility in degrading recalcitrant organic contaminants[1]. While homogeneous first-row transition metal ions such as Co$^{2+}$, Fe$^{2+}$, Cu$^{2+}$, and Mn$^{2+}$ have shown promising activation capabilities for oxyanions, they suffer from drawbacks including poor

recyclability and sludge buildup. In contrast, heterogeneous activators offer potential solutions to these challenges. However, the intrinsic heterogeneity of nanoparticles (NPs) lowers atom utilization efficiency and makes it difficult to identify true active sites and reaction pathways in Fenton-like reactions, compared to homogeneous catalysts[2,3].

Single-atom catalysts (SACs) represent a pivotal advancement, offering enhanced atomic utilization efficiency and adjustable

[1]Zhejiang Key Laboratory of Environment and Health of New Pollutants, School of Environment, Hangzhou Institute for Advanced Study, University of Chinese Academy of Sciences, Hangzhou, P.R. China. [2]Zhejiang Key Laboratory of Low-carbon Control Technology for Industrial Pollution, College of Environment, Zhejiang University of Technology, Hangzhou, Zhejiang, P.R. China. [3]Office for Environmental Programs, Faculty of Science, The University of Melbourne, Parkville, VIC, Australia. [4]State Key Laboratory of Environmental Chemistry and Ecotoxicology, Research Center for Eco-Environmental Sciences, Chinese Academy of Sciences, Beijing, P.R. China. [5]Shaoxing Research Institute, Zhejiang University of Technology, Shaoxing, P. R. China. ✉e-mail: zengtao@ucas.ac.cn

electronic structure, thereby bridging the divide between homogeneous and heterogeneous catalysis[4–6]. Within this category, atomically dispersed metals supported on nitrogen-doped carbon frameworks (M-N-C) stand out due to their remarkable catalytic efficacy in activating PMS, in which the planar four-coordinated configuration of metal-N$_4$ (M-N$_4$) moieties has been widely recognized as the active sites[7,8]. Despite this, a lack of comprehensive understanding regarding the true active sites and the intricate structure-property relationships has resulted in an opaque and largely empirical understanding of the catalytic activation mechanisms. Using Fe-N$_4$ catalysts as an exemplar, various pathways, including radical (sulfate radical (SO$_4^{\cdot-}$) and hydroxyl radical ($\cdot$OH)), non-radical (singlet oxygenation), and direct electron-transfer processes (ETP), have all been proposed as potential routes for PMS activation[9]. This seeming contradiction highlights the need for a more quantitative grasp and precise assessment of reactive sites and reaction mechanisms, which are vital for advancing the rational design of SACs and optimizing PMS activation efficiency.

In general, the catalytic efficacy of SACs in PMS activation hinges profoundly upon the coordination intricacies and electronic configurations of the catalytically active metal centers. This principle reflects the behavior of natural metalloenzymes[10], where catalytic activity is finely tuned by a well-aligned framework. The central metal site, metal–ligand clusters in the primary coordination sphere, and the binding pocket in the secondary sphere work together through noncovalent interactions to optimize intermediate adsorption[11,12]. Modifying the coordination number or introducing heteroatoms (e.g., O, S, or P) into the M−N$_4$ framework has been widely used to regulate the first coordination shell, which is directly bonded to the metal center. These strategies have led to the development of asymmetric coordination structures[13,14]. For instance, the newly formed active sites (e.g., Co-N$_3$O$_1$, Fe-N$_3$S$_1$, or Fe-N$_3$P$_1$) with tailored charge/spin states and optimized electron density near the Fermi level. These features adjust the binding energies of reactive intermediates, controlling both the activation efficiency and selectivity in PMS activation[15]. Additionally, the introduction of out-of-plane coordination at the central metal sites in the axial direction, known as M-N$_5$ species, further alters the electronic structure of the conventional M-N$_4$ configurations. This modification significantly influences the perpendicular adsorption behavior of PMS molecules and thereby tuning the overall catalytic performance[16]. Beyond the first-shell coordination, the environment of the second and higher coordination shells, which are not directly bonded to the metal sites, also plays a critical role in shaping the electronic structure of the active site and influencing catalytic efficiency. Engineered coordination defects in higher coordination shells significantly affect the Fe active sites through a remote modulation effect. This "remote modulation" refers to catalytic changes induced by structural defects or heteroatoms located more than ~4 Å from the Fe center[17–19], beyond the primary (directly bonded) and secondary (bonded to first-shell atoms) coordination spheres. This realm remains relatively unexplored and has seldom been addressed in studies concerning PMS activation, possibly attributed to uncertainties in the local structure of active sites stemming from uncontrolled pyrolysis synthesis. To overcome this limitation, precise engineering of the high coordination shell environment is essential for uncovering the structure–property relationships of SACs beyond conventional M-N$_4$ centers. By targeting Fe-N$_5$ configurations and drawing inspiration from the spatial precision of natural metalloenzymes, this work aims to advance the design of SACs for more efficient PMS activation.

In this work, to unravel the intricate fundamental relationship between the high coordination shell environment and the catalytic performance of SACs in Fenton-like reactions, we designed a model catalytic architecture by integrating fully conjugated polyphthalocyanine frameworks wrapped on nitrogen-doped carbon nanotubes (NCNTs). This strategy integrates axial Fe-N$_5$ configurations

in the primary coordination shell, complemented by planar structural defect in higher coordination shells (FeN$_5$-SD$_x$). The remote modulation of Fe-N$_5$ sites via high coordination shell defects significantly optimizes the electronic properties of Fe centers, thereby enhancing their efficacy in selectively activating PMS. Consequently, the FeN$_5$-SD$_2$ catalyst demonstrated exceptional performance in PMS activation, achieving a turnover frequency (TOF) value of 0.338 min$^{-1}$ for bisphenol A (BPA) degradation. In-depth theoretical calculations have demonstrated that the higher coordination shells defects modifying the $d$-band center of the isolated Fe sites, coupled with a defect content-dependent volcano-like trend in PMS adsorption and activation. Additionally, it promotes charge migration at the reaction interface and notably elongates the O − O bond. Consequently, this modification reduces the energy barrier for generating crucial reaction intermediates, thereby facilitating the selective production of singlet oxygen ($^1$O$_2$). This study broadens the design of SACs beyond the conventional M-N$_4$ motif by tuning multiple coordination shells simultaneously. It reveals fundamental structure–property relationships for PMS activation and offering insights into the rational design of next-generation catalysts for water purification.

## Results

### Synthesis and structure characterization

This synthesis strategy fundamentally differs from conventional M-N-C catalyst preparation by eliminating high-temperature pyrolysis while maintaining well-defined active site structure (Fig. 1a). The polyphthalocyanine framework inherently facilitates transition metal incorporation within its macrocyclic cavity. Simultaneously, nitrogen-rich NCNTs promote axial Fe−N coordination. Together, these features stabilize highly ordered Fe-N$_5$ centers as the dominant catalytic sites. To engineer the higher coordination shells around Fe centers, which typically include atoms beyond the primary and secondary shells, we introduced coordination defects (Fig. 1b). This was achieved by copolymerizing a controlled amount of 1,2-dicyanobenzene (DCB) with benzene-1,2,4,5-tetracarbonitrile (BTC). Incorporating DCB systematically disrupts the polymer matrix, creating tailored defects in the higher coordination shells. These defects lie beyond 5 Å from the Fe center, as confirmed by DFT calculations (Supplementary Figs. 1-2), and are denoted as FeN$_5$-SD$_x$. Scanning electron microscopy (SEM) and high-resolution transmission electron microscopy (HR-TEM) analyses confirm that FeN$_5$-SD$_2$ retains an interconnected three-dimensional network of nanofibers with no observable particulate aggregates on surfaces (Fig. 1c). The roughened texture observed along the curved NCNTs originates from a conformal polymer overlayer[16]. X-ray diffraction (XRD) patterns of FeN$_5$-SD$_x$ (Supplementary Fig. 3) display two characteristic peaks at ~26° and ~43°, indexed to the (002) and (100) crystallographic planes of graphitic carbon in NCNTs[20]. The absence of diffraction signals for metallic iron or iron oxides confirms that Fe species are atomically dispersed. Furthermore, Aberration-corrected high-angle annular dark-field scanning transmission electron microscopy (AC-HAADF-STEM) images directly visualizes isolated Fe single-atoms as discrete bright contrast features (red circles, Fig. 1d), with no evidence of clustering. Complementarily, energy-dispersive X-ray spectroscopy (EDS) elemental mapping reveals homogeneous spatial distributions of C, N, and Fe across the NCNT architecture (Supplementary Fig. 4), further corroborating the uniform anchoring of single-atom Fe sites along the nanotube surfaces without agglomeration.

Fourier transform infrared (FTIR) spectra of all FeN$_5$-SD$_x$ samples exhibit the characteristic absorption bands of NCNTs, along with additional peaks assigned to polyphthalocyanine vibrations (Fig. 1e). The progressive disappearance of the -C≡N stretching vibration at 2240 cm$^{-1}$, characteristic of the BTC monomer, is accompanied by the emergence of new peaks corresponding to C = N (1500 and 1135 cm$^{-1}$) and C-N (1310 cm$^{-1}$) stretching modes. These changes indicate successful polymerization into a polyphthalocyanine-like framework. A

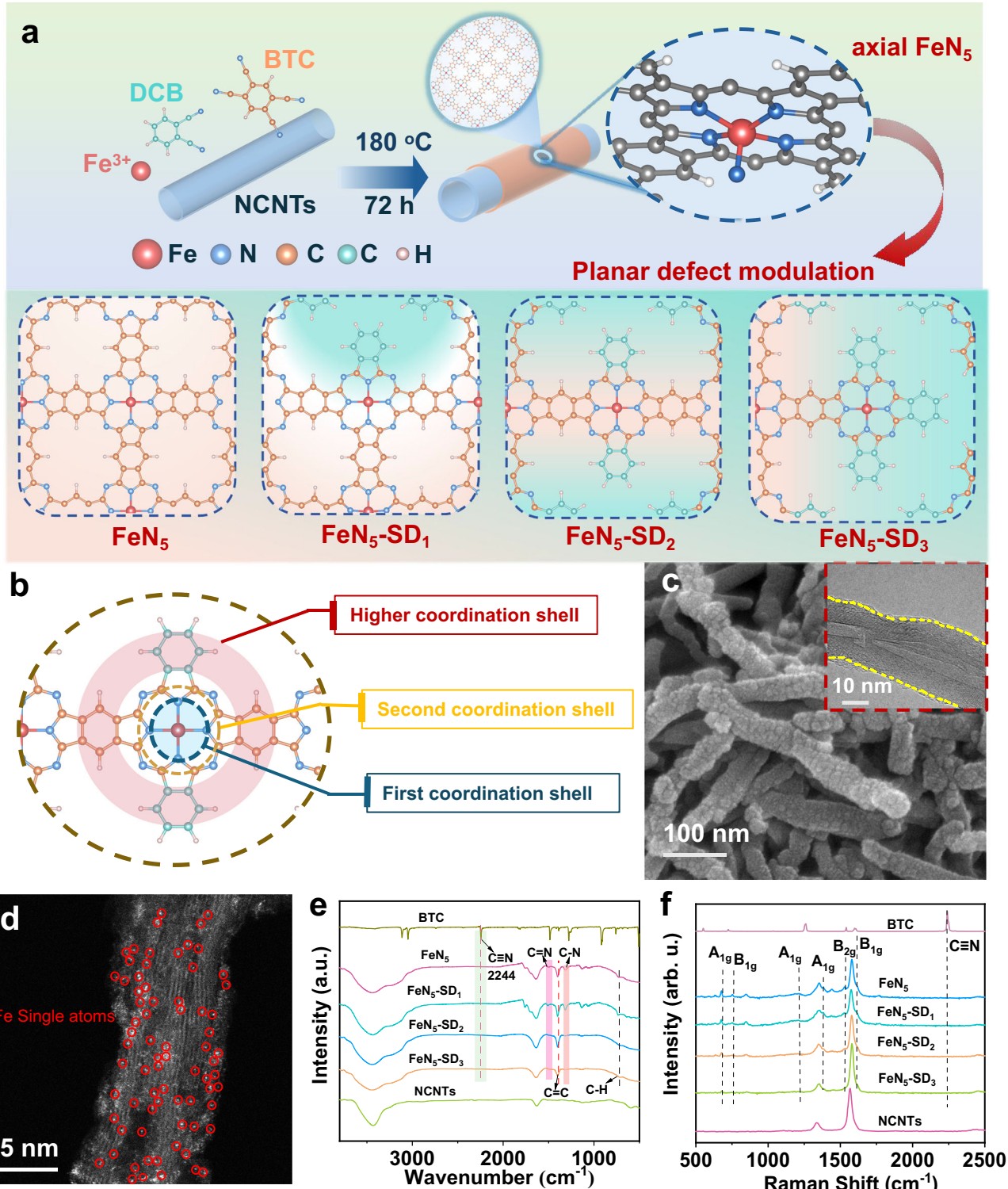

**Fig. 1 | Synthetic scheme and structural properties of as-prepared FeN₅ SACs.**
**a** Schematic of the preparation strategy for FeN₅-SD$_x$ using 1,2-dicyanobenzene (DCB), benzene-1,2,4,5-tetracarbonitrile (BTC) and nitrogen-doped carbon nanotubes (NCNTs) as precursors. Crystal structures visualized using VESTA software[49]. **b** Schematic illustration of the coordination structure of FeN₅-SD₂. Crystal structures visualized using VESTA software[49]. **c** SEM and HRTEM image of FeN₅-SD₂. **d** AC-HAADF-STEM of FeN₅-SD₂. **e** FTIR and **f** Raman spectra of BTC, FeN₅, FeN₅-SD₁, FeN₅-SD₂, FeN₅-SD₃ and NCNTs. Source data are provided as a Source Data file.

strong absorption at 1396 cm$^{-1}$ is attributed to C = C stretching within the conjugated macrocycle, while a peak at 731 cm$^{-1}$ arises from C-H vibrations, further confirming the formation of extended poly-phthalocyanine structure[21]. With increasing defect concentration, the intensity of the C = N and C-N bands moderately decrease, suggesting partial disruption of the pyrrole units within the phthalocyanine ring.

In contrast, the C = C band remains stable, implying that the benzene moieties are preserved. These observations are supported by Raman spectra (Fig. 1f), where in addition to the D and G bands of NCNTs, distinct A$_{1g}$, B$_{1g}$, and B$_{2g}$ modes related to C−C and C−N−C vibrations in the phthalocyanine ring are observed[22]. The gradual attenuation of these peaks implies localized structural perturbations rather than a

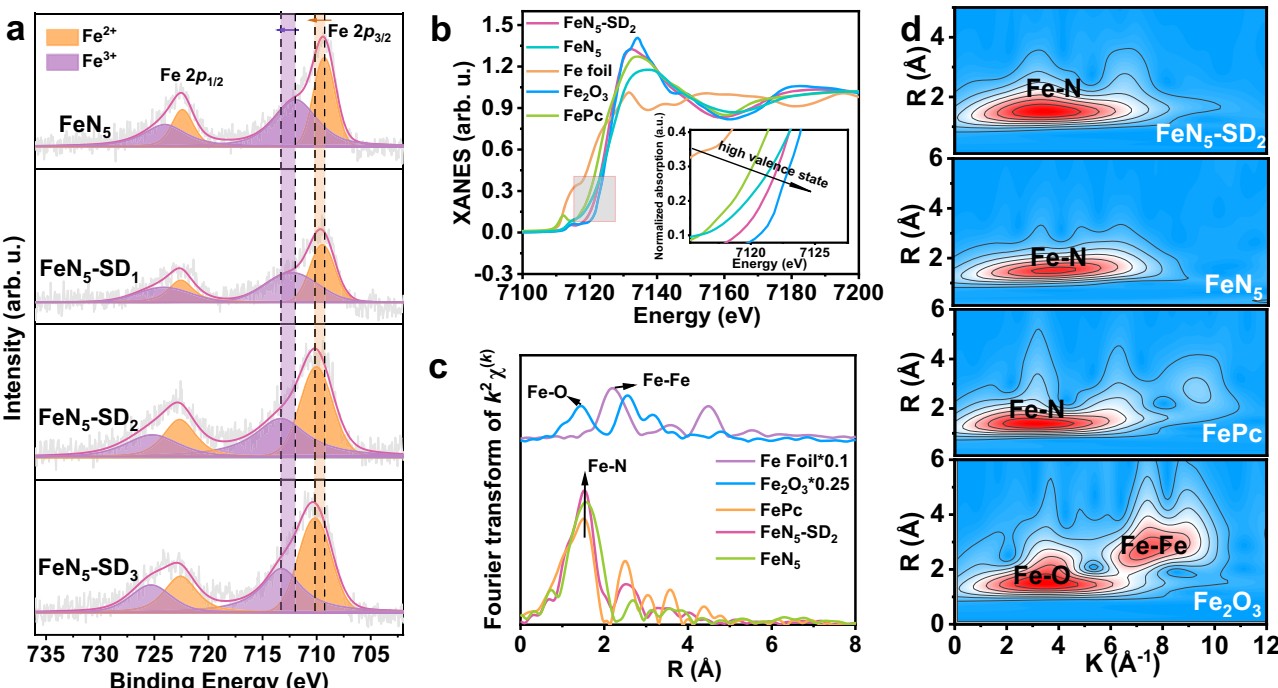

**Fig. 2 | Coordination geometry and chemical states of as-prepared FeN₅ SACs. a** XPS Fe 2*p* spectra of the as-prepared catalysts. **b** Fe *K*-edge XANES spectra, **c** FT- EXAFS spectra and (**d**) WT analysis of Fe foil, Fe₂O₃, FePc, FeN₅ and FeN₅-SD₂. Source data are provided as a Source Data file.

complete collapse of the polyphthalocyanine network. To further examine the coordination environment, solid-state $^{13}$C nuclear magnetic resonance (NMR) spectroscopy was conducted using NCNT-free FeN₄-SD$_x$ samples. As shown in Supplementary Fig. 5, a broad resonance at 100–130 ppm is assigned to aromatic carbons in benzene rings, while a distinct signal at -171 ppm corresponds to C = N groups in the pyrrole ring[21–23]. The progressive decrease in the 171 ppm signal from FeN₄ to FeN₄-SD₃ reflects a gradual loss of pyrrolic C = N groups, consistent with the formation of high coordination shell defects. Meanwhile, the signals from benzene carbons remain largely unchanged, confirming that structural disruption is confined to the pyrrole moieties. X-ray photoelectron spectroscopy (XPS) C 1*s* spectra (Supplementary Fig. 6) show a decrease in C = N and C−N components with increasing defect concentration, supporting the partial loss of coordinated nitrogen. Brunauer–Emmett–Teller (BET) analysis (Supplementary Table 1) further confirms a progressive increase in surface area and pore volume with defect introduction, while preserving the overall covalent framework.

## Local geometry and coordination environment

The chemical states of Fe in FeN₅-SD$_x$ catalysts were systematically investigated through high-resolution XPS measurements. Analysis of the Fe 2*p* spectra revealed distinct peaks corresponding to Fe$^{2+}$ and Fe$^{3+}$ at 709.2 eV and 711.9 eV, respectively, in the 2*p*$_{3/2}$ orbital, and at 722.4 eV and 724.0 eV, respectively, in the 2*p*$_{1/2}$ band (Fig. 2a). Crucially, the absence of a characteristic zero-valent iron (Fe$^0$) signal at 706.7 eV confirms the atomic dispersion of Fe species, eliminating the possibility of metallic clusters or nanoparticles. A notable positive binding energy shift of 0.74–1.30 eV was observed for both Fe$^{2+}$ and Fe$^{3+}$ species in FeN₅-SD$_x$ relative to FeN₅, indicative of strengthened electronic interaction between Fe nuclei and outer-shell electrons. This phenomenon, consistent with defect-induced modulation in higher coordination shells[16], suggests altered orbital configurations and energy levels at Fe active sites.

Complementary insights into the coordination geometry and electronic structure were obtained through Fe *K*-edge X-ray

absorption fine-structure (XAFS) studies. The X-ray absorption near-edge structure (XANES) spectra revealed absorption edges for FeN₅ and FeN₅-SD₂ positioned between Fe foil and Fe₂O₃ references, confirming an intermediate valence state between 0 and +3 (Fig. 2b). The distinct rightward shift of the FeN₅-SD₂ edge relative to FeN₅ demonstrates successful valence state elevation through defect engineering. Notably, both samples lack the pre-edge shoulder characteristic of square-planar FeN₄ configurations in FePc, confirming reconstruction of the Fe coordination environment[24]. Fourier transform (FT) analysis of EXAFS spectra identified a dominant peak at 1.53 Å corresponding to Fe-N coordination (Fig. 2c), with no detectable Fe-Fe scattering signal at 2.2 Å, corroborating XRD and AC-HAADF-STEM observations. Quantitative fitting analysis (Supplementary Fig. 7 and Supplementary Table 2) revealed an average Fe-N coordination number of 5 in FeN₅-SD₂ with an elongated bond length of 2.01 Å compared to FePc (1.93 Å). This bond lengthening is attributed to additional axial N coordination, confirming successful structural modification of Fe-N₅ basal planes. Wavelet transform (WT) analysis further validated the monatomic configuration, exhibiting a singular intensity maximum at 3.6 Å$^{-1}$ (Fig. 2d) corresponding exclusively to Fe−N interactions. This observation contrasts with reference materials such as Fe₂O₃, and FePc, where no intensity peaks corresponding to Fe−Fe and Fe−O interactions were detected. Collectively, these characterizations provide conclusive evidence for the precise engineering of defect-modulated Fe-N₅ architectures with tailored electronic and coordination structures.

## Remote defect enhances Fe-N₅ SACs for Fenton-like reactions

The role of coordination defects modulation in Fe-N₅ SACs for enhanced PMS activation was rigorously evaluated to achieve efficient BPA removal. Prior to comparative analysis of catalytic performances across FeN₅-SD$_x$ variants, systematic optimization of operational parameters (Supplementary Fig. 8a-d) was conducted. The FeN₅-SD₂/ PMS system showed the optimal performance at a catalyst dosage of 0.03 g L$^{-1}$ and a PMS concentration of 0.05 mM. As shown in Fig. 3a, all defect-engineered FeN₅-SD$_x$ catalysts outperformed pristine FeN₅,

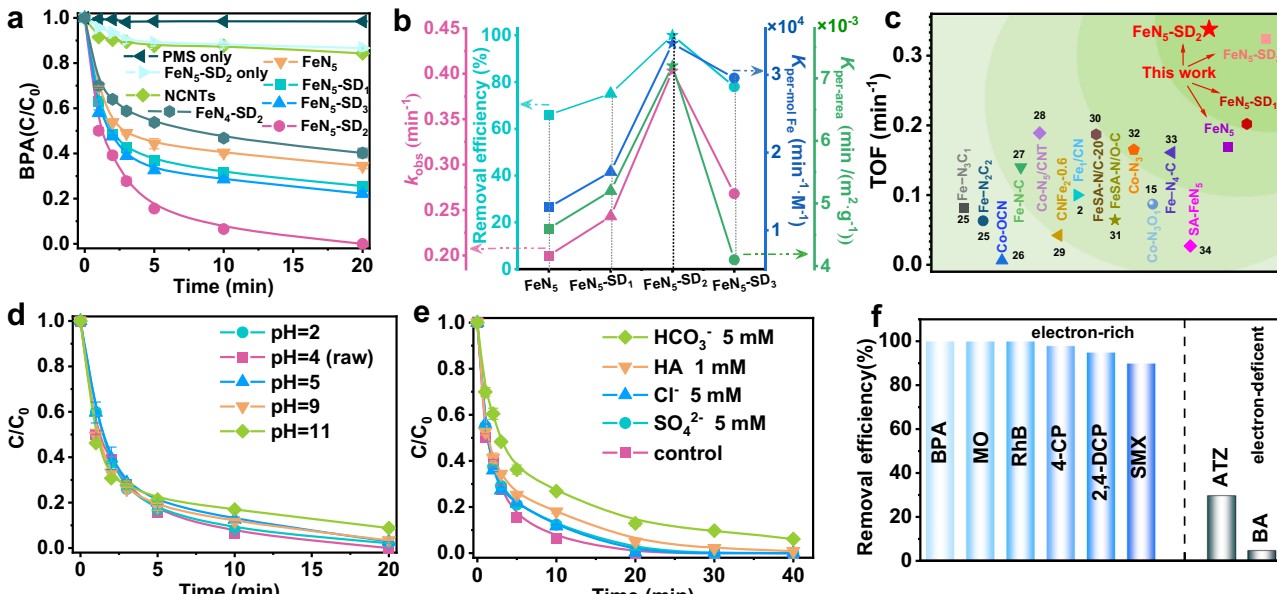

**Fig. 3 | Remote defect enhances Fe-N$_5$ SACs for Fenton-like reactions. a** The impact of diverse catalyst systems on the degradation of bisphenol A (BPA). Error bars indicate standard deviation derived from three parallel measurements. **b** Normalized rate constant of BPA degradation by per mole Fe, per surface area. **c** Comparison of TOF values for pollutant degradation in SAC/PMS systems featuring coordination environment regulation. The numbers indicated in the figure correspond to the respective references. The effects of **d** pH and **e** inorganic anions and HA on the removal of BPA in FeN$_5$-SD$_2$/PMS system. Error bars indicate standard deviation derived from three parallel measurements. **f** Different pollutants removal in FeN$_5$-SD$_2$/PMS system. Routine conditions: [pollutants] = 80 μM, [catalyst] = 0.03 g L$^{-1}$, [PMS] = 0.05 mM, temperature = 25 °C, without pH adjustment. Source data are provided as a Source Data file.

confirming that coordination defects enhance the intrinsic activity of FeN$_5$ SACs. FeN$_5$-SD$_2$ achieved complete BPA removal within 20 minutes, with a pseudo-first-order rate constant ($k_{obs}$) of 0.403 min$^{-1}$. This value is significantly higher than those of FeN$_5$ (0.200 min$^{-1}$), FeN$_5$-SD$_1$ (0.243 min$^{-1}$), and FeN$_5$-SD$_3$ (0.268 min$^{-1}$) (Fig. 3b), indicating an optimal defect density. Moderate levels accelerate reaction kinetics, while excessive defects apparently compromise activity. We performed kinetic normalization using both Fe-molar normalized rate constant ($K_{per-mol\ Fe}$) and specific surface area-normalized rate constant ($K_{per-area}$) to rule out other influencing factors (Fig. 3b). Inductively coupled plasma emission spectrometer (ICP-OES) data quantified metal contents across the FeN$_5$-SD$_x$ catalysts (Supplementary Table 3). Among these catalysts, FeN$_5$-SD$_2$ exhibited the highest $K_{per-mol\ Fe}$ (3.40 × 10$^4$ min$^{-1}$·M$^{-1}$), outperforming FeN$_5$ (1.30 × 10$^4$ min$^{-1}$·M$^{-1}$), FeN$_5$-SD$_1$ (1.75 × 10$^4$ min$^{-1}$·M$^{-1}$) and FeN$_5$-SD$_3$ (2.96 × 10$^4$ min$^{-1}$·M$^{-1}$). Similarly, $K_{per-area}$ values confirmed the highest surface-normalized rate for FeN$_5$-SD$_2$ (0.0072 min$^{-1}$/(m$^2$·g$^{-1}$)) (Supplementary Table 4). These metrics rule out Fe content and surface area as dominant performance factors. Thus, the improved performance primarily stems from electronic and geometric modulation induced by high coordination shell defects.

The TOF was employed to quantitatively evaluate the intrinsic reactivity against previously reported SAC-based PMS activation systems. As shown in Fig. 3c and Supplementary Table 5, FeN$_5$-SD$_2$ catalyst exhibits a significantly high TOF of 0.338 min$^{-1}$. This performance surpasses other recently reported coordination-regulated SACs in PMS systems[2,15,25–34]. Control experiments confirmed that neither FeN$_5$-SD$_2$ alone (BPA removal <10%) nor PMS alone (BPA removal <3%) made a significant contribution to contaminant elimination (Fig. 3a). This result excludes substantial adsorption or non-catalytic oxidative pathways. The pH tolerance and recyclability of FeN$_5$-SD$_2$ were systematically assessed to evaluate its operational robustness. The FeN$_5$-SD$_2$/PMS system showed negligible pH dependence across a wide pH range (2–11) (Fig. 3d and Supplementary Fig. 9), achieving >98% BPA removal under all tested conditions. Cycling tests showed excellent

stability, with over 95% pollutant removal maintained across four consecutive runs (Supplementary Fig. 10). To further evaluate catalyst durability, post-reaction XRD and XPS analyses were conducted. XRD patterns revealed no significant structural changes after cycling (Supplementary Fig. 11), while XPS C 1s spectra showed no notable increase in the C−O peak, indicating strong resistance to carbon oxidation (Supplementary Fig. 12). These results confirm that the FeN$_5$-SD$_x$ catalysts retain their structural integrity under reaction conditions. In addition, inductively coupled plasma mass spectrometry (ICP-MS) analysis detected minimal iron leaching (85.67 μg L$^{-1}$), well below the Class III surface water regulatory limit (0.3 mg L$^{-1}$)[26]. Control experiments further demonstrated that dissolved Fe ions contributed less than 15% removal even at concentrations up to 1 mg L$^{-1}$ (Supplementary Fig. 13), ruling out a homogeneous Fenton-like mechanism.

The system further displayed remarkable resistance to common environmental interferents, including inorganic anions (Cl$^-$, SO$_4^{2-}$, HCO$_3^-$) and humic acid (HA) (Fig. 3e and Supplementary Fig. 14). Field applicability tests in real water matrices (tap water, river water) demonstrated sustained efficacy, achieving > 97% BPA removal within 20 min (Supplementary Fig. 15), highlighting operational reliability in complex matrices. The FeN$_5$-SD$_2$/PMS system demonstrated distinct Fenton-like catalytic behavior across structurally diverse organic pollutants (Fig. 3f). Target analytes spanned multiple classes: dyes (rhodamine B, RhB; methyl orange, MO), chlorinated phenolics (2,4-dichlorophenol, 2,4-DCP; 4-chlorophenol, 4-CP), pharmaceutical contaminants (sulfamethoxazole, SMX), and other contaminants (atrazine, ATZ; benzoic acid, BA). Notably, substrates functionalized with electron-donating groups demonstrated enhanced degradation efficiencies, contrasting sharply with the recalcitrance of electron-withdrawing group-bearing species. Specifically, ATZ and BA, recognized as radical-specific probes[35], exhibited minimal degradation. This marked substrate selectivity strongly implies the prevalence of non-radical oxidation pathways over conventional radical-mediated mechanisms in the FeN$_5$-SD$_2$/PMS system.

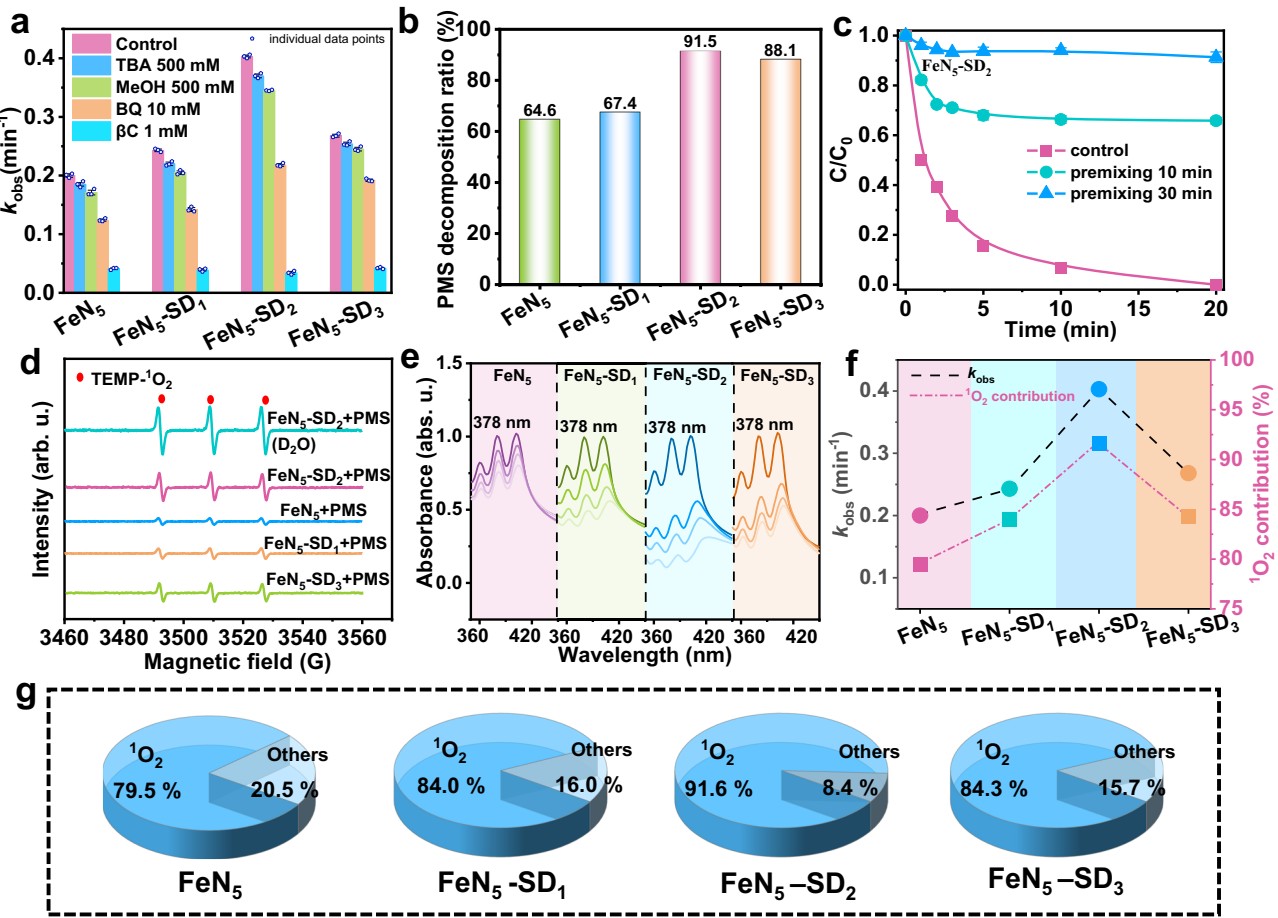

**Fig. 4 | Roles of ROS during Fenton-like reactions. a** Comparison of degradation kinetics under various quenching conditions. Error bars indicate standard deviation derived from three parallel measurements. **b** Decomposition rate of PMS in different systems. **c** Effect of premixing in FeN$_5$-SD$_2$/PMS system on bisphenol A (BPA) removal. Error bars indicate standard deviation derived from three parallel measurements. **d** EPR spectra of $^1O_2$ captured by TEMP. **e** 9,10-diphenylanthracene (DPA) signals in different systems. Darker shades represent earlier time points (0 min) and lighter shades represent later time points (20 min). **f** Trends in the $k_{obs}$ and $^1O_2$ contribution for various catalysts. **g** The contribution of ROS in different system. Source data are provided as a Source Data file.

## Roles of ROS during Fenton-like chemistry

Reactive species generation includes radical pathways, such as superoxide radicals ($O_2^{•-}$), •OH, and $SO_4^{•-}$, as well as non-radical pathways like $^1O_2$, high-valence metal oxidation, and ETP. To identify the primary active species in the FeN$_5$-SD$_x$/PMS system and understand how high coordination environments modulate their production, systematic scavenging assays were conducted.

Quenching experiments were first performed to determine the reactive oxygen species (ROS) involved (Fig. 4a). Tert-butanol (TBA) and methanol (MeOH), employed as scavengers for •OH and $SO_4^{•-}$, respectively (Supplementary Fig. 16), had negligible effects on BPA degradation across all FeN$_5$/PMS and FeN$_5$-SD$_x$/PMS systems, indicating minimal contributions from these radicals. In contrast, p-benzoquinone (p-BQ) partially inhibited BPA degradation (Supplementary Fig. 17), suggesting the involvement of $O_2^{•-}$. However, given its limited redox potential (−0.33 V vs. NHE), $O_2^{•-}$ is unlikely to drive BPA degradation directly, implying its role as an intermediate in ROS generation[36,37]. These findings suggest that non-radical pathways primarily govern BPA degradation.

To further explore this, additional quenching experiments using 2,2,6,6-tetramethylpiperidine (TEMP) as a $^1O_2$ scavenger revealed a near-complete inhibition of BPA degradation across all systems (Supplementary Fig. 18). Similarly, β-carotene (βC) and furfuryl alcohol

(FFA), both known $^1O_2$ quenchers[38], exhibited lower PMS decomposition efficiencies compared to other scavengers and demonstrated comparable inhibitory effects (Supplementary Figs. 19–20)[39]. These results strongly support $^1O_2$ as the dominant reactive species in BPA degradation within FeN$_5$/PMS and FeN$_5$-SD$_x$/PMS systems. To examine the role of high-valence iron-oxo species, dimethyl sulfoxide (DMSO) was introduced as a selective trapping agent[40]. Despite its presence, both FeN$_5$/PMS and FeN$_5$-SD$_x$/PMS systems maintained efficient degradation performance (Supplementary Fig. 21), suggesting minimal involvement of high-valence iron-oxo species in BPA oxidation. Moreover, PMS decomposition remained unaffected by the presence of BPA, with the FeN$_5$-SD$_2$/PMS system exhibiting a significantly higher PMS decomposition rate than FeN$_5$, FeN$_5$-SD$_1$, and FeN$_5$-SD$_3$ (Fig. 4b, Supplementary Fig. 22), correlating with its superior pollutant degradation efficiency. Notably, pre-mixing the catalyst with PMS before adding BPA led to a marked suppression of BPA removal (Fig. 4c, Supplementary Figs. 23–24), contradicting the expected characteristics of a direct electron-transfer pathway[7]. To further investigate this aspect, graphite electrodes coated with FeN$_5$-SD$_x$ were employed in galvanic oxidation processes (GOP)[41]. In this setup, PMS and BPA were exposed to separate half-cells connected by a salt bridge to isolate electron transfer as the only possible oxidation pathway (Supplementary Fig. 25a). However, no current was detected, and BPA

remained undegraded after 12 h (Supplementary Fig. 25b–e). These findings confirm that none of the $FeN_5$-$SD_x$/PMS systems facilitate a direct electron-transfer mechanism.

The quenching experiment results were corroborated by electron paramagnetic resonance (EPR) analysis (Fig. 4d). The addition of TEMP led to an enhanced triplet EPR signal characteristic of $^1O_2$ in all $FeN_5$-$SD_x$/PMS systems. The signal intensity followed the order $FeN_5$-$SD_2$ > $FeN_5$-$SD_3$ > $FeN_5$-$SD_1$ > $FeN_5$, aligning with the observed $k_{obs}$ values. In contrast, using 5,5-dimethyl-1-pyrroline-N-oxide (DMPO) to detect •OH and $SO_4^{•-}$ yielded negligible signals in the $FeN_5$-$SD_2$ system (Supplementary Fig. 26), confirming the absence of these radicals. The TEMP-$^1O_2$ signal intensity decreased upon the addition of p-BQ (Supplementary Fig. 27), indicating that $O_2^{•-}$ serves as an intermediate in $^1O_2$ formation. Control experiments under $O_2$, air, and $N_2$ atmospheres revealed negligible differences in degradation efficiency (Supplementary Fig. 28), indicating that dissolved oxygen (DO) is not essential to the reaction. This suggests that $^1O_2$ generation proceeds via direct activation of PMS at $FeN_5$ sites, independent of external oxygen input.

Further evidence for the critical role of $^1O_2$ was obtained through solvent exchange experiments using deuterium oxide ($D_2O$), which extends the lifetime of $^1O_2$[42]. As shown in Fig. 4d and Supplementary Fig. 29, the TEMP-$^1O_2$ signal intensity increased, and BPA degradation accelerated in $D_2O$ relative to $H_2O$, reinforcing the significance of $^1O_2$ in the oxidation process. The production of $^1O_2$ was further validated through the oxidation of 9,10-diphenylanthracene (DPA)[43], where the characteristic absorption peak at 378 nm progressively diminished over time in all $FeN_5$-$SD_x$/PMS systems (Fig. 4e), with the most pronounced effect observed in $FeN_5$-$SD_2$.

Kinetic analysis based on pseudo-first-order kinetics[44] revealed a volcano-type correlation between defect content and $^1O_2$ generation. Notably, the proportion of $^1O_2$ production closely matched the BPA removal efficiency across the $FeN_5$-$SD_x$/PMS systems (Fig. 4f, g). These findings collectively demonstrate that although variations in high coordination environments within Fe-$N_5$ structures do not alter the fundamental reaction mechanism, they significantly enhance $^1O_2$ production, thereby improving catalytic performance for water purification.

## Mechanistic understanding of the coordination structure modulation in $FeN_5$ SACs

To discern the active sites in $FeN_5$-$SD_2$ for PMS activation, an experimental study was conducted using potassium thiocyanate (KSCN) poisoning. $SCN^-$ has a high affinity for Fe and poisons the single-atom Fe sites during the Fenton-like reaction[45]. The introduction of KSCN led to a significant decrease in BPA degradation, with the $k_{obs}$ value dropping from 0.403 to 0.025 min$^{-1}$ (Supplementary Fig. 30). This result unequivocally confirms the dominant role of single-atom Fe sites in PMS activation within the $FeN_5$-$SD_2$ system.

To further elucidate the intrinsic catalytic mechanism, density functional theory (DFT) calculations were conducted to investigate the impact of higher coordination shell modulation in Fe-$N_5$ SACs on PMS activation. The total density of states (DOS) for $FeN_5$, $FeN_5$-$SD_1$, $FeN_5$-$SD_2$, and $FeN_5$-$SD_3$ revealed a high concentration of electronic activity at the Fe center (Fig. 5a), attributed to nonzero Fe-3d occupation near the Fermi level. Notably, the overlap of Fe d-orbitals with the DOS near the Fermi level signifies enhanced electron transfer capability upon Fe incorporation. In both $FeN_5$ and $FeN_5$-$SD_x$, the population of antibonding states is governed by the Fe d-orbitals. A d-band closer to the Fermi level suggests lower occupancy of these states, which promotes PMS adsorption[27,46]. The d-band center of $FeN_5$-$SD_x$ is upshifted relative to $FeN_5$, shifting closer to the Fermi level in the order of $FeN_5$ (−2.385 eV) < $FeN_5$-$SD_1$ (−2.295 eV) < $FeN_5$-$SD_2$ (−2.251 eV) < $FeN_5$-$SD_3$ (−2.108 eV). Thus, structural defects in the higher coordination shell induce a positive shift in the d-band center, optimizing electronic occupancy and enhancing PMS adsorption.

To clarify PMS adsorption preferences, the adsorption configurations and energies for three oxygen atoms on $FeN_5$, $FeN_5$-$SD_1$, $FeN_5$-$SD_2$ and $FeN_5$-$SD_3$ were calculated. In all cases, the terminal oxygen bonded to sulfur exhibited the strongest adsorption, identifying it as the preferred site for PMS activation (Supplementary Figs. 31–34). The optimized structures also show a gradual decrease in adsorption energy ($E_{ads}$) with increasing defect density, consistent with the DOS results (Fig. 5b). Specifically, $FeN_5$ exhibits a weaker adsorption energy (−1.72 eV), whereas $FeN_5$-$SD_3$ displays a stronger PMS binding affinity (−1.91 eV), suggesting that higher coordination shell defects enhance the interaction between Fe sites and PMS molecules. However, the observed $k_{obs}$ follows a volcano trend, with $FeN_5$-$SD_2$ exhibiting the highest Fenton-like activity due to its moderate adsorption energy. The lower reactivity on $FeN_5$-$SD_3$ results from excessive binding strength, which inhibits catalytic turnover by saturating active sites[47]. These findings highlight that controlled tuning of higher coordination shells can strategically shift the d-band center, optimizing electronic occupancy to enhance PMS adsorption and accelerate Fenton reaction kinetics.

Charge density analysis and Bader charge calculations further elucidate the electron transfer dynamics between Fe-$N_5$ sites and PMS. Charge density difference maps reveal robust electron transfer between $FeN_5$/$FeN_5$-$SD_x$ and PMS (Fig. 5c, Supplementary Fig. 35a–d), with an evident charge redistribution between the Fe atom and adjacent N atoms, indicating preferential electron flow from Fe to PMS. Notably, $FeN_5$-$SD_2$ exhibits the highest electron transfer (0.811 e), surpassing $FeN_5$ (0.795 e), $FeN_5$-$SD_1$ (0.798 e), and $FeN_5$-$SD_3$ (0.807 e). Bader charge analysis confirms that tuning higher coordination shells significantly modulates Fe-centered charge density, thereby enhancing electron transfer efficiency. Electrochemical impedance spectroscopy (EIS) and chronoamperometry further support the improved charge transfer interactions in $FeN_5$-$SD_x$. In Supplementary Fig. 36a, $FeN_5$-$SD_2$ exhibits the smallest semicircle diameter, while current responses at the $FeN_5$-$SD_2$ electrode increase significantly upon PMS injection (Supplementary Fig. 36b), confirming that moderate defects in higher coordination shells enhance PMS activation.

To further understand PMS activation in $FeN_5$-$SD_x$, we calculated the energy barriers along the $^1O_2$ evolution pathway (Fig. 5d-e). We first compared the energy barriers for cleaving three key PMS bonds (O-O, S-O, and O-H) on $FeN_5$ and $FeN_5$-$SD_x$. As shown in Fig. 5d and Supplementary Figs. 37-41, the O−O bond consistently has the lowest barrier, indicating it is the easiest to break and thus the preferred initiation step. Optimized free energy diagrams depict the sequential steps of $^1O_2$ generation (Fig. 5f and Supplementary Figs. 42-45): PMS → *PMS (I) → *OH + *$SO_4$ (II) → *OH + $H_2SO_4$ (III) → *OH + *OH (IV) → *O (V) → *O + *O (VI) → $^1O_2$. In this pathway, PMS undergoes O−O bond cleavage, forming into *OH and *$SO_4$ fragments. The *OH adsorbing onto $FeN_5$, while *$SO_4$ exothermically convert to $H_2SO_4$. Subsequently, *OH transforms into *O, which recombines into $^1O_2$. The highest energy barrier occurs at step I (*PMS → II *OH + *$SO_4$), suggesting it as the rate-determining step (RDS) in $^1O_2$ generation. Introducing moderate defects in the higher coordination shell lowers this barrier from 0.85 to 0.73 eV, thereby facilitating $^1O_2$ formation during PMS activation. Furthermore, O−O bond length ($l_{O−O}$) analysis (Fig. 5g) reveals $FeN_5$-$SD_2$ exhibits the longest bond (1.474 Å), indicating that moderate defects extend the O−O bond, lowering its cleavage energy barrier and enhancing $^1O_2$ evolution kinetics.

Theoretical insights confirm that defect site modulation in the higher coordination shell of Fe-$N_5$ SACs optimizes Fe 3d orbital electron distribution. This modulation facilitates PMS adsorption and explains the observed volcano trend in catalytic activity. Furthermore, it enhances charge migration at the reaction interface and extends the O−O bond length, thereby reducing the energy barrier for O−O bond cleavage. As a result, key reaction intermediates form more efficiently, selectively promoting $^1O_2$ production. These findings provide

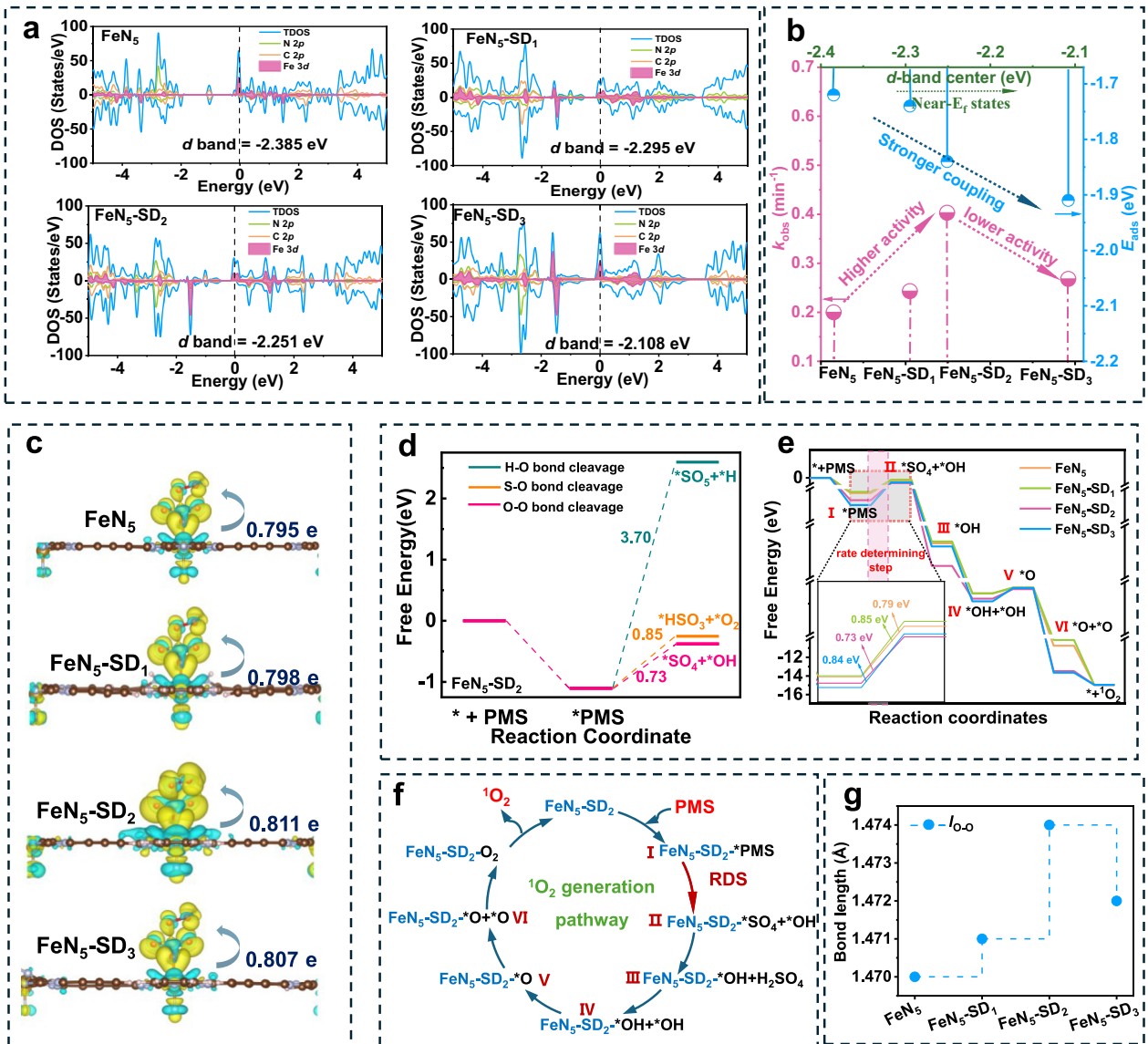

**Fig. 5 | The mechanistic of high coordination shell defects in regulating Fe-N₅ active sites. a** Density of states of FeN₅, FeN₅-SD₁, FeN₅-SD₂ and FeN₅-SD₃.
**b** Correlation among the oxidation capacity, coupling strength to PMS, and *d*-band structure of FeN₅, FeN₅-SD₁, FeN₅-SD₂ and FeN₅-SD₃. **c** Electron density difference diagrams and Bader charges for PMS adsorbed on FeN₅, FeN₅-SD₁, FeN₅-SD₂ and FeN₅-SD₃, with blue and yellow indicating electron depletion and accumulation,

respectively. Crystal structures visualized using VESTA software[49]. **d** Calculated potential energy diagrams for various bond-cleavage pathways during PMS decomposition on FeN₅-SD₂. **e** Calculated potential energy diagrams for the ¹O₂ formation pathway. **f** Schematic diagram of ¹O₂ generation process. **g** Length of the O – O bonds after adsorption of PMS adsorbed on different catalysts. Source data are provided as a Source Data file.

fundamental theoretical validation of enhanced PMS activation through precise defect coordination engineering.

**Possible removal pathways of BPA and toxicity analysis**
The proposed BPA degradation pathways in the FeN₅-SD₂/PMS system were deduced by combining liquid chromatography-mass spectrometry (LC-MS) results (Supplementary Fig. 46) with gas chromatography-mass spectrometry (GC-MS) spectrometry data (Supplementary Figs. 47-48 and Supplementary Table 6). As shown in Supplementary Fig. 49, two primary pathways were identified based on the detected intermediates. In Pathway 1, BPA undergoes hydroxylation to form hydroxylated intermediates (e.g., P1), which are further oxidized to P2-P4, containing ketone and carboxyl groups, accompanied by aromatic ring cleavage. This pathway is likely driven by the predominant presence of ¹O₂ in the reaction medium, which exhibits high reactivity towards hydroxylated intermediates. In Pathway 2, β-

scission of the quaternary carbon connecting the two phenyl rings leads to the formation of isopropenylphenol (P6), which subsequently converts into 2-phenyl-1-propene (P7) and p-hydroxyacetophenone (P8). These intermediates are further oxidized to form open-chain compounds, including hexa-1,5-diene-3,4-diol (P9), phenol (P10), and p-benzoquinone (P11). Ultimately, the aromatic structures are mineralized into CO₂ and H₂O through sequential oxidation steps.

To assess the real-world applicability of the catalytic system, we first predicted the toxicity of BPA and its intermediates using the Ecological Structure-Activity Relationships (ECOSAR) and Toxicity Estimation Software Tool (T.E.S.T.) systems (Fig. 6a-b, Supplementary Fig. 50, and Supplementary Table 7). The half-lethal concentration (LC₅₀) values for BPA in Fathead minnow and Daphnia magna were within the "toxic" range. However, most degradation intermediates exhibited reduced toxicity. For instance, the 96 h LC₅₀ value for BPA in Fathead minnow was 3.24 mg L⁻¹, while intermediates like P8, P10, and

P11 showed significantly higher $LC_{50}$ values of 35.51 mg L$^{-1}$, 38.69 mg L$^{-1}$, and 31.96 mg L$^{-1}$, respectively, indicating a reduced toxicity classification (Fig. 6a). Similarly, the 48-h $LC_{50}$ value for BPA in Daphnia magna was 1.58 mg L$^{-1}$, while most intermediates showed a marked increase in $LC_{50}$ (Fig. 6b). As illustrated in Supplementary Fig. 50, BPA is classified as a "developmental toxicant", while P8 and P9 are classified as "developmental non-toxicants". Supplementary Table 7 shows that both acute and chronic toxicity values of the oxidized products decreased as degradation progressed. Notably, advanced oxidation products such as P9, P10, and P11 were predicted to be "not harmful", underscoring the detoxification capacity of the FeN$_5$-SD$_x$ system.

Beyond computational predictions, experimental toxicity validation was performed using zebrafish and microbial viability assays. Zebrafish exposed to BPA solution exhibited early mortality and behavioral abnormalities over a 12 h period. In contrast, those cultured in the post-reaction BPA solution from the FeN$_5$-SD$_2$/PMS system showed no abnormalities or fatalities (Supplementary Fig. 51), providing strong evidence for the biocompatibility of the treated water. Finally, flow cytometry analysis was conducted to evaluate the cyto-toxic effects of treated and untreated solutions on microbial populations (Fig. 6c-d). After 40 minutes of treatment, the proportion of live cells increased from 35.1% to 70.6%, while dead cells decreased from 3.92% to 0.73%. These results highlight the robust detoxification capability of the FeN$_5$-SD$_2$/PMS system.

### Application prospect of FeN$_5$-SD$_2$
The exceptional catalytic performance of the FeN$_5$-SD$_2$/PMS system has led to its widespread adoption in practical applications. Through synergistic integration of advanced oxidation processes (AOPs) with polyvinylidene fluoride (PVDF) membrane confinement technology, we engineered a catalytic filtration platform that effectively restricts organic pollutant diffusion while conferring autonomous self-cleaning functionality. Fabrication involved vacuum-assisted deposition of FeN$_5$-SD$_2$ suspensions onto PVDF substrates, producing membranes with elasticity and malleability suitable for continuous-flow wastewater treatment configurations (Fig. 6e-g). Comparative hydraulic performance evaluation of pristine PVDF and PVDF/FeN$_5$-SD$_2$ membranes was conducted through systematic analysis of mass transfer dynamics and solute rejection characteristics for BPA contaminated solution (Fig. 6h). The functionalized membrane exhibited stable performance over 20 consecutive cycles (10 h), maintaining a water flux of 910.1 L·m$^{-2}$·h$^{-1}$ with 5.2% solute rejection. This flux approaches that of unmodified PVDF (946.6 L·m$^{-2}$·h$^{-1}$, 3.0% rejection), while offering improved contaminant interception.

Continuous-flow evaluations demonstrated broad-spectrum contaminant elimination, sustaining >95% BPA removal efficiency over 10 h operational duration. The system exhibited universal remediation capability for recalcitrant pollutants including 4-CP, phenol (PE), and SMX, achieving near-quantitative removal (Fig. 6i). Validation across diverse aqueous matrices (TW: tap water; SW: surface water; Fig. 6j) confirmed consistent performance in complex ionic environments, with iron leaching levels (80 μg·L$^{-1}$) remaining 75% below China's Class III surface water standard (0.3 mg·L$^{-1}$), verifying catalyst stability. The confluence of high contaminant removal efficiency, matrix tolerance, and exceptional metal retention positions the PVDF/FeN$_5$-SD$_2$ system as a technologically viable solution for advanced water remediation applications.

## Discussion
The strategic engineering of primary and higher coordination spheres represents an innovative design paradigm for tailoring SACs in Fenton-like chemistry. In this study, we engineered a model catalyst with a meticulously defined axial Fe-N$_5$ active center and defect sites in higher coordination shells, employing a pyrolysis-free methodology. The

FeN$_5$-SD$_x$ configuration demonstrates advantages in both PMS activation and selective $^1O_2$ generation, maintaining robust stability with a TOF value of 0.338 min$^{-1}$ within FeN$_5$-SD$_2$/PMS, outperforming FeN$_5$/PMS system and other reported SAC-based systems. Mechanistic investigations reveal that the efficiency of PMS activation and the selective generation of ROS are highly dependent on the extent of defects in the higher coordination shells of SACs. Controlled tuning of higher coordination shells can strategically shift the Fe d-band center, optimizing electronic occupancy to enhance PMS adsorption. Only FeN$_5$-SD$_2$ with an optimal level of defects and moderate adsorption energy can effectively elongate the O–O bond of PMS, thereby reducing the cleavage energy barrier and accelerating the kinetics of $^1O_2$ generation. Following immobilization on a PVDF membrane, FeN$_5$-SD$_2$ was integrated into a continuous flow system, achieving sustained (>95% BPA removal over 10 h) and broad-spectrum elimination of recalcitrant contaminants (4-CP, PE, SMX) across diverse aqueous matrices. This study offers profound insights into the relationships between local atomic environments and properties of Fe-N$_5$ SACs concerning PMS activation in water purification processes.

## Methods
### Chemicals and reagents
1,2-Dicyanobenzene (DCB), 1,2,4,5-tetracyanobenzene (BTC), ethylene glycol, N, N-dimethylformamide (DMF), 2,2′-azino-bis (3-ethylben-zothiazoline-6-sulfonic acid) diammonium salt (ABTS), nitrogen-doped carbon nanotubes (NCNTs), and β-carotene (βC) were obtained from Shanghai Macklin Biochemical Co., Ltd. Ferric chloride (FeCl$_3$), potassium thiocyanate (KSCN), bisphenol A (BPA), and nitro-blue tetrazolium chloride (NBT) were supplied by Aladdin Industrial Corporation. Potassium monopersulfate triple salt (PMS, 2KHSO$_5$·KHSO$_4$·K$_2$SO$_4$), 5,5-dimethyl-1-pyrroline-N-oxide (DMPO), and 2,2,6,6-tetramethylpiperidinyloxyl (TEMP) were purchased from Sigma-Aldrich Chemical Co., Ltd. Potassium iodide (KI), 1,8-diazabi-cyclo[5.4.0]undec-7-ene (DBU), and p-benzoquinone (p-BQ) were obtained from J&K Chemical Co., Ltd. Tert-butyl alcohol (TBA), methanol (MeOH), ethanol (EtOH), hydrochloric acid (HCl), furfuryl alcohol (FFA), and dimethyl sulfoxide (DMSO) were purchased from Sinopharm Chemical Reagent Co., Ltd. All chemicals were of analytical grade or higher and were used without further purification. Throughout the study, deionized water was used for all experiments.

### Synthetic strategy for FeN$_5$-SD$_x$
Synthesis of FeN$_5$-SD$_2$ was achieved employing a one-step protocol. In detail, a mixture comprising BTC (0.7 mmol), DCB (0.7 mmol), and FeCl$_3$ (0.085 mmol) was dispersed in a solvent system comprising ethylene glycol and DMF, followed by ultrasonication for 30 minutes. Subsequently, DBU (0.15 mmol) was introduced as a catalyst, initiating the reaction under a nitrogen atmosphere for 30 minutes. Introduction of 100 mg of NCNTs to the solution ensued, which was then stirred for 24 h. The resultant mixture underwent heating at 180 °C for 72 h, followed by gradual cooling to room temperature. The precipitated solid was harvested via centrifugation and consecutively washed with 1 M hydrochloric acid, anhydrous ethanol, and deionized water. The ensuing product, designated as FeN$_5$-SD$_2$, was vacuum dried at 70 °C for 48 h.

For comparative analysis, the molar weight ratio of DCB to the combined weight of BTC and DCB was adjusted to 0%, 25% and 75%, resulting in the production of samples denoted as FeN$_5$, FeN$_5$-SD$_1$ and FeN$_5$-SD$_3$, respectively.

### Characterizations
X-ray diffraction (XRD) patterns were recorded on a PANalytical Empyrean powder diffractometer using Cu Kα radiation (λ = 0.1541 nm). Fourier transform infrared (FTIR) spectra were acquired with a Thermo Scientific Nicolet iS20 spectrometer using KBr pellets. X-ray photoelectron spectroscopy (XPS) was performed on a

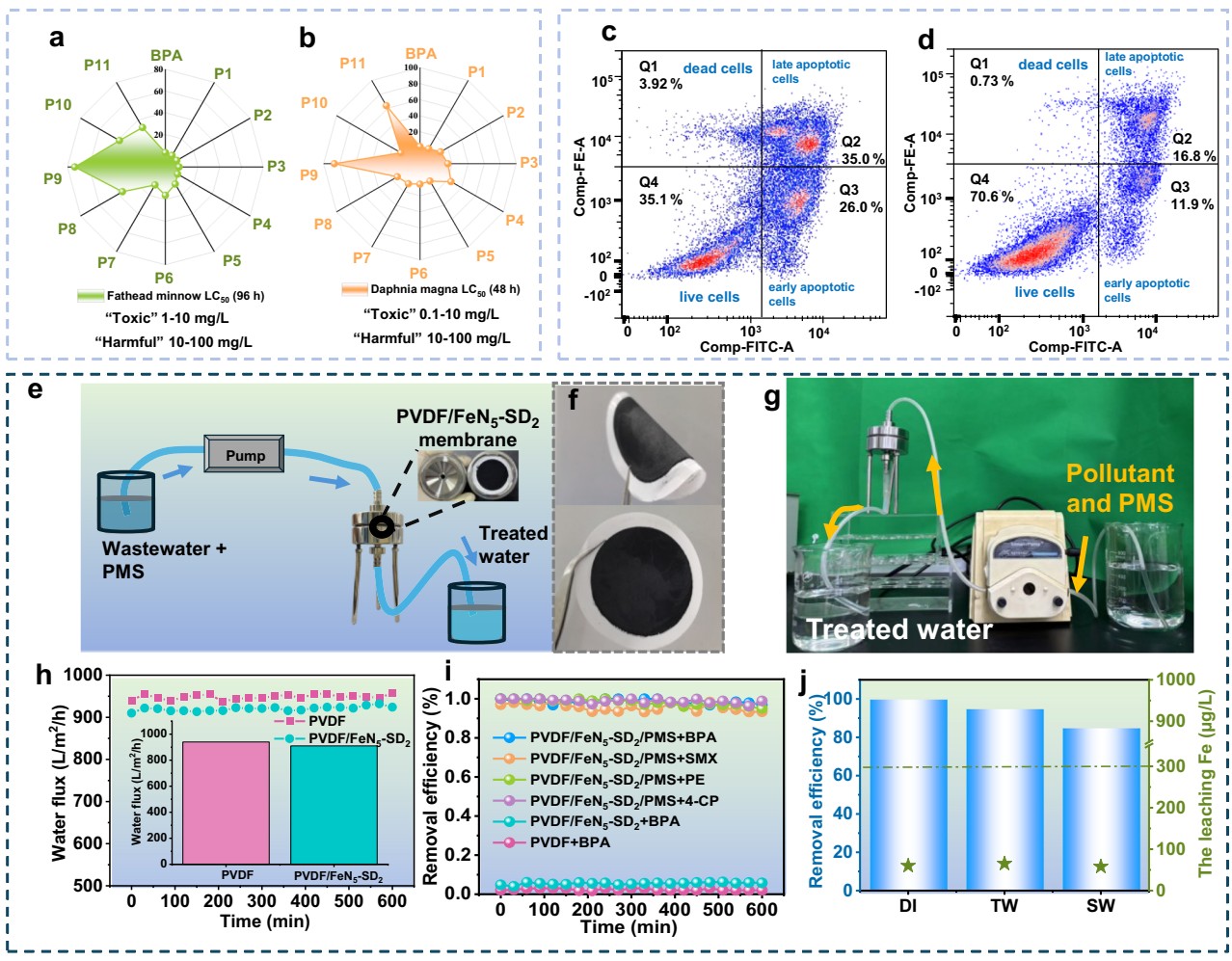

**Fig. 6 | Toxicity analysis and environmental application. a** Fathead minnow $LC_{50}$-96 h and **b** Daphnia magna $LC_{50}$-48 h of bisphenol A (BPA) and its degradation intermediates. Flow cytometry analysis of *E. coli* cells during BPA degradation in $FeN_5$-$SD_2$/PMS system at different reaction time: **c** 0 min, **d** 40 min. The four sections in flow cytometry images represent as below: Q1 dead cells; Q2 late apoptotic cells; Q3 early apoptotic cells; Q4 live cells. **e** Schematic illustration of wastewater treatment process. **f** PVDF/$FeN_5$-$SD_2$ membrane. **g** Photograph of the wastewater treatment experimental equipment. **h** Water flux of different membranes during the continuous-flow process. Insets show the averaged water flux values. **i** Removal efficiency of different systems. **j** BPA removal efficiency and Fe leaching of PVDF/$FeN_5$-$SD_2$ membrane system in different water matrices. Source data are provided as a Source Data file.

PerkinElmer PHI 5000 C system equipped with a monochromatized Al Kα source (200 W). Sample morphology was examined by field emission scanning electron microscopy (FE-SEM, HITACHI Regulus 8100), and high-resolution transmission electron microscopy (HRTEM) images were obtained using a FEI TalosF200S microscope. The Fe content of the samples was quantified via inductively coupled plasma optical emission spectroscopy (ICP-OES, Agilent 720ES), while iron leaching concentrations were measured with an Agilent 7700 ICP-MS. High-angle annular dark-field scanning transmission electron microscopy (HAADF-STEM) images were recorded on a FEI Titan Themis 60-300 TEM/STEM. Fe K-edge X-ray absorption fine structure (XAFS) spectra were collected at beamline BL44B2 of the SPring-8 synchrotron (Japan), where the storage ring operated at 8.0 GeV with a maximum current of 250 mA. Data were acquired in transmission mode using a Si (111) double-crystal monochromator and ionization chambers under ambient conditions. Electron paramagnetic resonance (EPR) measurements were conducted on a Bruker A300 spectrometer.

### Evaluation of catalyst decontamination performance in batch experiments

The catalytic performance was evaluated through BPA degradation experiments to assess the catalyst's ability to activate PMS. The

standard experimental procedure was conducted as follows: In a thermostatic magnetic stirring system (25 °C, 0.7 × $g$), 0.03 g L$^{-1}$ catalyst was uniformly dispersed in 80 μM BPA solution, followed by the injection of 0.05 mM PMS to initiate the catalytic reaction. Samples were collected at predetermined intervals (with immediate addition of 0.5 mL methanol to quench the reaction) and filtered through 0.22 μm aqueous membranes for phase separation. The residual BPA concentration in supernatant was quantified using high-performance liquid chromatography (HPLC). All experiments were performed with 2-3 parallel tests, with data presented as mean ± standard deviation. Error bars were derived from parallel experimental results. The initial pH was adjusted using 1 M HCl or 1 M NaOH, with pH 4.0 used as the default unless specified otherwise. For reusability tests, the catalyst was recovered after each cycle by filtration and rinsing with deionized water.

### Continuous-flow experiment for water treatment

In the practical implementation of membrane technology, a 5 mg sample of $FeN_5$-$SD_2$ underwent ultrasonic dispersion in deionized water. It was subsequently deposited onto a polyvinylidene fluoride (PVDF) membrane via vacuum-assisted filtration, resulting in the fabrication of a PVDF/ $FeN_5$-$SD_2$ composite membrane. After loading the

membrane into the reactor, a composite solution comprising 80 µM of BPA and 0.10 mM PMS was delivered to the membrane reactor using a peristaltic pump. Samples were collected at the reactor outlet and subjected to filtration for the precise determination of BPA concentrations.

## Quantification of organic contaminants

The concentrations of organic dyes, rhodamine B (RhB) and methyl orange (MO), were measured using a UV–vis spectrophotometer at 552 nm and 465 nm, respectively. Contaminant concentrations were further analyzed by high-performance liquid chromatography (HPLC) using a Waters e2695 system equipped with a UV detector and a C18 column (4.6 mm × 150 mm, 5 µm). Detection wavelengths and mobile phases were optimized for each compound: BPA at 225 nm, 2,4-DCP at 284 nm, and ATZ at 224 nm using methanol/water (70:30, v/v); SMX at 265 nm with methanol/water (50:50, v/v); 4-CP and BA at 280 nm and 227 nm, respectively, with methanol/water (60:40, v/v). The degradation efficiency (η) of contaminants was calculated as:

$$\eta = \frac{C_0 - C_t}{C_0} \times 100\%$$

where $C_0$ and $C_t$ represent the initial and time-dependent contaminant concentrations, respectively. Total organic carbon (TOC) was determined using a Shimadzu TOC-L Analyzer.

For further identification of degradation products, liquid chromatography–mass spectrometry (LC/MS) was performed with an Agilent 1290 system coupled to a QQQ (Agilent 6460) tandem mass spectrometer.

Post-reaction BPA solutions (50 ppm) were acidified to pH 3.0 with HCl, mixed with 1 g NaCl, and transferred to a separatory funnel. Organic extraction was performed with dichloromethane (5 mL × 2), and the combined extracts were dried over $Na_2SO_4$, concentrated, and re-dissolved in 1 mL of acetone.

Degradation products were analyzed using GC/MS (Shimadzu QP2020 NX) equipped with a DB-17MS quartz capillary column (20 × 0.18 mm, 0.18 µm) and electron impact (EI) detector at 70 eV. Helium was used as the carrier gas at 1 mL/min. Injection volume was 1 µL in splitless mode. The oven program was: 40 °C for 3 min, ramp at 15 °C/min to 320 °C, hold for 6 min. Injector, interface, and ion source temperatures were 300, 300, and 230 °C, respectively.

## PMS concentration and active species identification

The concentration of PMS was measured via the ABTS method. Briefly, 0.1 mL of sample was mixed with 0.5 mL of 2 mM ABTS solution, 1 mL acetate buffer (pH 4), and 20 µL of 1.5 mM KI solution, then diluted to 3 mL with water. Absorbance at 415 nm (ε = 34 000 $M^{-1}$ $cm^{-1}$) was used to quantify PMS based on the generation of $ABTS^{\bullet+}$.

The active species composition in the catalytic system was elucidated through radical scavenger experiments and non-radical probe analysis. MeOH and TBA were employed as scavengers for •OH and $SO_4^{\bullet-}$ radicals, respectively, with a molar ratio of 1000:1 relative to PMS. Specific concentration parameters for other scavengers are provided in the figure captions. To further validate reactive species identification, EPR spectroscopy was performed using a Bruker Model A300 spectrometer under ambient conditions. The following trapping agents were utilized: DMPO for detecting •OH and $SO_4^{\bullet-}$ radicals, methanol-dissolved DMPO as a specific probe for $O_2^{\bullet-}$, and TEMP for singlet oxygen ($^1O_2$) identification. Systematic analysis of characteristic radical signal intensity variations provided mechanistic insights into the catalytic reaction. 9,10-diphenylanthracene (DPA) oxidation was employed to estimate the production of $^1O_2$ in the systems. A 100 µL DPA solution (1 g $L^{-1}$, in DMSO) was mixed with 1.8 mL deionized water. Subsequently, 1.6 mL of the time-dependent sample solution and 0.8 mL methanol (as a quencher) were added. The decomposition of

DPA was monitored using a UV-vis spectrophotometer, with the decomposition rate determined by the change in absorbance at 378 nm.

## Electrochemical measurements

Electrochemical impedance spectroscopy (EIS) and chronoamperometry were performed in a conventional three-electrode cell, with a Pt plate as counter electrode, Ag/AgCl as reference, and indium-tin oxide (ITO) glass as working electrode. The working electrode was prepared by dispersing 20 mg of catalyst in 300 µL isopropanol with 50 µL Nafion, sonicating, and coating onto pretreated ITO glass. After air drying, uncoated areas were masked with epoxy.

Galvanic oxidation processes (GOP) were carried out using PMS and BPA in separate half-cells (50 mL) connected via a KCl/agar salt bridge. The catalyst was immobilized on a graphite electrode for the GOP experiments.

## DFT computational methods

All DFT calculations were performed utilizing the Vienna Ab-initio Simulation Package (VASP). Perdew-Burke-Ernzerhof (PBE) functional within the generalized gradient approximation (GGA) method was employed to account for exchange-correlation effects. The projected augmented wave (PAW) method was utilized to incorporate core-valence interactions, while spin polarization was integrated into all computations. Plane wave expansions were truncated at an energy cutoff of 400 eV. Structural optimization was conducted with energy and force convergence thresholds set at $1.0 \times 10^{-4}$ eV and 0.05 eV $Å^{-1}$, respectively. Brillouin zone sampling was accomplished with a 2 × 2 × 1 grid centered at the gamma (Γ) point. Grimme's DFT-D3 methodology was employed to describe dispersion interactions.

The Gibbs free energy changes (ΔG) of the reaction are calculated using the following formula:

$$\Delta G = \Delta E + \Delta ZPE - T\Delta S$$

where ΔE is the electronic energy difference directly obtained from DFT calculations, ΔZPE is the zero-point energy difference, T is the room temperature (298.15 K) and ΔS is the entropy change.

## Calculation details of the turnover frequency

The turnover frequency (TOF) of each metal site, which represents the intrinsic activity of the catalyst could be calculated as follows:

$$\text{TOF}\left[\min^{-1}\right] = \frac{\text{moles of reactant converted}}{\text{moles of active sites} \times \text{reaction time}} = \frac{\Delta n(\text{pollutant}) \times M_{metal}}{m_0 \times \omega_{metal} \times t}$$

where Δn (pollutant) is the moles of pollutants converted, $M_{metal}$ is metal atomic weight, $m_0$ is catalyst mass, $\omega_{metal}$ is metal mass fraction, and t is reaction time.

## Toxicity assessment

Three healthy adult zebrafish (one female and two males) were placed together in a tank containing nutrient-rich medium for 12 h to induce spawning and maximize embryo yield. Fertilized embryos were subsequently rinsed thoroughly with deionized water to remove surface contaminants. Embryos at the eight-cell stage were identified under a microscope and collected for downstream experiments.

Exposure solutions were prepared according to the experimental design. Embryos in the experimental group were treated with the post-reaction solution from the $FeN_5$–$SD_2$ catalytic system. The blank group received only the nutrient medium, while the control group was exposed to a BPA solution. Embryos were distributed into 24-well culture plates, with ten embryos per group and one embryo per well. Each well contained 1 mL of nutrient medium mixed with 1 mL of the designated exposure solution. Embryo development was monitored

microscopically, and exposure solutions were refreshed every 24 h from the start of treatment until hatching.

Toxicity changes during BPA degradation in the $FeN_5$-$SD_2$/PMS system were assessed using LIVE/DEAD staining of *Escherichia coli* (*E. coli*) followed by flow cytometry. The *E. coli* strain was obtained from the American Type Culture Collection (ATCC), and beef extract was purchased from Beijing Coolabor Biotechnology Co., Ltd. Briefly, 10 μL of *E. coli* was inoculated into 10 mL of nutrient medium (3 g/L beef extract, 10 g/L tryptone, 5 g/L NaCl, pH 7.2) and incubated on a rotary shaker at 30 °C for 10 h. Cells were harvested by centrifugation at $402 \times g$ for 5 min and washed three times with phosphate-buffered saline (PBS, 0.5 M, pH 7.4).

Next, 1 mL of the prepared bacterial suspension was added to 10 mL of the reaction solution collected at different time points (0 and 40 min). After incubation for 9 h, the mixtures were centrifuged at $402 \times g$ for 5 min and washed three times with PBS. The cells were then stained with 50 μL propidium iodide (PI) and 50 μL SYTOX in the dark at 25 °C for 15 min. Cytotoxicity changes over time were subsequently analyzed by flow cytometry.

## Data availability

The data generated in this study are provided within the article and the Supplementary Information/Source Data file. Additional data are available from the corresponding authors upon request. Source data are provided with this paper[48].

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

## Acknowledgements
This study was supported by the National Natural Science Foundation of China (22276172 to T.Z., 22322602 to H.Y.Z., and 22278374 to Z.Q.H.), the Zhejiang Provincial Natural Science Foundation of China (LR23B070001 to H.Y.Z.), and Central Guiding Local Science and Technology Development Fund Projects (No.2025ZY01044 to T.Z. and H.Y.Z.).

## Author contributions
T.Z. conceived and planned the experiments, and S.J.J., W.X.T. and Y.W. conducted the synthesis, characterization, and catalytic performance tests. S.J.J. performed the theoretical studies. S.J.J., Z.Q.H., H.Y.Z. and W.X.T. conducted in collecting data and analyzing various characterizations. Y.L.H., T.Z., S.J.J., Y.Q.C. and S.S. wrote and revised the manuscript.

## Competing interests
The authors declare no competing interests.
