## [Transparent Peer Review File · Nature Communications]

Remote Tuning of Single-Atom Fe-N5 Sites via High-Coordination Defects for Enhanced Fenton-Like Water Decontamination

Corresponding Author: Professor Tao Zeng

Version 0:

Reviewer comments:

Reviewer #1

(Remarks to the Author)

The work (NCOMMS-25-25049) tries to regulate the high-shell coordination defects of a model SAC with well-defined axial Fe-N5 configurations to elucidate the impact of long-distance interactions on PMS activation. While the topic is conceptually interesting, it lacks clear innovation, particularly in the context of PMS activation for BPA removal, which has been extensively studied and offers limited novelty or practical potential in its current form. The central value of this work lies in its attempt to correlate high-shell coordination structures of metal centres with catalytic performance. However, the design of the probe catalyst, the core of this study, is questionable. As shown in Fig. 1a and described in the experimental section, varying amounts of DCB were used during synthesis, leading to different metal loadings across the FeN₅-SD_x series. This results in inconsistent iron concentrations during PMS activation, making it difficult to isolate the effect of coordination defects from the influence of overall Fe content. Since the metal centre is presumed to be the principal active site, the authors must clarify how the contribution of high-shell coordination defects to BPA degradation is distinguished from variations in metal loading. Furthermore, while the manuscript repeatedly emphasizes the role of high-shell coordination in PMS activation and BPA degradation, the structural evidence supporting these claims is weak. The coordination defects in FeN₅-SD_x are not well characterized. FTIR spectra show changes in other functional groups that are overlooked in the discussion, and XPS results indicate the coexistence of two Fe valence states, which appears inconsistent with the proposed structure. The stability of the proposed structures, particularly FeN₅-SD₃ as depicted in Fig. 1a, is also questionable and lacks adequate validation. In addition, several typos and formatting issues detract from the manuscript's clarity, for example, the ball colour for DCB appears to be mislabelled (carbon?), Fig. 1e incorrectly uses 'wavelength' instead of 'wavenumber', and abbreviations such as DCB and BTC should be introduced in full upon first mention. Overall, the conclusions presented are not sufficiently supported by the data and analysis in the current form. It is hard to recommend this manuscript for publication in Nature Communications.

Reviewer #2

(Remarks to the Author)

This manuscript presents an interesting work in single-atom catalysts (SACs) for Fenton-like water decontamination. The work elucidates the critical role of long-distance coordination defects in modulating the electronic structure and catalytic behavior of Fe-N5 active sites. While numerous SACs have been reported for enhancing Fenton-like reactions, this study distinguishes itself by achieving atomic-level engineering of ordered active sites coupled with rigorous establishment of structure-activity relationships through systematic methodologies. Notably, the authors develop a framework for understanding the remote effects of multi-shell coordination on reaction dynamics. The experimental design is rigorous and well-executed, addressing a timely and compelling research frontier. I think that this work could be accepted after minor revision, which might improve the quality of this work:

1. A detailed synthetic scheme is required to improve the understanding of these materials. Specifically, how are the phthalocyanine frameworks or defect-engineered phthalocyanine-like frameworks synthesized? What are the synthesis processes and underlying mechanisms involved?
2. If annealing treatment is applied to the catalyst, does it enhance the catalytic activity or stability of the axial Fe-N5 sites?

Does the pyrolysis-free synthetic strategy employed in your work effectively address the inherent carbon oxidation instability of conventional Fe-N-C SACs derived from thermal treatments? Please provide experimental evidence to verify the oxidation resistance of your conjugated polyphthalocyanine framework under operational conditions.

3. There have been numerous previous reports on constructing axially coordinated single-atom materials on carbon nanotubes, such as *Adv. Sci.* 2023, 10, 2206107. What are the specific advantages or novel aspects of this study compared to existing work in the field?

4. Fig. 1a displays the configurations of FeN5, FeN5-SD1, FeN5-SD2 and FeN5-SD3, but the accompanying explanation is inadequate. Please furnish more experimental or computational data to rigorously validate the rationality and objectivity of these structural configuration.

5. Most studies reporting the activation of PMS to generate singlet oxygen ($^1\text{O}_2$) involve materials connected to the terminal oxygen of PMS. The authors should compare the adsorption energies of the two distinct oxygen sites of PMS.

6. Comprehensive assessment of catalytic performance under controlled gaseous environments (e.g., O_2 , air, N_2) is essential to precisely delineate the role of dissolved oxygen, as it mediates both singlet oxygen ($^1\text{O}_2$) formation and potentially serves as a precursor for superoxide radical ($\text{O}_2^{\bullet-}$) generation via electron transfer mechanisms.

7. Comparisons with recent single-atom catalysts lack rigor due to inconsistent pollutant selection. Degradation efficiency varies significantly with molecular structures (e.g., bond types, substituent groups). To ensure valid benchmarking, use identical target pollutants for cross-study evaluations.

8. The authors ascribe significant importance to $^1\text{O}_2$ in the reaction mechanism, and this proposition appears to be corroborated by experimental evidence. However, given that $^1\text{O}_2$ is a less potent oxidizing agent compared to hydroxyl radicals and many organic compounds may exhibit limited susceptibility to its oxidative effects, what specific mechanistic advantages does the $^1\text{O}_2$ -dominated pathway confer within this system?

Reviewer #3

(Remarks to the Author)

The manuscript investigated the role of high-shell coordination defects in Fe-N5 single-atom catalysts (SACs) for peroxymonosulfate (PMS) activation in Fenton-like reactions. Many recent studies have already explored defect engineering in SACs for PMS activation, and the manuscript fails to articulate a distinct advance over prior work. While the study tried to correlate defect engineering with catalytic performance, the novelty and mechanistic insights remained unclear, and it does not convincingly demonstrate how this system was fundamentally different from existing Fe-N4 or other SACs. The following aspects require further improvement:

1. The authors focused heavily on the role of high-shell coordination defects and introduced the concept of "remote modulation" of Fe-N5 active sites. However, the manuscript did not provide a clear definition of what constituted "remote" in this context. It was important to clarify how the spatial range of remote modulation was determined. For example, what specific atomic or structural distance qualified a coordination interaction as remote rather than part of the primary or secondary coordination sphere? Additionally, the authors should have specified how this remote interaction was identified or quantified using experimental or theoretical methods. Without a measurable or defined parameter, the concept of "remote modulation" remained too vague, which undermined the mechanistic clarity of the study. A more explicit explanation and supporting data were needed to convincingly establish the structural basis and catalytic relevance of these high-shell coordination effects.

2. The authors compared the catalytic performance of various materials using normalized kobs while only accounting for differences in catalyst and PMS mass. However, the reliability and validity of the normalization method used were not clearly established. It remained unclear whether such a metric adequately reflected the intrinsic activity of the active sites. Given the single-atom characteristic of the catalysts and the potential variability in active site exposure or utilization efficiency, the choice of normalization method could have significantly influenced the interpretation of relative activity.

3. Regarding the mechanism of reactive species generation, the DFT calculations indicated that the highest energy barrier occurred in step 1 ($^* \text{PMS} \rightarrow ^* \text{OH} + ^* \text{SO}_4$), which was identified as the rate-determining step. This reaction pathway, involving the cleavage of the O-O bond in PMS to generate hydroxyl and sulfate radicals, was highly consistent with mechanisms widely reported in previous studies on Fenton-like catalysis (e.g. *Nat Commun* 16, 2402 (2025)). As such, it did not appear to offer fundamentally new mechanistic insights or demonstrate a significant difference from conventional PMS activation behavior. Moreover, Figures 4b and 4f showed that the FeN5-SD2 catalyst exhibited the highest PMS decomposition efficiency and the greatest yield of $^1\text{O}_2$ among the catalysts examined. However, the DFT results presented in the manuscript did not clearly support or explain this observation. In particular, there was a lack of theoretical evidence linking the structural modulation of FeN5-SD2 to enhanced $^1\text{O}_2$ generation, such as an energetically favorable non-radical pathway or lower activation barrier for $^1\text{O}_2$ evolution. This discrepancy between experimental findings and computational predictions raised questions about the proposed mechanism and its completeness.

4. The section concerning the degradation pathway of BPA and the associated toxicity analysis appeared to be relatively superficial and lacked depth in mechanistic insight. The current analysis did not convincingly emphasize the unique reactivity or selectivity resulting from the high shell coordination defects in the FeN5-SD2 catalyst. The proposed degradation pathways appeared to follow the general trends observed in previous studies and did not clearly show how the engineered coordination environment affected the formation or inhibition of specific intermediates. Furthermore, the identification of transformation products remained largely qualitative. The authors did not provide quantitative analyses of key intermediates, such as their concentrations or profiles over time. Without these quantitative data, subsequent toxicity assessments lost much of their persuasive power and did not accurately reflect the true environmental risks.

5. The overall writing style of the manuscript needed improvement. In particular, the frequent use of excessively long sentences with multiple subordinate clauses tended to obscure the original meaning and reduced the clarity of key scientific points. A more concise and direct sentence structure was recommended to improve readability and ensure effective communication of central ideas. In addition, some of the wording in the manuscript appeared imprecise or overly generalized. Given the complexity and technical depth of the study, authors were advised to carefully reconsider the use of terminology.

Version 1:

Reviewer comments:

Reviewer #1

(Remarks to the Author)

While the authors have clearly invested significant effort and conducted extensive work, as previously noted, the level of innovation does not meet the threshold required for publication in this journal.

Reviewer #2

(Remarks to the Author)

In the manuscript, the multiple coordination shells of FeN₅-SD_x single-atom catalysts (SACs) were precisely regulated for peroxymonosulfate (PMS) activation in Fenton-like reactions. A point-by-point response to each point raised has been made, the revisions now meet the reviewers' requirements. In my opinion, the manuscript can be considered for publication at current state.

Reviewer #3

(Remarks to the Author)

Accept

REVIEWER COMMENTS

Reviewer #1 (Remarks to the Author):

The work (NCOMMS-25-25049) tries to regulate the high-shell coordination defects of a model SAC with well-defined axial Fe-N₅ configurations to elucidate the impact of long-distance interactions on PMS activation. While the topic is conceptually interesting, it lacks clear innovation, particularly in the context of PMS activation for BPA removal, which has been extensively studied and offers limited novelty or practical potential in its current form.

1. The central value of this work lies in its attempt to correlate high-shell coordination structures of metal centres with catalytic performance. However, the design of the probe catalyst, the core of this study, is questionable. As shown in Fig. 1a and described in the experimental section, varying amounts of DCB were used during synthesis, leading to different metal loadings across the FeN₅-SD_x series. This results in inconsistent iron concentrations during PMS activation, making it difficult to isolate the effect of coordination defects from the influence of overall Fe content. Since the metal centre is presumed to be the principal active site, the authors must clarify how the contribution of high-shell coordination defects to BPA degradation is distinguished from variations in metal loading.

Response 1:

We appreciate the reviewer's insightful comment regarding the potential influence of varying Fe content on the observed catalytic performance. We agree that distinguishing the intrinsic contribution of high coordination shell defects from the influence of other variables (such as Fe loading and specific surface area of different catalysts) is critical to validating our structure-activity correlation. To address this concern, we have undertaken a comprehensive kinetic normalization analysis based on two independent metrics: the Fe-molar normalized rate constant ($K_{\text{per-mol Fe}}$) and the specific surface area-normalized rate constant ($K_{\text{per-area}}$).

Specifically, $K_{\text{per-mol Fe}}$ evaluates the catalytic efficiency per mole of Fe. The metal contents in different FeN₅-SD_x catalysts were quantified using an inductively coupled plasma emission spectrometer (ICP-OES), with the specific values presented in **Supplementary Table 3**. Among the FeN₅-SD_x catalysts, FeN₅-SD₂ exhibited the highest $K_{\text{per-mol Fe}}$ ($3.40 \times 10^4 \text{ min}^{-1} \cdot \text{M}^{-1}$), outperforming FeN₅ ($1.30 \times 10^4 \text{ min}^{-1} \cdot \text{M}^{-1}$), FeN₅-SD₁ ($1.75 \times 10^4 \text{ min}^{-1} \cdot \text{M}^{-1}$) and FeN₅-SD₃ ($2.96 \times 10^4 \text{ min}^{-1} \cdot \text{M}^{-1}$). These results clearly demonstrate that the superior catalytic activity of FeN₅-SD₂ cannot be attributed solely to Fe content, effectively ruling out Fe loading as the primary driver of

performance.

In parallel, we normalized the catalytic activity by the specific surface area $K_{\text{per-area}}$, determined *via* Brunauer-Emmett-Teller (BET) analysis (**Supplementary Table 4**). Again, FeN₅-SD₂ showed the highest surface-area-normalized activity (0.0072 min⁻¹/(m²·g⁻¹)), compared to FeN₅ (0.0046), FeN₅-SD₁ (0.0052), and FeN₅-SD₃ (0.0041). This further excludes the possibility that enhanced performance arises from improved surface dispersion or accessibility.

Collectively, these normalized results rule out the influence of other variables, including metal loading and specific surface area, on the observed catalytic enhancement. Therefore, the improved performance can be primarily ascribed to the electronic and geometric modulation induced by high coordination shell defects.

Action 1: We have added relevant statements in the revised manuscript (**Fig. 3b**) and **Supplementary Tables 3-4** have also been added in the Supporting Information.

“We performed kinetic normalization using both Fe-molar normalized rate constant ($K_{\text{per-mol Fe}}$) and specific surface area-normalized rate constant ($K_{\text{per-area}}$) to rule out other influencing factors (Fig. 3b). Inductively coupled plasma emission spectrometer (ICP-OES) data quantified metal contents across the FeN₅-SD_x catalysts (Supplementary Table 3). Among these catalysts, FeN₅-SD₂ exhibited the highest $K_{\text{per-mol Fe}}$ ($3.40 \times 10^4 \text{ min}^{-1} \cdot \text{M}^{-1}$), outperforming FeN₅ ($1.30 \times 10^4 \text{ min}^{-1} \cdot \text{M}^{-1}$), FeN₅-SD₁ ($1.75 \times 10^4 \text{ min}^{-1} \cdot \text{M}^{-1}$) and FeN₅-SD₃ ($2.96 \times 10^4 \text{ min}^{-1} \cdot \text{M}^{-1}$). Similarly, $K_{\text{per-area}}$ values confirmed the highest surface-normalized rate for FeN₅-SD₂ (0.0072 min⁻¹/(m²·g⁻¹)) (Supplementary Table 4). These metrics rule out Fe content and surface area as dominant performance factors. Thus, the improved performance primarily stems from electronic and geometric modulation induced by high coordination shell defects.”

Fig. 3 (b) Normalized rate constant of BPA degradation by per mole Fe, per surface area.

Supplementary Table 3. Comparison of molar iron-normalized rate constants ($K_{\text{per-mol Fe}}$) for FeN₅-Based PMS systems.

Catalysts	$m_{\text{cat.}}$ (g·L ⁻¹)	ω_{metal} (%)	k_{obs} (min ⁻¹)	$K_{\text{per-mol Fe}}$ ($\times 10^4 \text{ min}^{-1} \cdot \text{M}^{-1}$)
FeN ₅	0.03	2.88	0.200	1.30
FeN ₅ -SD ₁	0.03	2.59	0.243	1.75
FeN₅-SD₂	0.03	2.21	0.403	3.40
FeN ₅ -SD ₃	0.03	1.69	0.268	2.96

Supplementary Table 4. Comparison of area-normalized rate constants ($K_{\text{per-area}}$) for FeN₅-based PMS systems with varying defect densities.

Catalysts	BET (m ² /g)	k_{obs} (min ⁻¹)	$K_{\text{per-area}}$ (min ⁻¹ /(m ² ·g ⁻¹))
FeN ₅	43.7834	0.200	0.0046
FeN ₅ -SD ₁	46.8609	0.243	0.0052
FeN₅-SD₂	55.6287	0.403	0.0072
FeN ₅ -SD ₃	65.8330	0.268	0.0041

2. **(2.1)** Furthermore, while the manuscript repeatedly emphasizes the role of high-shell coordination in PMS activation and BPA degradation, the structural evidence supporting these claims is weak. The coordination defects in FeN₅-SD_x are not well characterized. **(2.2)** FTIR spectra show changes in other functional groups that are overlooked in the discussion, **(2.3)** and XPS results indicate the coexistence of two Fe valence states, which appears inconsistent with the proposed structure. **(2.4)** The stability of the proposed structures, particularly FeN₅-SD₃ as depicted in Fig. 1a, is also questionable and lacks adequate validation.

Response 2.1: Coordination defects and structural evidence

We sincerely thank the reviewer for pointing out the need for stronger structural evidence to support the presence and role of high coordination shell defects in the FeN₅-SD_x catalysts. In response, we have carefully re-examined and expanded our discussion on the spectroscopic data, including FTIR, Raman and BET, along with newly added NMR and XPS analyses to better demonstrate the structural evolution and stability of the phthalocyanine-based framework, as well as the controlled introduction of coordination defect.

In the FTIR spectra, all FeN₅-SD_x samples retain the characteristic absorption bands

of nitrogen-doped carbon nanotubes (NCNTs), while additional peaks corresponding to polyphthalocyanine vibrations are clearly observed (**Fig. 1e**). Specifically, the $\text{-C}\equiv\text{N}$ stretching vibration at 2240 cm^{-1} , characteristic of the BTC monomer, disappears progressively, accompanied by the emergence of $\text{C}=\text{N}$ (1500 cm^{-1} and 1135 cm^{-1}) and $\text{C}-\text{N}$ (1310 cm^{-1}) vibrations¹. These spectral changes confirm the successful polymerization of BTC into a conjugated polyphthalocyanine network *via* $\text{C}=\text{N}$ bond formation. As the defect concentration increases, the intensity of the polyphthalocyanine-related $\text{C}=\text{N}$ and $\text{C}-\text{N}$ bands decrease moderately, suggesting partial disruption of the polyphthalocyanine framework due to coordination defects. Meanwhile, the $\text{C}=\text{C}$ aromatic stretching band remains largely unchanged, indicating that the incorporation of DCB predominantly interferes with the formation of pyrrole ring *via* the loss of $\text{C}=\text{N}$ or $\text{C}-\text{N}$ units during phthalocyanine polymerization, while preserving the integrity of the benzene rings. It is noteworthy that several phthalocyanine vibrational bands in $\text{FeN}_5\text{-SD}_2$ and $\text{FeN}_5\text{-SD}_3$ are almost undetectable, which can be attributed to significant spectral overlap with the NCNTs background. To clarify this, we further analyzed the FTIR spectra of NCNTs-free samples (denoted as $\text{FeN}_4\text{-SD}_2$ and $\text{FeN}_4\text{-SD}_3$). As shown in **Fig. R1**, well-defined phthalocyanine vibrational modes are clearly retained in both $\text{FeN}_4\text{-SD}_2$ and $\text{FeN}_4\text{-SD}_3$, confirming that the structural framework of the polyphthalocyanine remains largely intact despite the presence of defects.

These trends are corroborated by Raman spectroscopy (**Fig. 1f**). In addition to the prominent D and G bands of NCNTs, the $\text{FeN}_5\text{-SD}_x$ catalysts exhibit distinct peaks corresponding to the A_{1g} , B_{1g} , and B_{2g} modes, which originate from $\text{C}-\text{C}$ stretching within the aromatic macrocycle and $\text{C}-\text{N}-\text{C}$ bridging vibrations—typical features of polyphthalocyanine². The gradual attenuation of these peaks with increasing defect concentration supports the interpretation that the defects induce a localized and controlled structural perturbation, rather than a complete collapse of the polyphthalocyanine network.

To further confirm the structural integrity and local coordination environment of $\text{FeN}_5\text{-SD}_x$, we employed solid-state ^{13}C NMR spectroscopy using NCNT-free $\text{FeN}_4\text{-SD}_x$ samples as models. As shown in **Supplementary Fig. 5**, a broad resonance signal at approximately 100-130 ppm is assigned to aromatic carbons in the benzene rings of the covalent framework, while a distinct signal at ~ 171 ppm corresponds to the $\text{C}=\text{N}$ groups in the pyrrole ring of phthalocyanine unit^{1, 2, 3}. The presence of this 171 ppm signal provides direct evidence for the successful formation of the conjugated

phthalocyanine framework. Importantly, as the defect concentration increases from FeN₄ to FeN₄-SD₃, the 171 ppm signal exhibits a systematic decrease in intensity, indicative of a gradual reduction in pyrrolic C=N content. This trend strongly suggests the formation of high coordination shell defects *via* selective disruption of the pyrrole units within the phthalocyanine ring. In contrast, the resonance signals corresponding to the benzene ring carbons remain largely unaffected, confirming that the structural perturbations are localized and do not compromise the integrity of the aromatic backbone.

Complementary XPS analysis corroborates these findings. The C 1s spectra (**Supplementary Fig. 6**) show a clear decrease in the relative proportion of C=N and C-N components with increasing defect concentration. In addition, quantitative analysis of N content derived from XPS (**Table R1**) reveals a consistent downward trend, further supporting the partial loss of coordinated nitrogen due to defect incorporation. Brunauer–Emmett–Teller (BET) analysis (**Supplementary Table 1**) further confirms a gradual increase in surface area and pore volume with defect introduction, while maintaining the overall covalent framework.

In summary, the combined spectroscopic data provide robust evidence for the presence of high coordination shell defects in the FeN₅-SD_x catalysts. These defects arise from a targeted and controllable perturbation of the phthalocyanine coordination structure, rather than random distributed.

Fig. 1 (e) FTIR spectra of BTC, FeN₅, FeN₅-SD₁, FeN₅-SD₂, FeN₅-SD₃ and NCNTs.

Fig. R1 FTIR spectra of BTC, FeN₄-SD₂, and FeN₄-SD₃.

Fig. 1 (f) Raman spectra of BTC, FeN₅, FeN₅-SD₁, FeN₅-SD₂, FeN₅-SD₃ and NCNTs.

Supplementary Fig. 5 Solid-state ¹³C NMR spectra and ChemDraw 20.0 “predicted” of FeN₄, FeN₄-SD₁, FeN₄-SD₂ and FeN₄-SD₃.

Supplementary Fig. 6 XPS C1s spectra of of FeN₄, FeN₄-SD₁, FeN₄-SD₂ and FeN₄-SD₃.

Table R1. XPS analysis of N content in FeN₅, FeN₅-SD₁, FeN₅-SD₂ and FeN₅-SD₃.

Catalysts	N 1s (wt%)
FeN ₅	7.94
FeN ₅ -SD ₁	6.71
FeN ₅ -SD ₂	5.66
FeN ₅ -SD ₃	4.87

Supplementary Table 1. BET surface area and total pore volume a of FeN₅, FeN₅-SD₁, FeN₅-SD₂ and FeN₅-SD₃

Catalysts	BET specific surface area (m ² g ⁻¹)	Total pore volume (cm ³ g ⁻¹)
FeN ₅	43.78	0.1913
FeN ₅ -SD ₁	46.86	0.3856
FeN ₅ -SD ₂	55.63	0.4295
FeN ₅ -SD ₃	65.83	0.4335

Action 2.1: We have the added relevant statements in the revised manuscript (Fig. 1e-f). Supplementary Table 1 and Supplementary Figs. 5-6 have also been included in the supporting information.

“Fourier transform infrared (FTIR) spectra of all FeN₅-SD_x samples exhibit the

characteristic absorption bands of NCNTs, along with additional peaks assigned to polyphthalocyanine vibrations (Fig. 1e). The progressive disappearance of the $\text{-C}\equiv\text{N}$ stretching vibration at 2240 cm^{-1} , characteristic of the BTC monomer, is accompanied by the emergence of new peaks corresponding to $\text{C}=\text{N}$ (1500 and 1135 cm^{-1}) and $\text{C}-\text{N}$ (1310 cm^{-1}) stretching modes. These changes indicate successful polymerization into a polyphthalocyanine-like framework. A strong absorption at 1396 cm^{-1} is attributed to $\text{C}=\text{C}$ stretching within the conjugated macrocycle, while a peak at 731 cm^{-1} arises from $\text{C}-\text{H}$ vibrations, further confirming the formation of extended polyphthalocyanine structure¹. With increasing defect concentration, the intensity of the $\text{C}=\text{N}$ and $\text{C}-\text{N}$ bands moderately decreases, suggesting partial disruption of the pyrrole units within the phthalocyanine ring. In contrast, the $\text{C}=\text{C}$ band remains stable, implying that the benzene moieties are preserved. These observations are supported by Raman spectra (Fig. 1f), where in addition to the D and G bands of NCNTs, distinct A_{1g} , B_{1g} , and B_{2g} modes related to $\text{C}-\text{C}$ and $\text{C}-\text{N}-\text{C}$ vibrations in the phthalocyanine ring are observed.². The gradual attenuation of these peaks with implies localized structural perturbations rather than a complete collapse of the polyphthalocyanine network. To further examine the coordination environment, solid-state ^{13}C nuclear magnetic resonance (NMR) spectroscopy was conducted using NCNT-free $\text{FeN}_4\text{-SD}_x$ samples. As shown in Supplementary Fig. 5, a broad resonance at $100\text{-}130\text{ ppm}$ is assigned to aromatic carbons in benzene rings, while a distinct signal at $\sim 171\text{ ppm}$ corresponds to $\text{C}=\text{N}$ groups in the pyrrole ring of phthalocyanine unit^{1, 2, 3}. The progressive decrease in the 171 ppm signal from FeN_4 to $\text{FeN}_4\text{-SD}_3$ reflects a gradual loss of pyrrolic $\text{C}=\text{N}$ groups, consistent with the formation of high coordination shell defects. Meanwhile, the signals from benzene carbons remain largely unchanged, confirming that structural disruption is confined to the pyrrole moieties. X-ray photoelectron spectroscopy (XPS) $\text{C } 1s$ spectra (Supplementary Fig. 6) show a decrease in $\text{C}=\text{N}$ and $\text{C}-\text{N}$ components with increasing defect concentration, supporting the partial loss of coordinated nitrogen. Brunauer–Emmett–Teller (BET) analysis (Supplementary Table 1) further confirms a progressive increase in surface area and pore volume with defect introduction, while preserving the overall covalent framework.”

Fig. 1 (e) FTIR spectra of BTC, FeN₅, FeN₅-SD₁, FeN₅-SD₂, FeN₅-SD₃ and NCNTs.

Fig. 1 (f) Raman spectra of BTC, FeN₅, FeN₅-SD₁, FeN₅-SD₂, FeN₅-SD₃ and NCNTs.

Supplementary Fig. 5 Solid-state ¹³C NMR spectra and ChemDraw 20.0 “predicted” of FeN₄, FeN₄-SD₁, FeN₄-SD₂ and FeN₄-SD₃.

Supplementary Fig. 6 XPS C1s spectra of of FeN₄, FeN₄-SD₁, FeN₄-SD₂ and FeN₄-SD₃.

Supplementary Table 1. BET surface area and total pore volume a of FeN₅, FeN₅-SD₁, FeN₅-SD₂ and FeN₅-SD₃

Catalysts	BET specific surface area (m ² g ⁻¹)	Total pore volume (cm ³ g ⁻¹)
FeN ₅	43.78	0.1913
FeN ₅ -SD ₁	46.86	0.3856
FeN ₅ -SD ₂	55.63	0.4295
FeN ₅ -SD ₃	65.83	0.4335

Response 2.2: FTIR spectra and functional group changes

We appreciate the reviewer pointing out the overlooked changes in functional groups in the FTIR spectra.

Action 2.2: In the revised manuscript, we now include a more comprehensive discussion of the observed changes:

“Fourier transform infrared (FTIR) spectra of all FeN₅-SD_x samples exhibit the characteristic absorption bands of NCNTs, along with additional peaks assigned to polyphthalocyanine vibrations (Fig. 1e). The progressive disappearance of the -C≡N stretching vibration at 2240 cm⁻¹, characteristic of the BTC monomer, is accompanied by the emergence of new peaks corresponding to C=N (1500 and 1135 cm⁻¹) and C-N (1310 cm⁻¹) stretching modes. These changes indicate successful polymerization into a

polyphthalocyanine-like framework. A strong absorption at 1396 cm^{-1} is attributed to $\text{C}=\text{C}$ stretching within the conjugated macrocycle, while a peak at 731 cm^{-1} arises from $\text{C}-\text{H}$ vibrations, further confirming the formation of extended polyphthalocyanine structure¹. With increasing defect concentration, the intensity of the $\text{C}=\text{N}$ and $\text{C}-\text{N}$ bands moderately decreases, suggesting partial disruption of the pyrrole units within the phthalocyanine ring. In contrast, the $\text{C}=\text{C}$ band remains stable, implying that the benzene moieties are preserved.”

Fig. 1 (e) FTIR spectra of BTC, FeN_5 , $\text{FeN}_5\text{-SD}_1$, $\text{FeN}_5\text{-SD}_2$, $\text{FeN}_5\text{-SD}_3$ and NCNTs.

Response 2.3: XPS results and Fe valence states

We appreciate the reviewer’s observation regarding the coexistence of two Fe valence states (Fe^{2+} and Fe^{3+}) and the concern about its apparent inconsistency with the proposed structure. We would like to clarify that the presence of mixed Fe valence states is a commonly observed phenomenon in Fe-based SACs and does not imply a structural inconsistency. Instead, this observation reflects a dynamic redox equilibrium arising from asymmetric coordination environments. Such structural asymmetry can locally modulate the electronic environment of the Fe center, thereby stabilizing multiple oxidation states. In fact, this redox flexibility is a characteristic feature of SACs and is considered advantageous for catalytic performance, as it facilitates electron transfer and enhances PMS activation.

Importantly, the coexistence of Fe^{2+} and Fe^{3+} states observed by XPS is consistent with our Fe K-edge X-ray absorption fine-structure (XAFS) results. As shown in the XANES spectra (**Fig. 2b**), the absorption edges of FeN_5 and $\text{FeN}_5\text{-SD}_2$ lie between those of metallic Fe and Fe_2O_3 references, confirming an average oxidation state between 0 and +3. Moreover, the distinct rightward shift of the $\text{FeN}_5\text{-SD}_2$ absorption

edge relative to FeN₅ indicates a higher Fe valence, further validating the effect of coordination defect engineering on electronic structure modulation.

The coexistence of multiple metal valence states, therefore, not only aligns with the proposed structural model but also supports the functional role of redox-active metal centers in catalytic oxidation. Similar mixed-valence features have been extensively documented in Fe-N-C systems and are often associated with enhanced catalytic reactivity. (*Angew. Chem. Int. Ed.* 2024, 63, e202318246. (**Fig. R2**), *Proc. Natl. Acad. Sci. U.S.A.* 120 (15) e2300281120 (**Fig. R3**). and *Adv. Funct. Mater.* 2021, 31, 2103857. (**Fig. R4**)).^{4, 5, 6, 7, 8}

Fig. 2 (a) XPS Fe 2p spectra of the as-prepared catalysts. (b) Fe k-edge XANES spectra, of Fe foil, Fe₂O₃, FePc, FeN₅ and FeN₅-SD₂.

[Figure Redacted]

Fig. R2 (a) High-resolution Fe 2p spectrum of FeN₄, FeS₁N₃, and FeB₁N₃. (b) The normalized Fe K-edge XANES spectra of Fe Foil, Fe₂O₃, FePc, FeN₄, FeS₁N₃ and FeB₁N₃. The inset shows the enlarged spectra.⁴ (*Angew. Chem. Int. Ed.* 2024, 63, e202318246)

[Figure Redacted]

Fig. R3 (a) Fe 2p XPS, (b) EXAFS analysis of TiFeSA, TiFeAS, and TiFeNP.⁵ (*Proc. Natl. Acad. Sci. U.S.A.* 120 (15) e2300281120)

[Figure Redacted]

Fig. R4 (a) High-resolution XPS Fe 2p spectra of SA-Fe-HPC and control sample. (b) The Fe K-edge XANES spectra of Fe–N/P–C-700 and control samples. (c) Linear fitting curve of Fe K-edge energy and Fe valence for Fe–N/P–C-700 and reference samples.⁶ (*Adv. Funct. Mater.* 2021, 31, 2103857)

Response 2.4: Stability of FeN₅-SD_x structure

We sincerely thank the reviewer for raising the important concern regarding the structural stability of the proposed FeN₅-SD_x catalysts, particularly the FeN₅-SD₃ model presented in **Fig. 1a**. We would like to clarify that **Fig. 1a** presents only a local structural

unit to highlight the coordination environment, which may have inadvertently suggested a risk of structural collapse. However, from a broader perspective, the FeN₅-SD_x catalysts are constructed as extended covalently bonded polymeric frameworks. While the introduction of high coordination shell defects leads to increased porosity, the overall network remains robust due to the intrinsic covalent polymerization connectivity.

Theoretical validation *via* DFT calculations

To further substantiate the structural stability at the atomic level, we conducted full geometry optimizations for FeN₅, FeN₅-SD₂, and FeN₅-SD₃ models using density functional theory (DFT). The optimized structures, now provided in **Fig. R5-R7**, reveal that all configurations converge to thermodynamically stable geometries without distortion or collapse. Crucially, the preservation of key coordination motifs across these models confirms that the introduction of coordination defects does not induce Fe aggregation or destabilization, even at higher defect densities. These results provide strong theoretical validation for the structural integrity of the proposed FeN₅-SD_x framework

Non-pyrolytic synthesis enables defect control and structural retention

We present a pyrolysis-free, low-temperature polymerization strategy that enables precise incorporation of structural defects at predetermined sites through rational functionalization of organic monomers (**Fig. 1a and Supplementary Fig. 1**). This approach affords controlled modulation of the coordination environment surrounding the FeN₅ active site. Such precise control is difficult to achieve with traditional high-temperature pyrolysis methods, where active site formation is inherently stochastic. The polyphthalocyanine framework inherently facilitates transition metal incorporation within its macrocyclic cavity. Concurrently, nitrogen-rich NCNTs promote axial Fe–N coordination, collectively stabilizing highly ordered FeN₅ centers as the dominant catalytic motifs. To introduce coordination defects, a controlled amount of 1,2-dicyanobenzene (DCB) was co-polymerized with benzene-1,2,4,5-tetracarbonitrile (BTC). The incorporation of DCB systematically perturbs the polymeric framework, generating high coordination shell defects localized around the metal centers (denoted as FeN₅-SD_x).

The structural stability of the phthalocyanine framework across the FeN₅-SD_x series is supported by a range of complementary spectroscopic techniques. FTIR and Raman analyses reveal consistent vibrational features attributable to C=N, C-N, and aromatic C-C bonds characteristic of the phthalocyanine scaffold (**Fig. 1e-f**). XPS C 1s spectra

exhibit distinct peaks for C-C/C=C, C-N, and C=N bonds (**Supplementary Fig. 6**), with a noticeable decrease in the relative proportions of C=N and C-N components as the defect concentration increases. Furthermore, solid-state ^{13}C NMR spectra display a distinct resonance at ~ 171 ppm assigned to C=N in the pyrrolic ring, as well as broad peaks at ~ 100 - 130 ppm associated with aromatic carbons in the benzene moieties of the covalent framework (**Supplementary Fig. 5**). While these signals show moderate attenuation at higher defect concentrations (e.g., $\text{FeN}_5\text{-SD}_3$), their persistence affirms the retention of the core conjugated framework despite defect introduction. Brunauer–Emmett–Teller (BET) analysis (**Supplementary Table 1**) further confirms a gradual increase in surface area and pore volume with defect introduction, while maintaining the overall covalent framework.

Direct imaging of atomic dispersion at high defect density

To provide direct evidence of atomic dispersion at the highest defect level ($\text{FeN}_5\text{-SD}_3$), aberration-corrected high-angle annular dark-field scanning transmission electron microscopy (AC-HAADF-STEM) coupled with elemental mapping was performed. As shown in **Fig. R8**, bright isolated spots corresponding to single Fe atoms are clearly visible, with no signs of clustering or nanoparticle formation. These findings unequivocally confirm that single-atom dispersion is maintained even in highly defective coordination environments.

Post-reaction structural robustness

To further assess the stability of the $\text{FeN}_5\text{-SD}_x$ catalysts under catalytic conditions, we performed post-reaction characterizations including XRD and XPS. The XRD results indicate no significant changes in the bonding features, crystalline structure after catalytic cycling (**Supplementary Fig. 10**). XPS analysis of the C1s spectra was also performed to assess carbon oxidation, with a particular focus on the C-O peak (**Supplementary Fig. 11**). The results showed no significant increase in C-O peak intensity after the reaction, indicating the catalyst's strong resistance to oxidation and stability in Fenton-like reactions. These findings confirm that the catalysts maintain their structural integrity throughout the reaction process.

Collectively, our DFT calculations, synthetic methodology, multi-modal spectroscopic validation, atomic-resolution imaging, and post-reaction analyses provide strong and consistent evidence that the proposed $\text{FeN}_5\text{-SD}_x$ structures, particularly the high-defect $\text{FeN}_5\text{-SD}_3$ configuration, are both thermodynamically stable and structurally robust. We believe these new results address the reviewer's concerns and strongly support the validity of our structural model.

Editorial Note: The crystal structures in Fig. R5 were visualized using VESTA software (Momma, K. & Izumi, F. VESTA 3 for three-dimensional visualization of crystal, volumetric and morphology data, J. Appl. Crystallogr. 44, 1272–1276 (2011).)

[Figure Redacted]

Fig. 1 (a) Schematic of the preparation strategy for FeN₅-SD_x.

Supplementary Fig. 1 Formation procedure of FeN₅-SD_x.

Fig. R5 The optimal configurations of FeN₅.

Editorial Note: The crystal structures in Fig. R6 and R7 were visualized using VESTA software (Momma, K. & Izumi, F. VESTA 3 for three-dimensional visualization of crystal, volumetric and morphology data, J. Appl. Crystallogr. 44, 1272–1276 (2011).)

Fig. R6 The optimal configurations of FeN₅-SD₂.

Fig. R7 The optimal configurations of FeN₅-SD₃.

Fig. 1 (e) FTIR spectra of BTC, FeN₅, FeN₅-SD₁, FeN₅-SD₂, FeN₅-SD₃ and NCNTs.

Fig. 1 (f) Raman spectra of BTC, FeN₅, FeN₅-SD₁, FeN₅-SD₂, FeN₅-SD₃ and NCNTs.

Supplementary Fig. 5 Solid-state ¹³C NMR spectra and ChemDraw 20.0 “predicted” of FeN₄, FeN₄-SD₁, FeN₄-SD₂ and FeN₄-SD₃.

Supplementary Fig. 6 XPS C1s spectra of of FeN₄, FeN₄-SD₁, FeN₄-SD₂ and FeN₄-SD₃.

Supplementary Table 1. BET surface area and total pore volume a of FeN₅, FeN₅-SD₁, FeN₅-SD₂ and FeN₅-SD₃

Catalysts	BET specific surface area (m ² g ⁻¹)	Total pore volume (cm ³ g ⁻¹)
FeN ₅	43.78	0.1913
FeN ₅ -SD ₁	46.86	0.3856
FeN ₅ -SD ₂	55.63	0.4295
FeN ₅ -SD ₃	65.83	0.4335

Fig. R8 (a-d) EDS images of FeN₅-SD₃.

Supplementary Fig. 10 XRD spectra of (a) FeN₅, (b) FeN₅-SD₁, (c) FeN₅-SD₁, and (d) FeN₅-SD₁ before and after use.

Supplementary Fig. 11 XPS C 1s spectra of (a) FeN₅, (b) FeN₅-SD₁, (c) FeN₅-SD₁, and (d) FeN₅-SD₁ before and after use.

Action 2.4: We have added related statements to validate the structural stability of $\text{FeN}_5\text{-SD}_x$ in revised manuscript (Fig. 1a, Fig. 1e-f). Supplementary Table 1, Supplementary Figs. 5-6 and 10-11 have also been added in the Supporting Information.

“This synthesis strategy fundamentally differs from conventional M-N-C catalyst preparation by eliminating high-temperature pyrolysis while maintaining well-defined active site structure (Fig. 1a). The polyphthalocyanine framework inherently facilitates transition metal incorporation within its macrocyclic cavity. Simultaneously, nitrogen-rich NCNTs promote axial Fe–N coordination. Together, these features stabilize highly ordered FeN_5 centers as the dominant catalytic sites.”

“To engineer the higher coordination shells around Fe centers, which typically include atoms beyond the primary and secondary shells, we introduced coordination defects (Fig. 1b). This was achieved by co-polymerizing a controlled amount of 1,2-dicyanobenzene (DCB) with benzene-1,2,4,5-tetracarbonitrile (BTC). Incorporating DCB systematically disrupts the polymer matrix, creating tailored defects in the higher coordination shells. These defects lie beyond 5 Å from the Fe center, as confirmed by DFT calculations (Supplementary Figs. 1-2), and are denoted as $\text{FeN}_5\text{-SD}_x$.”

[Figure Redacted]

Fig. 1 (a) Schematic of the preparation strategy for $\text{FeN}_5\text{-SD}_x$.

Supplementary Fig. 1 Formation procedure of $\text{FeN}_5\text{-SD}_x$.

“Fourier transform infrared (FTIR) spectra of all $\text{FeN}_5\text{-SD}_x$ samples exhibit the characteristic absorption bands of NCNTs, along with additional peaks assigned to polyphthalocyanine vibrations (Fig. 1e). The progressive disappearance of the $\text{-C}\equiv\text{N}$ stretching vibration at 2240 cm^{-1} , characteristic of the BTC monomer, is accompanied by the emergence of new peaks corresponding to $\text{C}=\text{N}$ (1500 and 1135 cm^{-1}) and $\text{C}-\text{N}$ (1310 cm^{-1}) stretching modes. These changes indicate successful polymerization into a polyphthalocyanine-like framework. A strong absorption at 1396 cm^{-1} is attributed to $\text{C}=\text{C}$ stretching within the conjugated macrocycle, while a peak at 731 cm^{-1} arises from $\text{C}-\text{H}$ vibrations, further confirming the formation of extended polyphthalocyanine structure¹. With increasing defect concentration, the intensity of the $\text{C}=\text{N}$ and $\text{C}-\text{N}$ bands moderately decreases, suggesting partial disruption of the pyrrole units within the phthalocyanine ring. In contrast, the $\text{C}=\text{C}$ band remains stable, implying that the benzene moieties are preserved. These observations are supported by Raman spectra (Fig. 1f), where in addition to the D and G bands of NCNTs, distinct A_{1g} , B_{1g} , and B_{2g} modes related to $\text{C}-\text{C}$ and $\text{C}-\text{N}-\text{C}$ vibrations in the phthalocyanine ring are observed.² The gradual attenuation of these peaks with implies localized structural perturbations rather than a complete collapse of the polyphthalocyanine network. To further examine the coordination environment, solid-state ^{13}C nuclear magnetic resonance (NMR) spectroscopy was conducted using NCNT-free $\text{FeN}_4\text{-SD}_x$ samples. As shown in Supplementary Fig. 5, a broad resonance at $100\text{-}130\text{ ppm}$ is assigned to aromatic carbons in benzene rings, while a distinct signal at $\sim 171\text{ ppm}$ corresponds to $\text{C}=\text{N}$ groups in the pyrrole ring of phthalocyanine unit^{1, 2, 3}. The progressive decrease in the 171 ppm signal from FeN_4 to $\text{FeN}_4\text{-SD}_3$ reflects a gradual loss of pyrrolic $\text{C}=\text{N}$ groups, consistent with the formation of high coordination shell defects. Meanwhile, the signals from benzene carbons remain largely unchanged, confirming that structural disruption

is confined to the pyrrole moieties. X-ray photoelectron spectroscopy (XPS) C 1s spectra (Supplementary Fig. 6) show a decrease in C=N and C-N components with increasing defect concentration, supporting the partial loss of coordinated nitrogen. Brunauer–Emmett–Teller (BET) analysis (Supplementary Table 1) further confirms a progressive increase in surface area and pore volume with defect introduction, while preserving the overall covalent framework.”

Fig. 1 (e) FTIR spectra of BTC, FeN₅, FeN₅-SD₁, FeN₅-SD₂, FeN₅-SD₃ and NCNTs.

Fig. 1 (f) Raman spectra of BTC, FeN₅, FeN₅-SD₁, FeN₅-SD₂, FeN₅-SD₃ and NCNTs.

Supplementary Fig. 5 Solid-state ^{13}C NMR spectra and ChemDraw 20.0 “predicted” of FeN_4 , $\text{FeN}_4\text{-SD}_1$, $\text{FeN}_4\text{-SD}_2$ and $\text{FeN}_4\text{-SD}_3$.

Supplementary Fig. 6 XPS $\text{C}1\text{s}$ spectra of FeN_4 , $\text{FeN}_4\text{-SD}_1$, $\text{FeN}_4\text{-SD}_2$ and $\text{FeN}_4\text{-SD}_3$.

Supplementary Table 1. BET surface area and total pore volume of FeN_5 , $\text{FeN}_5\text{-SD}_1$, $\text{FeN}_5\text{-SD}_2$ and $\text{FeN}_5\text{-SD}_3$

Catalysts	BET specific surface area ($\text{m}^2 \text{g}^{-1}$)	Total pore volume ($\text{cm}^3 \text{g}^{-1}$)
FeN_5	43.78	0.1913
$\text{FeN}_5\text{-SD}_1$	46.86	0.3856
$\text{FeN}_5\text{-SD}_2$	55.63	0.4295

Catalysts	BET specific surface area ($\text{m}^2 \text{g}^{-1}$)	Total pore volume ($\text{cm}^3 \text{g}^{-1}$)
$\text{FeN}_5\text{-SD}_3$	65.83	0.4335

“To further evaluate catalyst durability, post-reaction XRD and XPS analyses were conducted. XRD patterns revealed no significant structural changes after cycling (Supplementary Fig. 10), while XPS C 1s spectra showed no notable increase in the C–O peak, indicating strong resistance to carbon oxidation (Supplementary Fig. 11). These results confirm that the $\text{FeN}_5\text{-SD}_x$ catalysts retain their structural integrity under reaction conditions.”

Supplementary Fig. 10 XRD spectra of (a) FeN_5 , (b) $\text{FeN}_5\text{-SD}_1$, (c) $\text{FeN}_5\text{-SD}_1$, and (d) $\text{FeN}_5\text{-SD}_1$ before and after use.

Supplementary Fig. 11 XPS C 1s spectra of (a) FeN₅, (b) FeN₅-SD₁, (c) FeN₅-SD₁, and (d) FeN₅-SD₁ before and after use.

3. In addition, several typos and formatting issues detract from the manuscript's clarity, for example, the ball colour for DCB appears to be mislabelled (carbon?), Fig. 1e incorrectly uses 'wavelength' instead of 'wavenumber', and abbreviations such as DCB and BTC should be introduced in full upon first mention. Overall, the conclusions presented are not sufficiently supported by the data and analysis in the current form.

Response 3:

We sincerely thank the reviewer for identifying several typographical and formatting issues that may have hindered the clarity of our manuscript. In Fig. 1a, both the orange and green spheres represent carbon atoms. The green-colored carbon atoms in DCB were intentionally distinguished from the orange carbon atoms in BTC to highlight the precise doping of defect sites within the framework. This visual differentiation is meant solely to emphasize structural modulation and does not indicate a difference in elemental identity.

Action 3: To enhance clarity, we have revised the figure legend to explicitly state this distinction and avoid potential misinterpretation.

In Fig. 1e, the axis label has been revised from wavelength to wavenumber (cm⁻¹) to correctly reflect the FTIR spectral convention.

As suggested, we have revised the manuscript to spell out all abbreviations at first

mention for clarity and consistency.

Specifically, “1,2-dicyanobenzene (DCB)”, “benzene-1,2,4,5-tetracarbonitrile (BTC)”, and “inductively coupled plasma mass spectrometry (ICP-MS)” are now introduced in full when they first appear in the main text.

[Figure Redacted]

Fig. 1 (a) Schematic of the preparation strategy for FeN₅-SD_x.

Fig. 1 (e) FTIR spectra of BTC, FeN₅, FeN₅-SD₁, FeN₅-SD₂, FeN₅-SD₃ and NCNTs.

We understand the concern regarding the support for the conclusions presented in the manuscript. In response, we have revisited the data and conducted additional analyses to strengthen the claims. We have included supplementary results, including additional characterization and stability tests, to provide more robust evidence for the conclusions drawn in the manuscript. Furthermore, we have updated the discussion section to better align the conclusions with the experimental data, providing a clearer connection between the findings and the proposed mechanism.

We hope these revisions will adequately address your concerns and enhance the clarity and impact of our work.

Reviewer #2 (Remarks to the Author):

This manuscript presents an interesting work in single-atom catalysts (SACs) for Fenton-like water decontamination. The work elucidates the critical role of long-distance coordination defects in modulating the electronic structure and catalytic behavior of Fe-N₅ active sites. While numerous SACs have been reported for enhancing Fenton-like reactions, this study distinguishes itself by achieving atomic-level engineering of ordered active sites coupled with rigorous establishment of structure-activity relationships through systematic methodologies. Notably, the authors develop a framework for understanding the remote effects of multi-shell coordination on reaction dynamics. The experimental design is rigorous and well-executed, addressing a timely and compelling research frontier. I think that this work could be accepted after minor revision, which might improve the quality of this work:

1. A detailed synthetic scheme is required to improve the understanding of these materials. Specifically, how are the phthalocyanine frameworks or defect-engineered phthalocyanine-like frameworks synthesized? What are the synthesis processes and underlying mechanisms involved?

Response 1:

We sincerely thank the reviewer for the insightful suggestion. We agree that a detailed synthetic scheme is essential for improving the clarity of our work. Accordingly, we have revised the manuscript to include a comprehensive synthetic route that elucidates the formation of the defect-engineered phthalocyanine-like frameworks.

The defect-engineered phthalocyanine-like frameworks were synthesized *via* a non-pyrolytic, solvent-assisted polymerization strategy, employing metal (Fe) precursors, 1,2-dicyanobenzene (DCB) and benzene-1,2,4,5-tetracarbonitrile (BTC) under mild reaction conditions. This low-temperature approach preserves the structural integrity and ensures the formation of a highly conjugated, crystalline polyphthalocyanine network without the undesired aggregation or structural disorder commonly associated with high-temperature pyrolysis.

To engineer defects within the framework, we introduced a controlled amount of DCB as a defect-modulating co-monomer. The incorporation of DCB partially perturbs the polymeric structure, thereby introducing high coordination shell defects localized around the metal centers. These defects modulate the electronic environment and

coordination geometry of the Fe sites, effectively tailoring the catalytic properties of the resulting material.

Mechanisms Involved:

The polymerization mechanism is initiated by the nucleophilic attack of ethylene glycol-derived alkoxide anions on the cyano groups of BTC or DCB, forming reactive intermediates. The presence of 1,8-diazabicyclo[5.4.0]undec-7-ene (DBU) serves as a mild organic base to facilitate the condensation reaction. Coordination and cyclization of Fe³⁺ with these intermediates drive the formation of polyphthalocyanine macrocycles. In the DCB-assisted synthesis, the presence of DCB partially disrupts the conjugation and cyclization process, thereby introducing localized coordination defects within the framework.

Action 1: We have added a synthetic scheme and relevant statements in the revised manuscript (Fig. 1a-b). and Supplementary Figs. 1-2 has been added in the supporting information.

“This synthesis strategy fundamentally differs from conventional M-N-C catalyst preparation by eliminating high-temperature pyrolysis while maintaining well-defined active site structure (Fig. 1a). The polyphthalocyanine framework inherently facilitates transition metal incorporation within its macrocyclic cavity. Simultaneously, nitrogen-rich NCNTs promote axial Fe–N coordination. Together, these features stabilize highly ordered FeN₅ centers as the dominant catalytic sites. To engineer the higher coordination shells around Fe centers, which typically include atoms beyond the primary and secondary shells, we introduced coordination defects (Fig. 1b). This was achieved by co-polymerizing a controlled amount of 1,2-dicyanobenzene (DCB) with benzene-1,2,4,5-tetracarbonitrile (BTC). Incorporating DCB systematically disrupts the polymer matrix, creating tailored defects in the higher coordination shells. These defects lie beyond 5 Å from the Fe center, as confirmed by DFT calculations (Supplementary Figs. 1-2), and are denoted as FeN₅-SD_x.”

[Figure Redacted]

Fig. 1 (a) Schematic of the preparation strategy for FeN₅-SD_x.

[Figure Redacted]

Fig. 1 (b) Schematic illustration of the coordination structure of FeN₅-SD_x.

Supplementary Fig. 1 Formation procedure of FeN₅-SD_x.

Editorial Note: The crystal structures in Supplementary Fig. 2 were visualized using VESTA software (Momma, K. & Izumi, F. VESTA 3 for three-dimensional visualization of crystal, volumetric and morphology data, J. Appl. Crystallogr. 44, 1272–1276 (2011).)

Supplementary Fig. 2 The spatial range of remote modulation in catalyst structures.

2. If annealing treatment is applied to the catalyst, does it enhance the catalytic activity or stability of the axial Fe-N₅ sites? Does the pyrolysis-free synthetic strategy employed in your work effectively address the inherent carbon oxidation instability of conventional Fe-N-C SACs derived from thermal treatments? Please provide experimental evidence to verify the oxidation resistance of your conjugated polyphthalocyanine framework under operational conditions.

Response 2:

We appreciate the reviewer's insightful comments regarding the impact of annealing treatment and the oxidation resistance of our pyrolysis-free synthesized catalyst. We have conducted additional performance tests comparing our material with those synthesized through high-temperature annealing (referred to as Fe-N-C-HT).

The results, presented in **Fig. R9a**, demonstrate that our material significantly outperforms Fe-N-C-HT in both performance and stability. Specifically, the kinetic constant (k_{obs}) for our material remains high at 0.403 min⁻¹, whereas that for the Fe-N-C-HT drops to 0.042 min⁻¹ (**Fig. R9b**). Furthermore, Fe-N-C-HT exhibits poor cycling stability, with a dramatic decrease in degradation efficiency to <5% after only the second cycle. In contrast, our material maintains excellent degradation efficiency even after four cycles (**Fig. R9c-d**).

High-temperature pyrolysis has inherent drawbacks, such as the tendency for metal atoms to aggregate into particles or clusters, leading to reduced metal content and diminished effectiveness of single-atom dispersion. Additionally, it transforms the organized organic carbon matrix into disordered inorganic carbon, elevating the risk of

carbon oxidation. Furthermore, the active sites generated through this process are often random, which complicates the study and regulation of these sites.

As shown in the FTIR spectra (**Fig. R10a**), Fe-N-C-HT lacks the characteristic vibrational features associated with the phthalocyanine framework, which are clearly retained in our pyrolysis-free FeN₅-SD₂ catalyst. Moreover, the FTIR profile of Fe-N-C-HT closely resembles that of NCNTs, indicating that the phthalocyanine backbone was destroyed during pyrolysis, thereby compromising the ability to stabilize atomically dispersed Fe centers. XPS analysis (**Fig. R10b**) further supports this conclusion, revealing a marked decrease in Fe content in Fe-N-C-HT relative to FeN₅-SD₂, suggesting significant loss of Fe during pyrolysis. This reduction in Fe loading translates directly to a lower density of active sites, resulting in a substantial decline in catalytic degradation performance.

In contrast, our material benefits from a pyrolysis-free synthesis approach, resulting in a conjugated framework with periodic structural characteristics. This unique structure, which contains ordered organic carbon, effectively reduces carbon oxidation and enhances the cycling stability of our material. The absence of high-temperature processing also avoids the issues related to the randomness of active sites. From an energy efficiency perspective, avoiding high-temperature annealing is advantageous as it reduces energy consumption. This approach contributes to a more sustainable and environmentally friendly synthesis process.

To address your concerns, we performed XPS analysis on the catalyst before and after the reaction. By examining the C 1s spectra, particularly the C-O peak, we assessed the extent of carbon oxidation (**Supplementary Fig. 11**). The results indicated no significant increase in the C-O peak intensity post-reaction, suggesting that the FeN₅-SD_x catalyst exhibits enhanced resistance to substrate oxidation.

In summary, our material not only outperforms those synthesized through high-temperature methods in terms of stability and activity but also offers significant benefits in terms of energy efficiency and structural integrity. We believe these factors underscore the advantages of our approach for practical applications.

Fig. R9 (a) Impact of different catalyst preparation methods on BPA removal and (b) the corresponding kinetic constant. Recyclability of (c) FeN₅-SD₂/PMS system and (d) Fe-N-C-HT/PMS system for BPA removal.

Fig. R10 (a) FTIR spectra of FeN₅-SD₂, Fe-N-C-HT and NCNTs. (b) XPS Fe 2p spectra of Fe-N-C-HT and FeN₅-SD₂.

Supplementary Fig. 11 XPS C 1s spectra of (a) FeN₅, (b) FeN₅-SD₁, (c) FeN₅-SD₁, and (d) FeN₅-SD₁ before and after use.

Action 2: We have added related statements to validate the oxidation resistance of FeN₅-SD_x in the revised manuscript and supporting information (Supplementary Fig. 11).

“While XPS C 1s spectra showed no notable increase in the C–O peak, indicating strong resistance to carbon oxidation (Supplementary Fig. 11). These results confirm that the FeN₅-SD_x catalysts retain their structural integrity under reaction conditions.”

Supplementary Fig. 11 XPS C 1s spectra of (a) FeN₅, (b) FeN₅-SD₁, (c) FeN₅-SD₂, and (d) FeN₅-SD₃ before and after use.

3. There have been numerous previous reports on constructing axially coordinated single-atom materials on carbon nanotubes, such as *Adv. Sci.* 2023, 10, 2206107. What are the specific advantages or novel aspects of this study compared to existing work in the field?

Response 3:

We thank the reviewer for highlighting the importance of clarifying the novelty of our work relative to existing studies, such as *Adv. Sci.* 2023, 10, 2206107. While previous reports have indeed made significant progress in constructing axially coordinated SACs on carbon-based substrates, particularly focusing on how axial or in-plane coordination modulates the electronic environment of MN₄ centers, our work introduces several advancements that have been seldom explored.

Coordination modulation beyond the MN₄ platform:

Most existing studies focus on tuning the local coordination environment of MN₄ centers, either by modulating the number of in-plane nitrogen atoms or by introducing heteroatom dopants such as S, P, or O, as well as incorporating axial ligands. However, our study moves beyond the conventional MN₄-centered regulation strategy by focusing on FeN₅ as the target coordination structure. Specifically, we investigate the modulation of the high coordination shell environment surrounding the FeN₅ center, which has not been systematically explored in the context of single-atom catalysis. To the best of our knowledge, this is the first report to systematically elucidate the role of

high coordination shell defects in regulating the electronic structure of FeN₅ centers for PMS activation. This work expands the design space of SACs beyond the widely studied MN₄ motif and establishes new structure–property relationships relevant to PMS activation.

High coordination shell engineering:

Another key innovation of this work is the introduction of high coordination shell regulation—a largely unexplored dimension in the design of SACs. While prior studies have focused predominantly on modulating first- and second-shell coordination environments, the effect of higher coordination remains elusive due to the inherent challenge of achieving precise spatial control at such length scales. To address this, we developed a pyrolysis-free, low-temperature polymerization strategy that enables the precise incorporation of structural defects at predefined locations through rational functionalization of organic monomers (**Fig. 1a and Supplementary Fig. 1**). This approach allows for precise modulation of the coordination microenvironment surrounding the FeN₅ site—an achievement that is virtually unattainable *via* conventional high-temperature pyrolysis methods, where active site formation is inherently stochastic.

Establishing a clear structure-property relationship through remote coordination engineering:

A longstanding challenge in the field of single-atom catalysis lies in the structural ambiguity of thermally derived active sites, which hinders the establishment of definitive structure-property correlations. Our pyrolysis-free strategy overcomes this limitation by enabling precise control over the spatial positioning of defect sites within a conjugated polyphthalocyanine framework. This design allows us to concurrently tailor multiple coordination shells around the FeN₅ center, thereby achieving a well-defined and quantifiable coordination environment. Importantly, we demonstrate that high-shell defect engineering exerts a remote electronic influence on the FeN₅ center by modulating its d-band electronic structure (**Fig. 5a**). This coordination tuning correlates with a defect-content-dependent volcano-like trend in PMS adsorption and activation, providing direct evidence for a structure-activity relationship (**Fig. 5b**). Mechanistically, the introduced remote defects enhance charge delocalization across the reaction interface and significantly elongate the O-O bond in adsorbed PMS molecules. This structural change lowers the energy barrier for the generation of key reactive intermediates, ultimately favoring the selective formation of ¹O₂ over non-selective radical pathways (**Fig. 5c-g**).

These findings not only provide mechanistic clarity, previously obscured by the stochastic nature of active site formation, but also establish a new paradigm for designing SACs with tunable electronic structures through precise high coordination shell engineering.

[Figure Redacted]

Fig. 1 (a) Schematic of the preparation strategy for FeN₅-SD_x.

Supplementary Fig. 1 Formation procedure of FeN₅-SD_x.

Editorial Note: The crystal structures in Fig. 5c were visualized using VESTA software (Momma, K. & Izumi, F. VESTA 3 for three-dimensional visualization of crystal, volumetric and morphology data, J. Appl. Crystallogr. 44, 1272–1276 (2011).)

Fig. 5 The mechanistic of high coordination shell defects in regulating FeN₅ active sites. (a) Density of states of FeN₅, FeN₅-SD₁, FeN₅-SD₂ and FeN₅-SD₃. (b) Correlation among the oxidation capacity, coupling strength to PMS, and d-band structure of FeN₅, FeN₅-SD₁, FeN₅-SD₂ and FeN₅-SD₃. (c) Electron density difference diagrams and Bader charges for PMS adsorbed on FeN₅, FeN₅-SD₁, FeN₅-SD₂ and FeN₅-SD₃, with blue and yellow indicating electron depletion and accumulation, respectively. (d) Calculated potential energy diagrams for various bond-cleavage pathways during PMS decomposition on FeN₅-SD₂. (e) Calculated potential energy diagrams for the ¹O₂ formation pathway. (f) Schematic diagram of ¹O₂ generation process. (g) Length of the O–O bonds after adsorption of PMS adsorbed on different catalysts.

Action 3: We have added related statement in the Introduction section of the revised manuscript.

“Beyond the first-shell coordination, the environment of the second and higher coordination shells, which are not directly bonded to the metal sites, also plays a critical role in shaping the electronic structure of the active site and influencing

catalytic efficiency. Engineered coordination defects in higher coordination shells significantly affect the Fe active sites through a remote modulation effect. This “remote modulation” refers to catalytic changes induced by structural defects or heteroatoms located more than $\sim 4 \text{ \AA}$ from the Fe center^{9, 10, 11}, beyond the primary (directly bonded) and secondary (bonded to first-shell atoms) coordination spheres. This realm remains relatively unexplored and has seldom been addressed in studies concerning PMS activation, possibly attributed to uncertainties in the local structure of active sites stemming from uncontrolled pyrolysis synthesis. To overcome this limitation, precise engineering of the high coordination shell environment is essential for uncovering the structure–property relationships of SACs beyond conventional MN_4 centers. By targeting FeN_5 configurations and drawing inspiration from the spatial precision of natural metalloenzymes, this work aims to advance the design of SACs for more efficient PMS activation.”

“This study broadens the design of SACs beyond the conventional MN_4 motif by tuning multiple coordination shells simultaneously. It establishes new structure–property relationships for PMS activation and offering insights into the rational design of next-generation catalysts for water purification.”

4. Fig. 1a displays the configurations of FeN_5 , $\text{FeN}_5\text{-SD}_1$, $\text{FeN}_5\text{-SD}_2$ and $\text{FeN}_5\text{-SD}_3$, but the accompanying explanation is inadequate. Please furnish more experimental or computational data to rigorously validate the rationality and objectivity of these structural configuration.

Response 4:

We thank the reviewer for this insightful comment regarding the need for more rigorous validation of the structural configurations proposed in **Fig. 1a**. In response, we have conducted a comprehensive suite of experimental and computational analyses to substantiate the rationality and structural integrity of the $\text{FeN}_5\text{-SD}_x$ series, particularly focusing on the impact and control of high coordination shell defects.

In the FTIR spectra, all $\text{FeN}_5\text{-SD}_x$ samples retain the characteristic absorption bands of nitrogen-doped carbon nanotubes (NCNTs), while additional peaks corresponding to phthalocyanine vibrations are clearly observed (**Fig. 1e**). Specifically, the $\text{-C}\equiv\text{N}$ stretching vibration at 2240 cm^{-1} , characteristic of the BTC monomer, disappears progressively, accompanied by the emergence of $\text{C}=\text{N}$ (1500 cm^{-1} and 1135 cm^{-1}) and $\text{C}-\text{N}$ (1310 cm^{-1}) vibrations¹. These spectral changes confirm the successful polymerization of BTC into a conjugated phthalocyanine network *via* $\text{C}=\text{N}$ bond formation. As the defect concentration increases, the intensity of the phthalocyanine-

related C=N and C-N bands decrease moderately, suggesting partial disruption of the polyphthalocyanine framework due to coordination defects. Meanwhile, the C=C aromatic stretching band remains largely unchanged, indicating that the incorporation of DCB predominantly interferes with the formation of pyrrole ring *via* the loss of C=N or C-N units during phthalocyanine polymerization, while preserving the integrity of the benzene rings. It is noteworthy that several phthalocyanine vibrational bands in FeN₅-SD₂ and FeN₅-SD₃ are almost undetectable, which can be attributed to significant spectral overlap with the NCNTs background. To clarify this, we further analyzed the FTIR spectra of NCNTs-free samples (denoted as FeN₄-SD₂ and FeN₄-SD₃). As shown in **Fig. R1**, well-defined phthalocyanine vibrational modes are clearly retained in both FeN₄-SD₂ and FeN₄-SD₃, confirming that the structural framework of the polyphthalocyanine remains largely intact despite the presence of defects.

These trends are corroborated by Raman spectroscopy (**Fig. 1f**). In addition to the prominent D and G bands of NCNTs, the FeN₅-SD_x catalysts exhibit distinct peaks corresponding to the A_{1g}, B_{1g}, and B_{2g} modes, which originate from C-C stretching within the aromatic macrocycle and C-N-C bridging vibrations—typical features of polyphthalocyanine². The gradual attenuation of these peaks with increasing defect concentration supports the interpretation that the defects induce a localized and controlled structural perturbation, rather than a complete collapse of the polyphthalocyanine network.

To further confirm the structural integrity and local coordination environment of FeN₅-SD_x, we employed solid-state ¹³C NMR spectroscopy using NCNT-free FeN₄-SD_x samples as models. As shown in **Supplementary Fig. 5**, a broad resonance signal at approximately 100-130 ppm is assigned to aromatic carbons in the benzene rings of the covalent framework, while a distinct signal at ~171 ppm corresponds to the C=N groups in the pyrrole ring of phthalocyanine unit^{1, 2, 3}. The presence of this 171 ppm signal provides direct evidence for the successful formation of the conjugated phthalocyanine framework. Importantly, as the defect concentration increases from FeN₄ to FeN₄-SD₃, the 171 ppm signal exhibits a systematic decrease in intensity, indicative of a gradual reduction in pyrrolic C=N content. This trend strongly suggests the formation of high coordination shell defects *via* selective disruption of the pyrrole units within the phthalocyanine ring. In contrast, the resonance signals corresponding to the benzene ring carbons remain largely unaffected, confirming that the structural perturbations are localized and do not compromise the integrity of the aromatic backbone.

Complementary XPS analysis corroborates these findings. The C 1s spectra (**Supplementary Fig. 6**) show a clear decrease in the relative proportion of C=N and C-N components with increasing defect concentration. In addition, quantitative analysis of N content derived from XPS (**Table R1**) reveals a consistent downward trend, further supporting the partial loss of coordinated nitrogen due to defect incorporation. Brunauer–Emmett–Teller (BET) analysis (**Supplementary Table 1**) further confirms a gradual increase in surface area and pore volume with defect introduction, while maintaining the overall covalent framework.

In summary, the combined spectroscopic data provide robust evidence for the presence of high coordination shell defects in the FeN₅-SD_x catalysts. These defects arise from a targeted and controllable perturbation of the phthalocyanine coordination structure, rather than random distributed.

Theoretical validation *via* DFT calculations

We would like to clarify that **Fig. 1a** presents only a local structural unit to highlight the coordination environment, which may have inadvertently suggested a risk of structural collapse. However, from a broader perspective, the FeN₅-SD_x catalysts are constructed as extended covalently bonded polymeric frameworks. While the introduction of high coordination shell defects lead to increased porosity, the overall network remains robust due to the intrinsic covalent connectivity.

To further substantiate the structural stability at the atomic level, we conducted full geometry optimizations for FeN₅, FeN₅-SD₂, and FeN₅-SD₃ models using density functional theory (DFT). The optimized structures, now provided in **Fig. R5-R7**, reveal that all configurations converge to thermodynamically stable geometries without distortion or collapse. Crucially, the preservation of key coordination motifs across these models confirms that the introduction of coordination defects does not induce Fe aggregation or destabilization, even at higher defect densities. These results provide strong theoretical validation for the structural integrity of the proposed FeN₅-SD_x framework.

Fig. 1 (e) FTIR spectra of BTC, FeN₅, FeN₅-SD₁, FeN₅-SD₂, FeN₅-SD₃ and NCNTs.

Fig. R1 FTIR spectra of BTC, FeN₄-SD₂, and FeN₄-SD₃.

Fig. 1 (f) Raman spectra of BTC, FeN₅, FeN₅-SD₁, FeN₅-SD₂, FeN₅-SD₃ and NCNTs.

Supplementary Fig. 5 Solid-state ^{13}C NMR spectra and ChemDraw 20.0 “predicted” of FeN_4 , $\text{FeN}_4\text{-SD}_1$, $\text{FeN}_4\text{-SD}_2$ and $\text{FeN}_4\text{-SD}_3$.

Supplementary Fig. 6 XPS C1s spectra of FeN_4 , $\text{FeN}_4\text{-SD}_1$, $\text{FeN}_4\text{-SD}_2$ and $\text{FeN}_4\text{-SD}_3$.

Table R1. XPS analysis of N content in FeN_5 , $\text{FeN}_5\text{-SD}_1$, $\text{FeN}_5\text{-SD}_2$ and $\text{FeN}_5\text{-SD}_3$.

Catalysts	N 1s (wt%)
FeN_5	7.94
$\text{FeN}_5\text{-SD}_1$	6.71
$\text{FeN}_5\text{-SD}_2$	5.66
$\text{FeN}_5\text{-SD}_3$	4.87

Editorial Note: The crystal structures in Fig. R5 and R6 were visualized using VESTA software (Momma, K. & Izumi, F. VESTA 3 for three-dimensional visualization of crystal, volumetric and morphology data, J. Appl. Crystallogr. 44, 1272–1276 (2011).)

Supplementary Table 1. BET surface area and total pore volume a of FeN₅, FeN₅-SD₁, FeN₅-SD₂ and FeN₅-SD₃

Catalysts	BET specific surface area (m ² g ⁻¹)	Total pore volume (cm ³ g ⁻¹)
FeN ₅	43.78	0.1913
FeN ₅ -SD ₁	46.86	0.3856
FeN ₅ -SD ₂	55.63	0.4295
FeN ₅ -SD ₃	65.83	0.4335

Fig. R5 The optimal configurations of FeN₅.

Fig. R6 The optimal configurations of FeN₅-SD₂.

Editorial Note: The crystal structures in Fig. R7 were visualized using VESTA software (Momma, K. & Izumi, F. VESTA 3 for three-dimensional visualization of crystal, volumetric and morphology data, J. Appl. Crystallogr. 44, 1272–1276 (2011).)

Fig. R7 The optimal configurations of FeN₅-SD₃.

Action 4: We have the added relevant statements in the revised manuscript (Fig. 1e-f). Supplementary Table 1 and Supplementary Figs. 5-6 have also been included in the supporting information.

“Fourier transform infrared (FTIR) spectra of all FeN₅-SD_x samples exhibit the characteristic absorption bands of NCNTs, along with additional peaks assigned to polyphthalocyanine vibrations (Fig. 1e). The progressive disappearance of the -C≡N stretching vibration at 2240 cm⁻¹, characteristic of the BTC monomer, is accompanied by the emergence of new peaks corresponding to C=N (1500 and 1135 cm⁻¹) and C-N (1310 cm⁻¹) stretching modes. These changes indicate successful polymerization into a polyphthalocyanine-like framework. A strong absorption at 1396 cm⁻¹ is attributed to C=C stretching within the conjugated macrocycle, while a peak at 731 cm⁻¹ arises from C-H vibrations, further confirming the formation of extended polyphthalocyanine structure¹. With increasing defect concentration, the intensity of the C=N and C-N bands moderately decreases, suggesting partial disruption of the pyrrole units within the phthalocyanine ring. In contrast, the C=C band remains stable, implying that the benzene moieties are preserved. These observations are supported by Raman spectra (Fig. 1f), where in addition to the D and G bands of NCNTs, distinct A_{1g}, B_{1g}, and B_{2g} modes related to C-C and C-N-C vibrations in the phthalocyanine ring are observed.². The gradual attenuation of these peaks with implies localized structural perturbations rather than a complete collapse of the polyphthalocyanine network. To further examine the coordination environment, solid-state ¹³C nuclear magnetic resonance (NMR) spectroscopy was conducted using NCNT-free FeN₄-SD_x samples. As shown in

Supplementary Fig. 5, a broad resonance at 100-130 ppm is assigned to aromatic carbons in benzene rings, while a distinct signal at ~171 ppm corresponds to C=N groups in the pyrrole ring of phthalocyanine unit^{1, 2, 3}. The progressive decrease in the 171 ppm signal from FeN₄ to FeN₄-SD₃ reflects a gradual loss of pyrrolic C=N groups, consistent with the formation of high coordination shell defects. Meanwhile, the signals from benzene carbons remain largely unchanged, confirming that structural disruption is confined to the pyrrole moieties. X-ray photoelectron spectroscopy (XPS) C 1s spectra (Supplementary Fig. 6) show a decrease in C=N and C-N components with increasing defect concentration, supporting the partial loss of coordinated nitrogen. Brunauer–Emmett–Teller (BET) analysis (Supplementary Table 1) further confirms a progressive increase in surface area and pore volume with defect introduction, while preserving the overall covalent framework.”

Fig. 1 (e) FTIR spectra of BTC, FeN₅, FeN₅-SD₁, FeN₅-SD₂, FeN₅-SD₃ and NCNTs.

Fig. 1 (f) Raman spectra of BTC, FeN₅, FeN₅-SD₁, FeN₅-SD₂, FeN₅-SD₃ and NCNTs.

Supplementary Fig. 5 Solid-state ^{13}C NMR spectra and ChemDraw 20.0 “predicted” of FeN_4 , $\text{FeN}_4\text{-SD}_1$, $\text{FeN}_4\text{-SD}_2$ and $\text{FeN}_4\text{-SD}_3$.

Supplementary Fig. 6 XPS $\text{C}1\text{s}$ spectra of FeN_4 , $\text{FeN}_4\text{-SD}_1$, $\text{FeN}_4\text{-SD}_2$ and $\text{FeN}_4\text{-SD}_3$.

Supplementary Table 1. BET surface area and total pore volume of FeN_5 , $\text{FeN}_5\text{-SD}_1$, $\text{FeN}_5\text{-SD}_2$ and $\text{FeN}_5\text{-SD}_3$

Catalysts	BET specific surface area ($\text{m}^2 \text{g}^{-1}$)	Total pore volume ($\text{cm}^3 \text{g}^{-1}$)
FeN_5	43.78	0.1913
$\text{FeN}_5\text{-SD}_1$	46.86	0.3856
$\text{FeN}_5\text{-SD}_2$	55.63	0.4295

Editorial Note: The crystal structures in Supplementary Fig. 28 were visualized using VESTA software (Momma, K. & Izumi, F. VESTA 3 for three-dimensional visualization of crystal, volumetric and morphology data, J. Appl. Crystallogr. 44, 1272–1276 (2011).)

Catalysts	BET specific surface area ($\text{m}^2 \text{g}^{-1}$)	Total pore volume ($\text{cm}^3 \text{g}^{-1}$)
FeN ₅ -SD ₃	65.83	0.4335

5. Most studies reporting the activation of PMS to generate singlet oxygen (¹O₂) involve materials connected to the terminal oxygen of PMS. The authors should compare the adsorption energies of the two distinct oxygen sites of PMS.

Response 5:

We appreciate the reviewer’s insightful suggestion. To clarify the adsorption preference of different oxygen atoms in the PMS molecule, we systematically calculated the adsorption configurations and corresponding adsorption energies for three oxygen atoms on FeN₅, FeN₅-SD₁, FeN₅-SD₂ and FeN₅-SD₃. The results consistently show that, across all catalysts, the oxygen atom directly bonded to the sulfur atom exhibits the strongest adsorption affinity. This indicates that this site is the most thermodynamically favorable for PMS activation in our catalyst (see **Supplementary Figs. 28-31**).

Action 5: We have included the detailed adsorption configurations and energy values in the revised manuscript and supporting information (Supplementary Figs. 28-31).

“To clarify PMS adsorption preferences, the adsorption configurations and energies for three oxygen atoms on FeN₅, FeN₅-SD₁, FeN₅-SD₂ and FeN₅-SD₃ were calculated. In all cases, the terminal oxygen bonded to sulfur exhibited the strongest adsorption, identifying it as the preferred site for PMS activation (Supplementary Figs. 28-31). The optimized structures also show a gradual decrease in adsorption energy (E_{ads}) with increasing defect density, consistent with the DOS results (Fig. 5b).”

Supplementary Fig. 28 Comparative adsorption energies of PMS oxygen atoms on FeN₅.

Editorial Note: The crystal structures in Supplementary Fig. 29 and 30 were visualized using VESTA software (Momma, K. & Izumi, F. VESTA 3 for three-dimensional visualization of crystal, volumetric and morphology data, J. Appl. Crystallogr. 44, 1272–1276 (2011).)

Supplementary Fig. 29 Comparative adsorption energies of PMS oxygen atoms on FeN₅-SD₁.

Supplementary Fig. 30 Comparative adsorption energies of PMS oxygen atoms on FeN₅-SD₂.

Editorial Note: The crystal structures in Supplementary Fig. 31 were visualized using VESTA software (Momma, K. & Izumi, F. VESTA 3 for three-dimensional visualization of crystal, volumetric and morphology data, J. Appl. Crystallogr. 44, 1272–1276 (2011).)

Supplementary Fig. 31 Comparative adsorption energies of PMS oxygen atoms on FeN₅-SD₃.

6. Comprehensive assessment of catalytic performance under controlled gaseous environments (e.g., O₂, air, N₂) is essential to precisely delineate the role of dissolved oxygen, as it mediates both singlet oxygen (¹O₂) formation and potentially serves as a precursor for superoxide radical (O₂^{•-}) generation *via* electron transfer mechanisms.

Response 6:

We thank the reviewer for the valuable suggestion. To investigate the role of dissolved oxygen (DO) in reactive oxygen species (ROS) generation, we conducted control experiments under different gaseous atmospheres (O₂, air, and N₂). Interestingly, no significant variation in degradation efficiency was observed among these conditions (**Supplementary Fig. 25**), indicating that DO does not play a dominant role in the reaction process.

This result suggests that the formation of ¹O₂ is not dependent on molecular oxygen as a reactant. Instead, the ¹O₂ are likely generated through the intrinsic activation of PMS at the FeN₅ catalytic sites, independent of external oxygen input. This finding highlights the capability of our catalyst to drive nonradical oxidation pathways even in oxygen-deficient environments, further confirming that the observed reactivity originates from catalyst-PMS interactions rather than DO-mediated pathways.

Action 6: We have added relevant statements in the revised manuscript and Supplementary Fig. 25 has been added in the supporting information.

“Control experiments under O₂, air, and N₂ atmospheres revealed negligible

differences in degradation efficiency (Supplementary Fig. 25), indicating that dissolved oxygen (DO) is not essential to the reaction. This suggests that $^1\text{O}_2$ generation proceeds via direct activation of PMS at FeN_5 sites, independent of external oxygen input.”

Supplementary Fig. 25 (a-d) Effect of different gas atmospheres (N_2 , air, and O_2) on BPA degradation in various catalytic systems.

7. Comparisons with recent single-atom catalysts lack rigor due to inconsistent pollutant selection. Degradation efficiency varies significantly with molecular structures (e.g., bond types, substituent groups). To ensure valid benchmarking, use identical target pollutants for cross-study evaluations.

Response 7:

Thank you for your insightful comment regarding the critical importance of consistent pollutant selection for the rigorous benchmarking of SACs. We fully agree with your assessment that variations in pollutant molecular structures, such as bond types and substituent groups, significantly impact degradation efficiency. We acknowledge the importance of consistent pollutant selection for benchmarking SAC performance. However, benchmarking against existing literature presents challenges, as relatively few studies on coordination-regulated SACs for PMS activation employ bisphenol A (BPA) as the target pollutant.

Furthermore, numerous additional variables in distinct catalytic systems (such as

catalyst dosage, pollutant concentration, and metal loading) complicate direct performance comparisons. To address these limitations and facilitate a more rigorous comparison that minimizes the influence of pollutant-specific kinetics and system-specific parameters, we adopted turnover frequency (TOF) as a key metric. TOF, defined as the number of moles of pollutant degraded per mole of active site per unit time, reflects the intrinsic activity of catalytic sites independent of external variables^{12, 13, 14, 15}. The TOF is calculated as follows:

$$\text{TOF} [\text{min}^{-1}] = \frac{\text{moles of reactant converted}}{\text{moles of active sites} \times \text{reaction time}} = \frac{\Delta n(\text{pollutant}) \times M_{\text{metal}}}{m_0 \times \omega_{\text{metal}} \times t}$$

Where Δn (pollutant) is moles converted, M_{metal} is metal atomic weight, m_0 is catalyst mass, ω_{metal} is metal mass fraction, and t is reaction time. All parameters were derived from our kinetic experiments and site quantification analyses.

This normalization reduces direct dependency on the specific degradation kinetics of a particular pollutant and provides a more standardized basis for comparing intrinsic site activity across diverse systems than raw degradation efficiencies under differing conditions.

By using TOF as a normalized performance indicator, we significantly reduce dependency on pollutant identity and experimental configuration, enabling more meaningful cross-study comparisons. As shown in **Fig. 3 c and Supplementary Table 5**, our FeN₅-SD₂ catalyst exhibits a notably high TOF value of 0.338 min⁻¹, outperforming other recently reported coordination-regulated SACs in PMS systems. We believe this approach offers a more robust and fair benchmarking strategy given the unavoidable heterogeneity in pollutant selection and experimental design across published studies.

Action 7: We have the added relevant statements in the revised manuscript (Fig. 3c) and Supplementary Table 5 has been added in the supporting information.

“The turnover frequency (TOF) was employed to quantitatively evaluate the intrinsic reactivity against previously reported SAC-based PMS activation systems. As shown in Fig. 3c and Supplementary Table 5, FeN₅-SD₂ catalyst exhibits a significantly high TOF of 0.338 min⁻¹. This performance surpasses other recently reported coordination-regulated SACs in PMS systems.”

Fig. 3 (c) Comparison of TOF values for pollutant degradation in SAC/PMS systems featuring coordination environment regulation.

Supplementary Table 5. Catalytic performances of SAC-based PMS systems regulated *via* coordination environment engineering.

Catalysts (g L ⁻¹)	$\Delta n(\text{pollutants})$ ($\times 10^{-6}$ mol)	ω_{metal} (%)	Reaction time (min)	TOF (min ⁻¹)	Ref.
Fe-N ₃ C ₁ (0.06)	SIZ (19.6)	0.75	30	0.081	16
Fe-N ₂ C ₂ (0.06)	SIZ (17)	0.84	30	0.063	16
Co-OCN (0.03)	APAP (13)	7.25	60	0.006	17
Fe-N-C (0.10)	BPA (100)	2.00	20	0.140	18
Co-N ₃ /CNT (0.03)	SMZ (40)	1.39	30	0.189	19
CNFe ₂ -0.6 (0.08)	SMZ (79)	16.64	8	0.042	20
Fe ₁ /CN (0.05)	4-CP (100)	11.2	10	0.100	21
FeSA-N/C-20 (0.15)	BPA (88)	0.88	20	0.187	22
FeSA-N/O-C (0.10)	BPA (66)	1.28	45	0.064	23
Co-N ₃ (0.10)	CBZ (42)	2.51	6	0.165	24

Catalysts (g L⁻¹)	$\Delta n(\text{pollutants})$ ($\times 10^{-6}$ mol)	ω_{metal} (%)	Reaction time (min)	TOF (min⁻¹)	Ref.
CoN ₃ O ₁ (0.10)	CIP (15)	0.51	20	0.087	25
Fe-N ₄ -C (0.10)	NPX (43)	1.50	10	0.161	26
SA-FeN ₅ (0.20)	APAP (30)	2.63	12	0.027	27
FeN ₅ (0.03)	BPA (52)	2.88	20	0.169	This work
FeN ₅ -SD ₁ (0.03)	BPA (56)	2.59	20	0.202	This work
FeN₅-SD₂ (0.03)	BPA (80)	2.21	20	0.338	This work
FeN ₅ -SD ₃ (0.03)	BPA (59)	1.69	20	0.325	This work

8. The authors ascribe significant importance to ¹O₂ in the reaction mechanism, and this proposition appears to be corroborated by experimental evidence. However, given that ¹O₂ is a less potent oxidizing agent compared to hydroxyl radicals and many organic compounds may exhibit limited susceptibility to its oxidative effects, what specific mechanistic advantages does the ¹O₂-dominated pathway confer within this system?

Response 8:

We thank the reviewer for raising this important question regarding the mechanistic advantages of the ¹O₂-dominated pathway in the reaction mechanism. We acknowledge that while ¹O₂ is indeed a less potent oxidizing agent compared to hydroxyl radicals, it offers distinct advantages in specific catalytic systems, particularly in the context of advanced oxidation processes (AOPs).

Moderate redox potential and broad pH compatibility:

One key advantage of ¹O₂ in our system is its moderate redox potential, which allows it to operate effectively across a wide range of pH values (Fig. 3d). Unlike hydroxyl radicals, which are highly reactive and typically operate under more restrictive conditions, ¹O₂ can be more stable and adaptable under varied pH conditions. This property is particularly beneficial in natural water environments, where pH can fluctuate, and AOPs often need to maintain stable oxidative activity in the presence of varying contaminants and ionic matrices.

Reduced interference from inorganic ions and organic matter:

One of the significant advantages of using $^1\text{O}_2$ over hydroxyl radicals is its reduced susceptibility to interference from inorganic ions (such as Cl^- , SO_4^{2-}) and background organic matter (**Fig. 3e**). In many real-water scenarios, hydroxyl radicals are highly reactive and can be quenched by such species, leading to a loss of oxidative efficiency. In contrast, $^1\text{O}_2$, with its lower reactivity, is more resistant to these interferences, making it particularly attractive for use in complex environmental matrices where the presence of competing species can otherwise hinder the performance of traditional radical-based AOPs.

Selective oxidation of organic pollutants:

While $^1\text{O}_2$ is indeed less reactive than hydroxyl radicals, it has been shown to exhibit high selectivity toward certain types of organic pollutants, particularly those containing conjugated double bonds, such as aromatic compounds (**Fig. 3f**). This selectivity allows $^1\text{O}_2$ to achieve highly efficient degradation of pollutants like BPA and other recalcitrant organic compounds, which are common targets in AOPs. The $^1\text{O}_2$ pathway thus allows for a more targeted oxidative attack, minimizing the generation of undesired by-products and enhancing the overall efficiency of pollutant removal.

Enhanced stability and reusability of catalysts:

The $^1\text{O}_2$ -dominated pathway offers the additional benefit of enhanced catalyst stability. Unlike hydroxyl radicals, which are highly reactive and can cause rapid catalyst degradation or initiate side reactions, $^1\text{O}_2$ exhibits a more controlled oxidative behavior. This controlled reactivity helps preserve the catalyst's stability over prolonged reaction periods, leading to longer operational lifetimes and reduced deactivation. This stability is particularly crucial in practical, continuous-flow applications, where catalyst longevity plays a vital role (**Fig. 6e-j**).

In summary, while $^1\text{O}_2$ is a less powerful oxidant than hydroxyl radicals, its moderate reactivity, selective oxidation capabilities, resistance to environmental interference, and positive impact on catalyst stability provide significant mechanistic advantages in certain AOP systems, particularly in the degradation of organic pollutants under diverse environmental conditions. These advantages make the $^1\text{O}_2$ -dominated pathway a promising approach for AOP, as demonstrated by our experimental results.

Fig. 3 The effects of (d) pH and (e) inorganic anions and HA on the removal of BPA in FeN₅-SD₂/PMS system. (f) Different pollutants removal in FeN₅-SD₂/PMS system. Routine conditions: [pollutants] = 80 μM, [catalyst] = 0.03 g L⁻¹, [PMS] = 0.05 mM, temperature = 25 °C, without pH adjustment.

[Figure Redacted]

Fig. 6 (e) Schematic illustration of wastewater treatment process. (f) FeN₅-SD₂/PVDF membrane. (g) Photograph of the wastewater treatment experimental equipment. (h) Water flux of the membrane. (i) Removal efficiency of different systems. (j) BPA removal efficiency and Fe leaching of FeN₅-SD₂/PVDF membrane system in different water matrices.

Once again, we truly thank you for the insightful comments and kind suggestions.

Reviewer #3 (Remarks to the Author):

The manuscript investigated the role of high-shell coordination defects in Fe-N₅ single-atom catalysts (SACs) for peroxymonosulfate (PMS) activation in Fenton-like reactions. Many recent studies have already explored defect engineering in SACs for PMS activation, and the manuscript fails to articulate a distinct advance over prior work. While the study tried to correlate defect engineering with catalytic performance, the

novelty and mechanistic insights remained unclear, and it does not convincingly demonstrate how this system was fundamentally different from existing Fe-N₄ or other SACs. The following aspects require further improvement:

Response: We thank the reviewer for the constructive comments. While previous studies have indeed explored defect engineering in SACs for PMS activation, particularly in terms of how first- and second-coordination environments modulate the electronic structure of MN₄ centers, our work introduces several advancements that have been seldom explored.

Coordination modulation beyond the MN₄ platform:

Most existing studies focus on tuning the local coordination environment of planar MN₄ sites, often by varying the number of nitrogen atoms or introducing heteroatom dopants in the vicinity of the metal center^{28, 29}. However, our study moves beyond this conventional approach by exploring a fundamentally different catalytic architecture—FeN₅. Specifically, we investigate the modulation of the high coordination shell environment surrounding the FeN₅ center, which has not been systematically explored in the context of single-atom catalysis. To the best of our knowledge, this is the first report to systematically elucidate the role of high coordination shell defects in regulating the electronic structure of FeN₅ centers for PMS activation. This work expands the design space of SACs beyond the widely studied MN₄ motif and establishes new structure–property relationships relevant to PMS activation.

High coordination shell engineering:

Another key innovation of this work is the introduction of high coordination shell regulation—a largely unexplored dimension in the design of SACs. While prior studies have focused predominantly on modulating first- and second-shell coordination environments, the effect of higher coordination remains elusive due to the inherent challenge of achieving precise spatial control at such length scales. To address this, we developed a pyrolysis-free, low-temperature polymerization strategy that enables the precise incorporation of structural defects at predefined locations through rational functionalization of organic monomers (**Fig. 1a** and **Supplementary Fig. 1**). This approach allows for precise modulation of the coordination microenvironment surrounding the FeN₅ site—an achievement that is virtually unattainable *via* conventional high-temperature pyrolysis methods, where active site formation is inherently stochastic.

Establishing a clear structure-property relationship through remote coordination engineering

A longstanding challenge in the field of single-atom catalysis lies in the structural ambiguity of thermally derived active sites, which hinders the establishment of definitive structure-property correlations. Our pyrolysis-free strategy overcomes this limitation by enabling precise control over the spatial positioning of defect sites within a conjugated polyphthalocyanine framework. This design allows us to concurrently tailor multiple coordination shells around the FeN₅ center, thereby achieving a well-defined and quantifiable coordination environment. Importantly, we demonstrate that high-shell defect engineering exerts a remote electronic influence on the FeN₅ center by modulating its d-band electronic structure (**Fig. 5a**). This coordination tuning correlates with a defect-content-dependent volcano-like trend in PMS adsorption and activation, providing direct evidence for a structure-activity relationship (**Fig. 5b**). Mechanistically, the introduced remote defects enhance charge delocalization across the reaction interface and significantly elongate the O-O bond in adsorbed PMS molecules. This structural change lowers the energy barrier for the generation of key reactive intermediates, ultimately favoring the selective formation of ¹O₂ over non-selective radical pathways (**Fig. 5c-g**).

These findings not only provide mechanistic clarity, previously obscured by the stochastic nature of active site formation, but also establish a new paradigm for designing SACs with tunable electronic structures through precise high coordination shell engineering.

[Figure Redacted]

Fig. 1 (a) Schematic of the preparation strategy for FeN₅-SD_x.

Editorial Note: The crystal structures in Fig. 5c were visualized using VESTA software (Momma, K. & Izumi, F. VESTA 3 for three-dimensional visualization of crystal, volumetric and morphology data, J. Appl. Crystallogr. 44, 1272–1276 (2011).)

Supplementary Fig. 1 Formation procedure of FeN₅-SD_x.

Fig. 5 The mechanistic of high coordination shell defects in regulating FeN₅ active sites. (a) Density of states of FeN₅, FeN₅-SD₁, FeN₅-SD₂ and FeN₅-SD₃. (b) Correlation among the oxidation capacity, coupling strength to PMS, and d-band structure of FeN₅, FeN₅-SD₁, FeN₅-SD₂ and FeN₅-SD₃. (c) Electron density difference diagrams and Bader charges for PMS adsorbed on FeN₅, FeN₅-SD₁, FeN₅-SD₂ and FeN₅-SD₃, with blue and yellow indicating electron depletion and accumulation, respectively. (d)

Calculated potential energy diagrams for various bond-cleavage pathways during PMS decomposition on FeN₅-SD₂. (e) Calculated potential energy diagrams for the ¹O₂ formation pathway. (f) Schematic diagram of ¹O₂ generation process. (g) Length of the O–O bonds after adsorption of PMS adsorbed on different catalysts.

1. **(1.1)** The authors focused heavily on the role of high-shell coordination defects and introduced the concept of "remote modulation" of Fe-N₅ active sites. However, the manuscript did not provide a clear definition of what constituted "remote" in this context. It was important to clarify how the spatial range of remote modulation was determined. For example, what specific atomic or structural distance qualified a coordination interaction as remote rather than part of the primary or secondary coordination sphere? **(1.2)** Additionally, the authors should have specified how this remote interaction was identified or quantified using experimental or theoretical methods. Without a measurable or defined parameter, the concept of "remote modulation" remained too vague, which undermined the mechanistic clarity of the study. A more explicit explanation and supporting data were needed to convincingly establish the structural basis and catalytic relevance of these high-shell coordination effects.

Response 1.1:

We sincerely thank the reviewer for raising this important point. To clarify the concept of "remote modulation" we have carefully reviewed the literature regarding the classification of coordination shells and provide the following definitions:

Coordination shells definition:

As defined in previous studies, the first coordination shell refers to heteroatoms that are directly bonded to the single metal atom. The second coordination shell consists of heteroatoms that are bonded to atoms in the first coordination shell but not directly to the metal center. Higher coordination shells are those beyond the second coordination shell, typically including atoms that are further away but still capable of influencing the active site through remote interactions.

In our study, "remote modulation" specifically refers to the influence exerted by atoms or structural elements that are located in higher coordination shells—meaning they are not directly bonded to the metal center but still affect its electronic configuration or coordination environment. Chen et al.³⁰ have reported that both the second and higher coordination shells can influence the active sites *via* long-range electronic delocalization, especially in systems involving exotic atoms or carbon-supported substrates. This type of modulation, as described in their work, is similar to the "remote modulation" observed in our study, where the atoms alter the active site's

properties without direct coordination to the metal center.

To clarify the spatial range of “remote modulation” we refer to the classification of coordination interactions as described in the literature. Specifically, coordination interactions are classified as “higher coordination shells interactions” if the distance between the modulating atom or structural feature and the central Fe atom exceeds the typical distances associated with the primary and secondary coordination spheres, which are generally within 1-4 Å^{9, 10, 11}. Interactions beyond this range, still capable of influencing the electronic properties of the Fe site, are considered as higher coordination shells interactions^{10, 31}. In our study, we use the term “remote modulation” to describe these interactions. Therefore, “remote” is effectively another term for higher coordination shells, and both concepts refer to the same phenomenon but with different terminology.

To quantify this effect, we measure bond lengths and assess their influence on the electronic structure of the Fe site. For example, our DFT calculations show that defect sites introduced by DCB, though located more than 5 Å from the Fe center, still significantly impact the electronic structure of the FeN₅ site (**Supplementary Fig. 2 and Fig. 5a-b**). This confirms that the term “remote” modulation is justified and aligns with the concept of higher coordination shells rather than part of the primary or secondary coordination sphere.

[Figure Redacted]

Fig. R11 Coordination environment of single-atom dispersed metal sites. Literature reports:(a)³², (b)³³, (c)³⁰ and (d)¹¹. (e) Our study.

Editorial Note: The crystal structures in Supplementary Fig. 2 were visualized using VESTA software (Momma, K. & Izumi, F. VESTA 3 for three-dimensional visualization of crystal, volumetric and morphology data, J. Appl. Crystallogr. 44, 1272–1276 (2011).)

Supplementary Fig. 2 The spatial range of remote modulation in catalyst structures.

Fig. 5 (a) Density of states of FeN₅, FeN₅-SD₁, FeN₅-SD₂ and FeN₅-SD₃. (b) Correlation among the oxidation capacity, coupling strength to PMS, and d-band structure of FeN₅, FeN₅-SD₁, FeN₅-SD₂ and FeN₅-SD₃.

Action 1.1: We have added the relevant definitions in the revised manuscript (Fig. 1b) and the Supporting Information (Supplementary Fig. 2). Figure R11e has been incorporated into Fig. 1b in the revised manuscript.

“Beyond the first-shell coordination, the environment of the second and higher coordination shells, which are not directly bonded to the metal sites, also plays a critical role in shaping the electronic structure of the active site and influencing catalytic efficiency. Engineered coordination defects in higher coordination shells significantly affect the Fe active sites through a remote modulation effect. This “remote modulation” refers to catalytic changes induced by structural defects or heteroatoms located more than ~4 Å from the Fe center^{9, 10, 11}, beyond the primary (directly bonded) and secondary (bonded to first-shell atoms) coordination spheres.”

Editorial Note: The crystal structures in Supplementary Fig. 2 were visualized using VESTA software (Momma, K. & Izumi, F. VESTA 3 for three-dimensional visualization of crystal, volumetric and morphology data, J. Appl. Crystallogr. 44, 1272–1276 (2011).)

“To engineer the higher coordination shells around Fe centers, which typically include atoms beyond the primary and secondary shells, we introduced coordination defects (Fig. 1b). This was achieved by co-polymerizing a controlled amount of 1,2-dicyanobenzene (DCB) with benzene-1,2,4,5-tetracarbonitrile (BTC). Incorporating DCB systematically disrupts the polymer matrix, creating tailored defects in the higher coordination shells. These defects lie beyond 5 Å from the Fe center, as confirmed by DFT calculations (Supplementary Figs. 1-2), and are denoted as FeN₅-SD_x.”

[Figure Redacted]

Fig. 1 (b) Schematic illustration of the coordination structure of FeN₅-SD_x.

Supplementary Fig. 2 The spatial range of remote modulation in catalyst structures.

Response 1.2:

We thank the reviewer for highlighting the need to clarify how remote interactions were quantified. The concept of remote interaction in our study is supported by both experimental observations and theoretical calculations. First, our synthetic strategy enabled precise creation of higher coordination defects, confirmed through structural

characterization. Second, the impact of these remote interactions is evidenced by distinct differences in catalytic performance across samples with varying defect densities. Furthermore, variations in $^1\text{O}_2$ generation, adsorption thermodynamics, electron transfer efficiency, and reaction energy barriers collectively provide mechanistic insight and corroborate the role of remote interactions.

Feasibility of synthetic strategy for higher coordination defect formation

We present a pyrolysis-free, low-temperature polymerization strategy that enables precise incorporation of structural defects at predetermined sites through rational functionalization of organic monomers (**Fig. 1a and Supplementary Fig. 1**). This approach affords controlled modulation of the coordination environment surrounding the FeN_5 active site. Such precise control is difficult to achieve with traditional high-temperature pyrolysis methods, where active site formation is inherently stochastic.

The polyphthalocyanine framework inherently facilitates transition metal incorporation within its macrocyclic cavity. Concurrently, nitrogen-rich NCNTs promote axial Fe–N coordination, collectively stabilizing highly ordered FeN_5 centers as the dominant catalytic motifs. To introduce coordination defects, a controlled amount of 1,2-dicyanobenzene (DCB) was co-polymerized with benzene-1,2,4,5-tetracarbonitrile (BTC). The incorporation of DCB systematically perturbs the polymeric framework, generating high coordination shell defects localized around the metal centers (denoted as $\text{FeN}_5\text{-SD}_x$).

[Figure Redacted]

Fig. 1 (a) Schematic of the preparation strategy for $\text{FeN}_5\text{-SD}_x$.

Supplementary Fig. 1 Formation procedure of $\text{FeN}_5\text{-SD}_x$.

Coordination defects and structural evidence

We first revisited the FTIR, Raman, NMR, BET and XPS spectra to clarify the evolution of the structure and the incorporation of defects:

In the FTIR spectra, all $\text{FeN}_5\text{-SD}_x$ samples retain the characteristic absorption bands of nitrogen-doped carbon nanotubes (NCNTs), while additional peaks corresponding to phthalocyanine vibrations are clearly observed (**Fig. 1e**). Specifically, the $\text{-C}\equiv\text{N}$ stretching vibration at 2240 cm^{-1} , characteristic of the BTC monomer, disappears progressively, accompanied by the emergence of $\text{C}=\text{N}$ (1500 cm^{-1} and 1135 cm^{-1}) and $\text{C}-\text{N}$ (1310 cm^{-1}) vibrations¹. These spectral changes confirm the successful polymerization of BTC into a conjugated phthalocyanine network *via* $\text{C}=\text{N}$ bond formation. As the defect concentration increases, the intensity of the phthalocyanine-related $\text{C}=\text{N}$ and $\text{C}-\text{N}$ bands decrease moderately, suggesting partial disruption of the polyphthalocyanine framework due to coordination defects. Meanwhile, the $\text{C}=\text{C}$ aromatic stretching band remains largely unchanged, indicating that the incorporation of DCB predominantly interferes with the formation of pyrrole ring *via* the loss of $\text{C}=\text{N}$ or $\text{C}-\text{N}$ units during phthalocyanine polymerization, while preserving the integrity of the benzene rings. It is noteworthy that several phthalocyanine vibrational bands in $\text{FeN}_5\text{-SD}_2$ and $\text{FeN}_5\text{-SD}_3$ are almost undetectable, which can be attributed to significant spectral overlap with the NCNTs background. To clarify this, we further analyzed the FTIR spectra of NCNTs-free samples (denoted as $\text{FeN}_4\text{-SD}_2$ and $\text{FeN}_4\text{-SD}_3$). As shown in **Fig. R1**, well-defined phthalocyanine vibrational modes are clearly retained in both $\text{FeN}_4\text{-SD}_2$ and $\text{FeN}_4\text{-SD}_3$, confirming that the structural framework of the polyphthalocyanine remains largely intact despite the presence of defects.

These trends are corroborated by Raman spectroscopy (**Fig. 1f**). In addition to the prominent D and G bands of NCNTs, the $\text{FeN}_5\text{-SD}_x$ catalysts exhibit distinct peaks

corresponding to the A_{1g} , B_{1g} , and B_{2g} modes, which originate from C-C stretching within the aromatic macrocycle and C-N-C bridging vibrations—typical features of polyphthalocyanine². The gradual attenuation of these peaks with increasing defect concentration supports the interpretation that the defects induce a localized and controlled structural perturbation, rather than a complete collapse of the polyphthalocyanine network.

To further confirm the structural integrity and local coordination environment of FeN_5-SD_x , we employed solid-state ^{13}C NMR spectroscopy using NCNT-free FeN_4-SD_x samples as models. As shown in **Supplementary Fig. 5**, a broad resonance signal at approximately 100-130 ppm is assigned to aromatic carbons in the benzene rings of the covalent framework, while a distinct signal at ~171 ppm corresponds to the C=N groups in the pyrrole ring of phthalocyanine unit^{1, 2, 3}. The presence of this 171 ppm signal provides direct evidence for the successful formation of the conjugated phthalocyanine framework. Importantly, as the defect concentration increases from FeN_4 to FeN_4-SD_3 , the 171 ppm signal exhibits a systematic decrease in intensity, indicative of a gradual reduction in pyrrolic C=N content. This trend strongly suggests the formation of high coordination shell defects *via* selective disruption of the pyrrole units within the phthalocyanine ring. In contrast, the resonance signals corresponding to the benzene ring carbons remain largely unaffected, confirming that the structural perturbations are localized and do not compromise the integrity of the aromatic backbone. Complementary XPS analysis corroborates these findings. The C 1s spectra (**Supplementary Fig. 6**) show a clear decrease in the relative proportion of C=N and C-N components with increasing defect concentration. Brunauer–Emmett–Teller (BET) analysis (**Supplementary Table 1**) further confirms a gradual increase in surface area and pore volume with defect introduction, while maintaining the overall covalent framework.

To further substantiate the structural stability at the atomic level, we conducted full geometry optimizations for FeN_5 , FeN_5-SD_2 , and FeN_5-SD_3 models using density functional theory (DFT). The optimized structures, now provided in **Fig. R5-R7**, reveal that all configurations converge to thermodynamically stable geometries without distortion or collapse.

Fig. 1 (e) FTIR spectra of BTC, FeN_5 , $\text{FeN}_5\text{-SD}_1$, $\text{FeN}_5\text{-SD}_2$, $\text{FeN}_5\text{-SD}_3$ and NCNTs.

Fig. R1 FTIR spectra of BTC, $\text{FeN}_4\text{-SD}_2$, and $\text{FeN}_4\text{-SD}_3$.

Fig. 1 (f) Raman spectra of BTC, FeN_5 , $\text{FeN}_5\text{-SD}_1$, $\text{FeN}_5\text{-SD}_2$, $\text{FeN}_5\text{-SD}_3$ and NCNTs.

Supplementary Fig. 5 Solid-state ^{13}C NMR spectra and ChemDraw 20.0 “predicted” of FeN_4 , $\text{FeN}_4\text{-SD}_1$, $\text{FeN}_4\text{-SD}_2$ and $\text{FeN}_4\text{-SD}_3$.

Supplementary Fig. 6 XPS $\text{C}1\text{s}$ spectra of FeN_4 , $\text{FeN}_4\text{-SD}_1$, $\text{FeN}_4\text{-SD}_2$ and $\text{FeN}_4\text{-SD}_3$.

Supplementary Table 1. BET surface area and total pore volume of FeN_5 , $\text{FeN}_5\text{-SD}_1$, $\text{FeN}_5\text{-SD}_2$ and $\text{FeN}_5\text{-SD}_3$

Catalysts	BET specific surface area ($\text{m}^2 \text{g}^{-1}$)	Total pore volume ($\text{cm}^3 \text{g}^{-1}$)
FeN_5	43.78	0.1913
$\text{FeN}_5\text{-SD}_1$	46.86	0.3856
$\text{FeN}_5\text{-SD}_2$	55.63	0.4295

Editorial Note: The crystal structures in Fig. R5 and R6 were visualized using VESTA software (Momma, K. & Izumi, F. VESTA 3 for three-dimensional visualization of crystal, volumetric and morphology data, J. Appl. Crystallogr. 44, 1272–1276 (2011).)

Catalysts	BET specific surface area ($\text{m}^2 \text{g}^{-1}$)	Total pore volume ($\text{cm}^3 \text{g}^{-1}$)
$\text{FeN}_5\text{-SD}_3$	65.83	0.4335

Fig. R5 The optimal configurations of FeN_5 .

Fig. R6 The optimal configurations of $\text{FeN}_5\text{-SD}_2$.

Editorial Note: The crystal structures in Fig. R7 were visualized using VESTA software (Momma, K. & Izumi, F. VESTA 3 for three-dimensional visualization of crystal, volumetric and morphology data, J. Appl. Crystallogr. 44, 1272–1276 (2011).)

Fig. R7 The optimal configurations of FeN₅-SD₃.

Catalytic performance differences reveal remote interactions induced by high coordination shell defects

We have undertaken a comprehensive kinetic normalization analysis based on two independent metrics: the Fe-molar normalized rate constant ($K_{\text{per-mol Fe}}$) and the specific surface area-normalized rate constant ($K_{\text{per-area}}$) (**Fig. 3b**). This approach is crucial for distinguishing the intrinsic contribution of high coordination shell defects from the influence of other variables (such as Fe loading and specific surface area of different catalysts), thus validating our structure-activity correlation.

Specifically, $K_{\text{per-mol Fe}}$ evaluates the catalytic efficiency per mole of Fe. The metal contents in different FeN₅-SD_x catalysts were quantified using an inductively coupled plasma emission spectrometer (ICP-OES), with the specific values presented in **Supplementary Table 3**. Among the FeN₅-SD_x catalysts, FeN₅-SD₂ exhibited the highest $K_{\text{per-mol Fe}}$ ($3.40 \times 10^4 \text{ min}^{-1} \cdot \text{M}^{-1}$), outperforming FeN₅ ($1.30 \times 10^4 \text{ min}^{-1} \cdot \text{M}^{-1}$), FeN₅-SD₁ ($1.75 \times 10^4 \text{ min}^{-1} \cdot \text{M}^{-1}$) and FeN₅-SD₃ ($2.96 \times 10^4 \text{ min}^{-1} \cdot \text{M}^{-1}$). These results clearly demonstrate that the superior catalytic activity of FeN₅-SD₂ cannot be attributed solely to Fe content, effectively ruling out Fe loading as the primary driver of performance. In parallel, we normalized the catalytic activity by the specific surface area $K_{\text{per-area}}$, determined *via* Brunauer-Emmett-Teller (BET) analysis (**Supplementary Table 4**). Again, FeN₅-SD₂ showed the highest surface-area-normalized activity ($0.0072 \text{ min}^{-1}/(\text{m}^2 \cdot \text{g}^{-1})$), compared to FeN₅ (0.0046), FeN₅-SD₁ (0.0052), and FeN₅-SD₃ (0.0041). This further excludes the possibility that enhanced performance arises from improved surface dispersion or accessibility. Collectively, these normalized

results rule out the influence of other factors, including metal loading and specific surface area, on the observed catalytic enhancement. Therefore, the improved performance can be primarily ascribed to the electronic and geometric modulation induced by high coordination shell defects.

Fig. 3 (b) Normalized rate constant of BPA degradation by per mole Fe, per surface area.

Supplementary Table 3. Comparison of molar iron-normalized rate constants ($K_{\text{per-mol Fe}}$) for FeN₅-Based PMS systems.

Catalysts	$m_{\text{cat.}}$ (g·L ⁻¹)	ω_{metal} (%)	k_{obs} (min ⁻¹)	$K_{\text{per-mol Fe}}$ ($\times 10^4$ min ⁻¹ ·M ⁻¹)
FeN ₅	0.03	2.88	0.200	1.30
FeN ₅ -SD ₁	0.03	2.59	0.243	1.75
FeN₅-SD₂	0.03	2.21	0.403	3.40
FeN ₅ -SD ₃	0.03	1.69	0.268	2.96

Supplementary Table 4. Comparison of area-normalized rate constants ($K_{\text{per-area}}$) for FeN₅-based PMS systems with varying defect densities.

Catalysts	BET (m ² /g)	k_{obs} (min ⁻¹)	$K_{\text{per-area}}$ (min ⁻¹ /(m ² ·g ⁻¹))
FeN ₅	43.7834	0.200	0.0046
FeN ₅ -SD ₁	46.8609	0.243	0.0052
FeN₅-SD₂	55.6287	0.403	0.0072
FeN ₅ -SD ₃	65.8330	0.268	0.0041

To provide a more accurate and rigorous comparison of the catalytic performance,

we have adopted turnover frequency (TOF) as the key metric for evaluating the intrinsic activity of the active sites. The TOF, defined as the number of moles of pollutant degraded per mole of active site per unit time, offers a more reliable measure of catalytic efficiency, independent of external factors such as pollutant-specific kinetics or system-specific parameters. Unlike the normalized k_{obs} , TOF is directly related to the efficiency of the active sites and accounts for potential variations in active site exposure and utilization^{12, 13, 14, 15}.

The TOF is calculated as follows:

$$\text{TOF} [\text{min}^{-1}] = \frac{\text{moles of reactant converted}}{\text{moles of active sites} \times \text{reaction time}} = \frac{\Delta n(\text{pollutant}) \times M_{\text{metal}}}{m_0 \times \omega_{\text{metal}} \times t}$$

Where Δn (pollutant) is moles converted, M_{metal} is metal atomic weight, m_0 is catalyst mass, ω_{metal} is metal mass fraction, and t is reaction time. All parameters were derived from our kinetic experiments and site quantification analyses.

As shown in **Fig. 3c and Supplementary Table 5**, our $\text{FeN}_5\text{-SD}_2$ catalyst exhibits a significantly high TOF of 0.338 min^{-1} , outperforming other recently reported coordination-regulated SACs in PMS systems, as well as the FeN_5/PMS , $\text{FeN}_5\text{-SD}_1/\text{PMS}$, and $\text{FeN}_5\text{-SD}_3/\text{PMS}$ systems. Moreover, as the defect content increases, the TOF initially increases and then decreases, indicating that an optimal level of defects enhances catalytic activity.

Fig. 3 (c) Comparison of TOF values for pollutant degradation in SAC/PMS systems featuring coordination environment regulation.

Supplementary Table 5. Catalytic performances of SAC-based PMS systems regulated *via* coordination environment engineering.

Catalysts (g L ⁻¹)	$\Delta n(\text{pollutants})$ ($\times 10^{-6}$ mol)	ω_{metal} (%)	Reaction time (min)	TOF (min ⁻¹)	Ref.
Fe-N ₃ C ₁ (0.06)	SIZ (19.6)	0.75	30	0.081	16
Fe-N ₂ C ₂ (0.06)	SIZ (17)	0.84	30	0.063	16
Co-OCN (0.03)	APAP (13)	7.25	60	0.006	17
Fe-N-C (0.10)	BPA (100)	2.00	20	0.140	18
Co-N ₅ /CNT (0.03)	SMZ (40)	1.39	30	0.189	19
CNFe ₂ -0.6 (0.08)	SMZ (79)	16.64	8	0.042	20
Fe ₁ /CN (0.05)	4-CP (100)	11.2	10	0.100	21
FeSA-N/C-20 (0.15)	BPA (88)	0.88	20	0.187	22
FeSA-N/O-C (0.10)	BPA (66)	1.28	45	0.064	23
Co-N ₃ (0.10)	CBZ (42)	2.51	6	0.165	24
CoN ₃ O ₁ (0.10)	CIP (15)	0.51	20	0.087	25
Fe-N ₄ -C (0.10)	NPX (43)	1.50	10	0.161	26
SA-FeN ₅ (0.20)	APAP (30)	2.63	12	0.027	27
FeN₅ (0.03)	BPA (52)	1.86	20	0.260	This work
FeN₅-SD₁ (0.03)	BPA (60)	2.11	20	0.265	This work
FeN₅-SD₃ (0.03)	BPA (80)	2.24	20	0.333	This work
FeN₅-SD₂ (0.03)	BPA (62)	10.63	20	0.054	This work

ROS variations reveal remote interactions induced by high coordination shell defects

Systematic scavenger, EPR, and electrochemical analyses revealed that $^1\text{O}_2$ is the predominant reactive species in both FeN_5/PMS and $\text{FeN}_5\text{-SD}_x/\text{PMS}$ systems (Fig. 4), whereas radical species ($\cdot\text{OH}$ and $\text{SO}_4^{\cdot-}$), high-valence iron-oxo intermediates and direct electron-transfer pathway contribute negligibly (Supplementary Figs. 14, 19, and 22). The enhanced $^1\text{O}_2$ generation in $\text{FeN}_5\text{-SD}_x$ catalysts, especially $\text{FeN}_5\text{-SD}_2$, arises from remote coordination defects, which modulate the electronic environment and promote more efficient PMS activation (Fig. 4b). A volcano-type relationship between defect content and $^1\text{O}_2$ production was observed, mirroring the BPA degradation efficiency (Fig. 4f). These findings demonstrate that while the overall mechanism remains dominated by $^1\text{O}_2$ pathways, remote interactions in higher coordination shells fine-tune $^1\text{O}_2$ generation, thereby dictating the catalytic performance.

Fig. 4 Roles of ROS during Fenton-like reactions. (a) Comparison of degradation kinetics under various quenching conditions. (b) Decomposition rate of PMS in different systems. (c) Effect of pre-mixing in $\text{FeN}_5\text{-SD}_2/\text{PMS}$ system on BPA removal. (d) EPR spectra of $^1\text{O}_2$ captured by TEMP. (e) DPA signals in different systems. (f) Trends in the k_{obs} and $^1\text{O}_2$ contribution for various catalysts. (g) The contribution of ROS in different system.

Supplementary Fig. 14 BPA degradation in the FeN_5/PMS and FeN_5-SD_x/PMS system with different scavengers. Routine conditions: $[BPA] = 80 \mu M$, $[catalyst] = 0.03 g L^{-1}$, $[PMS] = 0.05 mM$, temperature = $25 \text{ }^\circ C$, without pH adjustment.

Supplementary Fig. 19 BPA removal in the FeN_5/PMS and FeN_5-SD_x/PMS system with 10 mM DMSO. Routine conditions: $[BPA] = 80 \mu M$, $[catalyst] = 0.03 g L^{-1}$, $[PMS] = 0.05 mM$, temperature = $25 \text{ }^\circ C$, without pH adjustment

[Figure Redacted]

Supplementary Fig. 22 (a) schematic diagram of the GOP reaction device. (b)-(e) the variation of current magnitude and BPA removal during the reaction in the FeN₅/PMS and FeN₅-SD_x/PMS system.

Thermodynamic and electron transfer differences reveal remote interactions induced by high coordination shell defects

To elucidate the influence of remote coordination environments on catalytic behavior, DFT calculations were performed to assess the electronic structure and PMS activation pathway of FeN₅ and defect-engineered FeN₅-SD_x sites. The total density of states (DOS) for FeN₅, FeN₅-SD₁, FeN₅-SD₂, and FeN₅-SD₃ showed significant electronic activity at the Fe center, with Fe-3d states near the Fermi level indicating enhanced electron transfer (**Fig. 5a**). The d-band center for FeN₅-SD_x is upshifted compared to FeN₅, following the order: FeN₅ (-2.385 eV) < FeN₅-SD₁ (-2.295 eV) < FeN₅-SD₂ (-2.251 eV) < FeN₅-SD₃ (-2.108 eV), suggesting enhanced electronic occupancy and reactivity due to remote defect modulation.

These remote defects also modulate PMS adsorption energetics. PMS adsorption calculations showed a clear preference for the terminal oxygen atom bonded to sulfur, which consistently exhibited the strongest binding across all FeN₅-SD_x systems (**Supplementary Figs. 28-31**). The adsorption energy (E_{ads}) decreases systematically with increasing defect density, consistent with the DOS analysis. FeN₅ shows weaker PMS adsorption (-1.72 eV), while FeN₅-SD₃ exhibits stronger binding (-1.91 eV). However, the observed k_{obs} follows a volcano trend, with FeN₅-SD₂ showing the highest Fenton-like activity due to moderate adsorption strength (**Fig. 5b**). Excessive binding on FeN₅-SD₃ inhibits catalytic turnover. These findings demonstrate that controlled

Editorial Note: The crystal structures in Fig. 5c were visualized using VESTA software (Momma, K. & Izumi, F. VESTA 3 for three-dimensional visualization of crystal, volumetric and morphology data, J. Appl. Crystallogr. 44, 1272–1276 (2011).)

modulation of higher coordination shells optimizes PMS adsorption and enhances Fenton reaction kinetics. Charge density and Bader charge calculations reveal significant electron transfer from Fe to PMS. FeN₅-SD₂ shows the highest electron transfer (0.811 e), followed by FeN₅ (0.795 e), FeN₅-SD₁ (0.798 e), and FeN₅-SD₃ (0.807 e) (**Fig. 5c**). Electrochemical impedance spectroscopy further supports these trends, as FeN₅-SD₂ shows the smallest semicircle and a sharp current increase upon PMS injection (**Supplementary Fig. 33**).

DFT calculations show that the rate-determining step in ¹O₂ generation is the O-O dissociation (from *PMS to *OH + *SO₄), with the energy barrier reduced from 0.85 eV to 0.73 eV (**Fig. 5d-f and Supplementary Fig. 34**). FeN₅-SD₂ also exhibits the longest O-O bond (1.474 Å) (**Fig. 5g**), lowering the cleavage barrier and enhancing ¹O₂ production.

These results confirm that remote interactions, enabled by higher-shell defect engineering, modulate the Fe electronic environment and PMS binding energetics, thereby enhancing charge transfer and ¹O₂ generation.

Fig. 5 The mechanistic of high coordination shell defects in regulating FeN₅ active

Editorial Note: The crystal structures in Supplementary Fig. 28 and 29 were visualized using VESTA software (Momma, K. & Izumi, F. VESTA 3 for three-dimensional visualization of crystal, volumetric and morphology data, J. Appl. Crystallogr. 44, 1272–1276 (2011).)

sites. (a) Density of states of FeN₅, FeN₅-SD₁, FeN₅-SD₂ and FeN₅-SD₃. (b) Correlation among the oxidation capacity, coupling strength to PMS, and d-band structure of FeN₅, FeN₅-SD₁, FeN₅-SD₂ and FeN₅-SD₃. (c) Electron density difference diagrams and Bader charges for PMS adsorbed on FeN₅, FeN₅-SD₁, FeN₅-SD₂ and FeN₅-SD₃, with blue and yellow indicating electron depletion and accumulation, respectively. (d) Calculated potential energy diagrams for various bond-cleavage pathways during PMS decomposition over FeN₅-SD₂. (e) Calculated potential energy diagrams for the ¹O₂ formation pathway. (f) Schematic diagram of ¹O₂ generation process. (g) Length of the O–O bonds after adsorption of PMS adsorbed on different catalysts.

Supplementary Fig. 28 Comparative adsorption energies of PMS oxygen atoms on FeN₅

Supplementary Fig. 29 Comparative adsorption energies of PMS oxygen atoms on

Editorial Note: The crystal structures in Supplementary Fig. 30 and 31 were visualized using VESTA software (Momma, K. & Izumi, F. VESTA 3 for three-dimensional visualization of crystal, volumetric and morphology data, J. Appl. Crystallogr. 44, 1272–1276 (2011).)

FeN₅-SD₁

Supplementary Fig. 30 Comparative adsorption energies of PMS oxygen atoms on FeN₅-SD₂

Supplementary Fig. 31 Comparative adsorption energies of PMS oxygen atoms on FeN₅-SD₃

Supplementary Fig. 33 (a) electrochemical impedance analyses spectra and (b) chronoamperometry analysis of FeN₅, FeN₅-SD₁, FeN₅-SD₂, and FeN₅-SD₃.

Supplementary Fig. 34 (a-c) Calculated potential energy diagrams for various bond-cleavage pathways during PMS decomposition over different catalysts.

2. The authors compared the catalytic performance of various materials using normalized k_{obs} while only accounting for differences in catalyst and PMS mass. However, the reliability and validity of the normalization method used were not clearly established. It remained unclear whether such a metric adequately reflected the intrinsic activity of the active sites. Given the single-atom characteristic of the catalysts and the potential variability in active site exposure or utilization efficiency, the choice of normalization method could have significantly influenced the interpretation of relative activity.

Response: We sincerely thank the reviewer for their constructive feedback and for raising important concerns about the normalization method used to compare catalytic performance. We fully acknowledge the potential limitations of using normalized k_{obs} solely based on catalyst and PMS mass, as it may not fully account for the intrinsic activity of the active sites, particularly considering the single-atom nature of the catalysts in this study.

To address this issue and provide a more accurate and rigorous comparison of the

catalytic performance, we have adopted turnover frequency (TOF) as the key metric for evaluating the intrinsic activity of the active sites. The TOF, defined as the number of moles of pollutant degraded per mole of active site per unit time, offers a more reliable measure of catalytic efficiency, independent of external factors such as pollutant-specific kinetics or system-specific parameters. Unlike the normalized k_{obs} , TOF is directly related to the efficiency of the active sites and accounts for potential variations in active site exposure and utilization^{12, 13, 14, 15}.

The TOF is calculated as follows:

$$\text{TOF} [\text{min}^{-1}] = \frac{\text{moles of reactant converted}}{\text{moles of active sites} \times \text{reaction time}} = \frac{\Delta n(\text{pollutant}) \times M_{\text{metal}}}{m_0 \times \omega_{\text{metal}} \times t}$$

Where Δn (pollutant) is moles converted, M_{metal} is metal atomic weight, m_0 is catalyst mass, ω_{metal} is metal mass fraction, and t is reaction time. All parameters were derived from our kinetic experiments and site quantification analyses.

This approach ensures that the comparison of catalytic performance reflects the intrinsic activity of the active sites rather than being influenced by factors such as catalyst mass or system configuration. By using TOF, we believe we can provide a more accurate and fair comparison of the catalytic efficiencies of the various materials tested. As shown in **Fig. 3 c and Supplementary Table 5**, our FeN₅-SD₂ catalyst exhibits a notably high TOF value of 0.338 min⁻¹, outperforming other recently reported coordination-regulated SACs in PMS systems.

We have revised the manuscript to include TOF as the primary metric for evaluating catalytic performance and have updated the relevant sections to explain this new approach. We trust that this change addresses the reviewer's concerns and improves the clarity and robustness of our comparative analysis.

Action 2: We have added relevant statements in the revised manuscript (Fig. 3c) and Supplementary Table 5 has been added in the supporting information.

“The turnover frequency (TOF) was employed to quantitatively evaluate the intrinsic reactivity against previously reported SAC-based PMS activation systems. As shown in Fig. 3c and Supplementary Table 5, FeN₅-SD₂ catalyst exhibits a significantly high TOF of 0.338 min⁻¹. This performance surpasses other recently reported coordination-regulated SACs in PMS systems.”

Fig. 3 (c) Comparison of TOF values for pollutant degradation in SAC/PMS systems featuring coordination environment regulation.

Supplementary Table 5. Catalytic performances of SAC-based PMS systems regulated *via* coordination environment engineering.

Catalysts (g L ⁻¹)	$\Delta n(\text{pollutants})$ ($\times 10^{-6}$ mol)	ω_{metal} (%)	Reaction time (min)	TOF (min ⁻¹)	Ref.
Fe-N ₃ C ₁ (0.06)	SIZ (19.6)	0.75	30	0.081	16
Fe-N ₂ C ₂ (0.06)	SIZ (17)	0.84	30	0.063	16
Co-OCN (0.03)	APAP (13)	7.25	60	0.006	17
Fe-N-C (0.10)	BPA (100)	2.00	20	0.140	18
Co-N ₅ /CNT (0.03)	SMZ (40)	1.39	30	0.189	19
CNFe ₂ -0.6 (0.08)	SMZ (79)	16.64	8	0.042	20
Fe ₁ /CN (0.05)	4-CP (100)	11.2	10	0.100	21
FeSA-N/C-20 (0.15)	BPA (88)	0.88	20	0.187	22
FeSA-N/O-C (0.10)	BPA (66)	1.28	45	0.064	23
Co-N ₃ (0.10)	CBZ (42)	2.51	6	0.165	24
CoN ₃ O ₁ (0.10)	CIP (15)	0.51	20	0.087	25

Catalysts (g L ⁻¹)	$\Delta n(\text{pollutants})$ ($\times 10^{-6}$ mol)	ω_{metal} (%)	Reaction time (min)	TOF (min ⁻¹)	Ref.
Fe-N ₄ -C (0.10)	NPX (43)	1.50	10	0.161	²⁶
SA-FeN ₅ (0.20)	APAP (30)	2.63	12	0.027	²⁷
FeN₅ (0.03)	BPA (52)	1.86	20	0.260	This work
FeN₅-SD₁ (0.03)	BPA (60)	2.11	20	0.265	This work
FeN₅-SD₃ (0.03)	BPA (80)	2.24	20	0.333	This work
FeN₅-SD₂ (0.03)	BPA (62)	10.63	20	0.054	This work

3. **(3.1)** Regarding the mechanism of reactive species generation, the DFT calculations indicated that the highest energy barrier occurred in step I (*PMS \rightarrow *OH + *SO₄), which was identified as the rate-determining step. This reaction pathway, involving the cleavage of the O-O bond in PMS to generate hydroxyl and sulfate radicals, was highly consistent with mechanisms widely reported in previous studies on Fenton-like catalysis (e.g. Nat Commun 16, 2402 (2025)). As such, it did not appear to offer fundamentally new mechanistic insights or demonstrate a significant difference from conventional PMS activation behavior. **(3.2)** Moreover, Figs. 4b and 4f showed that the FeN₅-SD₂ catalyst exhibited the highest PMS decomposition efficiency and the greatest yield of ¹O₂ among the catalysts examined. However, the DFT results presented in the manuscript did not clearly support or explain this observation. In particular, there was a lack of theoretical evidence linking the structural modulation of FeN₅-SD₂ to enhanced ¹O₂ generation, such as an energetically favorable non-radical pathway or lower activation barrier for ¹O₂ evolution. This discrepancy between experimental findings and computational predictions raised questions about the proposed mechanism and its completeness.

Response 3.1:

We sincerely thank the reviewer for the constructive comments and for providing a highly valuable reference (Nat. Commun. 16, 2402 (2025)). Regarding the similarity of the initiation step for PMS activation (step I: *PMS \rightarrow *OH + *SO₄) between previous work and ours, we would like to clarify that this observation is not surprising, as the common activation of PMS typically starts with the cleavage of O-O, S-O, and O-H

bonds within the PMS molecule. All studies on PMS activation have followed one of the three established bond cleavage pathways for initiation, and our work likewise adheres to one of these pathways. However, it is important to note that both the initial bond cleavage and the subsequent evolution of reactive species in PMS activation are highly dependent on the active site properties of the catalyst. This is why many studies aim to explore the intricate relationship between the active site environment and the catalytic performance of SACs in PMS activation. In our study, while the initial bond cleavage step (O-O bond cleavage in PMS) may be similar to the referenced work (Nat. Commun. 16, 2402 (2025)), the downstream ROS evolution differs significantly. For example, in contrast to the catalysts in the referenced study (Nat. Commun. 16, 2402 (2025)), which primarily operate through high-valent metal-oxo species, our FeN₅-SD₂/PMS system, with its unique defect coordination environment, promotes ¹O₂ as the dominant ROS. This strategy integrates axial FeN₅ configurations in the primary coordination shell, complemented by planar defective coordination in higher coordination shells (FeN₅-SD_x). The remote modulation of FeN₅ sites through high coordination shell defects significantly optimize the electronic properties of Fe centers, thereby enhancing their efficacy in selectively activating PMS. To our knowledge, the impact of the higher coordination defect environment of FeN₅ SACs on PMS activation, including a comprehensive analysis of the thermodynamic and kinetic mechanisms and the evolution of ¹O₂, has not been previously explored. This constitutes the key innovation of our work, where we design model FeN₅ SACs capable of modulating higher coordination shells to investigate their effects on PMS activation.

Scavenger experiments, EPR spectroscopy, and electrochemical analyses confirm that ¹O₂ is the predominant reactive species in both FeN₅/PMS and FeN₅-SD_x/PMS systems, with minimal contributions from radicals and high-valent iron-oxo intermediates (**Fig. 4**). The enhanced ¹O₂ generation in FeN₅-SD_x catalysts, particularly FeN₅-SD₂, stems from remote coordination defects that optimize PMS activation. A volcano-type relationship between defect content and ¹O₂ production correlates with BPA degradation efficiency, underscoring the critical role of higher coordination shells in fine-tuning catalytic performance (**Fig. 4f**). In addition to the experimental data, we have conducted comprehensive theoretical calculations on each activation step. We first calculated the adsorption configurations and energies of three oxygen atoms in the PMS molecule on FeN₅, FeN₅-SD₁, FeN₅-SD₂, and FeN₅-SD₃. The results show that, in all catalysts, the terminal oxygen atom directly bonded to the sulfur center exhibits the strongest adsorption affinity, making it the most thermodynamically favorable site for

PMS activation (**Supplementary Figs. 28-31**). The adsorption energy (E_{ads}) decreases systematically with increasing defect density. FeN₅ shows weaker PMS adsorption (−1.72 eV), while FeN₅-SD₃ exhibits stronger binding (−1.91 eV). However, the observed k_{obs} follows a volcano trend, with FeN₅-SD₂ showing the highest Fenton-like activity due to moderate adsorption strength (**Fig. 5b**). Excessive binding on FeN₅-SD₃ inhibits catalytic turnover. These findings demonstrate that controlled modulation of higher coordination shells optimizes PMS adsorption and enhances Fenton reaction kinetics.

Additionally, we calculated initiation step and compared the energy barriers associated with the cleavage of the three key bonds in PMS (O-O, S-O, and O-H) across both FeN₅ and FeN₅-SD_x. As shown in **Fig. 5d and Supplementary Fig. 34**, the O-O bond consistently exhibited the lowest energy barrier, confirming that it is the most readily cleaved and represents the preferred initiation step of PMS activation. This result verifies that O-O bond scission is energetically favorable in all catalysts studied and serves as the initiation step for PMS activation. Optimized free energy diagrams depict the sequential steps of ¹O₂ generation (**Fig. 5f**): PMS → *PMS (I) → *OH + *SO₄ (II) → *OH + H₂SO₄ (III) → *OH + *OH (IV) → *O (V) → *O + *O (VI) → ¹O₂. In this pathway, PMS undergoes O–O bond cleavage, forming into *OH and *SO₄ fragments. The *OH adsorbing onto FeN₅, while *SO₄ exothermically convert to H₂SO₄. Subsequently, *OH transforms into *O, which recombines into ¹O₂. The highest energy barrier occurs at step I (*PMS → II *OH + *SO₄), suggesting it as the rate-determining step (RDS) in ¹O₂ generation. Introducing moderate defects in the higher coordination shell lowers this barrier from 0.85 to 0.73 eV, thereby facilitating ¹O₂ formation during PMS activation. Furthermore, O–O bond length ($l_{\text{O-O}}$) analysis (**Fig. 5g**) reveals FeN₅-SD₂ exhibits the longest bond (1.474 Å), indicating that moderate defects extend the O–O bond, lowering its cleavage energy barrier and enhancing ¹O₂ evolution kinetics.

Editorial Note: The crystal structures in Supplementary Fig. 28 were visualized using VESTA software (Momma, K. & Izumi, F. VESTA 3 for three-dimensional visualization of crystal, volumetric and morphology data, J. Appl. Crystallogr. 44, 1272–1276 (2011).)

Fig. 4 Roles of ROS during Fenton-like reactions. (a) Comparison of degradation kinetics under various quenching conditions. (b) Decomposition rate of PMS in different systems. (c) Effect of premixing in FeN₅-SD₂/PMS system on BPA removal. (d) EPR spectra of ¹O₂ captured by TEMP. (e) DPA signals in different systems. (f) Trends in the *k*_{obs} and ¹O₂ contribution for various catalysts. (g) The contribution of ROS in different system.

Supplementary Fig. 28 Comparative adsorption energies of PMS oxygen atoms on FeN₅

Editorial Note: The crystal structures in Supplementary Fig. 29 and 30 were visualized using VESTA software (Momma, K. & Izumi, F. VESTA 3 for three-dimensional visualization of crystal, volumetric and morphology data, J. Appl. Crystallogr. 44, 1272–1276 (2011).)

Supplementary Fig. 29 Comparative adsorption energies of PMS oxygen atoms on FeN₅-SD₁

Supplementary Fig. 30 Comparative adsorption energies of PMS oxygen atoms on FeN₅-SD₂

Editorial Note: The crystal structures in Supplementary Fig. 31 were visualized using VESTA software (Momma, K. & Izumi, F. VESTA 3 for three-dimensional visualization of crystal, volumetric and morphology data, J. Appl. Crystallogr. 44, 1272–1276 (2011)).

Supplementary Fig. 31 Comparative adsorption energies of PMS oxygen atoms on FeN₅-SD₃

Fig. 5 (b) Correlation among the oxidation capacity, coupling strength to PMS, and d-band structure of FeN₅, FeN₅-SD₁, FeN₅-SD₂ and FeN₅-SD₃.

Supplementary Fig. 34 (a-c) Calculated potential energy diagrams for various bond-cleavage pathways during PMS decomposition over different catalysts.

Fig. 5 (d) Calculated potential energy diagrams for various bond-cleavage pathways during PMS decomposition on FeN₅-SD₂. (e) Calculated potential energy diagrams for the ¹O₂ formation pathway. (f) Schematic diagram of ¹O₂ generation process. (g) Length of the O–O bonds after adsorption of PMS adsorbed on different catalysts.

Action 3.1: We have the added relevant statements in the revised manuscript (Fig. 3d) and Supplementary Fig. 34 has been added in the supporting information.

“To further understand PMS activation in FeN₅-SD_x, we calculated the energy barriers along the ¹O₂ evolution pathway (Fig. 5d-e). We first compared the energy barriers for cleaving three key PMS bonds (O-O, S-O, and O-H) on FeN₅ and FeN₅-SD_x. As shown in Fig. 5d and Supplementary Fig. 34, the O–O bond consistently has the lowest barrier, indicating it is the easiest to break and thus the preferred initiation step.”

Fig. 5 (d) Calculated potential energy diagrams for various bond-cleavage pathways during PMS decomposition over FeN₅-SD₂.

Supplementary Fig. 34 (a-c) Calculated potential energy diagrams for various bond-cleavage pathways during PMS decomposition over different catalysts.

Response 3.2:

We sincerely thank the reviewer for raising this important point. We acknowledge that the computational results presented in the manuscript may have appeared inconsistent with the experimental data due to an unfortunate labeling error in **Fig. 5d-e (original version)**. Specifically, the DFT curves for FeN₅-SD₂ and FeN₅-SD₃ were mislabeled, which led to confusion regarding the correlation between theoretical predictions and observed catalytic performance.

Upon re-examining the data, we confirm that the red line in **Fig. 5d-e (original version)** corresponds to the FeN₅-SD₂ system, while the blue line represents FeN₅-SD₃. This correction aligns with the trend observed in the previous version of **Fig. 5b**, where adsorption energies decrease progressively with increasing defect density, corresponding to the PMS→*PMS (I) step, thereby supporting the correct assignment of FeN₅-SD₂.

After correcting the labels, the computational results align well with the experimental observations. The FeN₅-SD₂ system exhibits the longest O-O bond length among all configurations (as shown in **Fig. 5g**), which facilitates bond activation. Importantly, the O-O bond cleavage, identified as the rate-determining step for PMS activation, has the lowest activation barrier in the FeN₅-SD₂ system (0.73 eV), compared to FeN₅ (0.79 eV), FeN₅-SD₁ (0.85 eV), and FeN₅-SD₃ (0.84 eV). This lower barrier indicates enhanced kinetic favorability for PMS activation and correlates directly with the experimentally observed superior performance of FeN₅-SD₂ in both PMS decomposition and ¹O₂ production (**Fig. 4b** and **Fig. 4d-g**).

We sincerely apologize for the oversight and appreciate the reviewer's careful reading, which allowed us to correct this critical error. With the corrected computational data, the mechanistic explanation for enhanced ¹O₂ generation in the FeN₅-SD₂ system is now well-supported, and the consistency between theoretical predictions and experimental findings is clearly established.

Fig. 5 (b) Correlation among the oxidation capacity, coupling strength to PMS, and d-band structure of FeN_5 , $\text{FeN}_5\text{-SD}_1$, $\text{FeN}_5\text{-SD}_2$ and $\text{FeN}_5\text{-SD}_3$.

Fig. 5 (g) Length of the O–O bonds after adsorption of PMS adsorbed on different catalysts.

Fig. 4 (b) Decomposition rate of PMS in different systems.

Fig. 4 (d) EPR spectra of $^1\text{O}_2$ captured by TEMP. (e) DPA signals in different systems. (f) Trends in the k_{obs} and $^1\text{O}_2$ contribution for various catalysts. (g) The contribution of ROS in different system.

Action 3.2: The labels in the original Fig. 5d-e have been corrected. In the revised manuscript, the updated version of this figure is now placed as Fig. 5e.

[Figure Redacted]

Fig. 5 (original submission) (d) Calculated potential energy diagrams for the $^1\text{O}_2$ formation pathway and (e) enlarged diagrams of the rate-determining step for various catalysts.

Fig. 5 (revised version) (e) Calculated potential energy diagrams for the $^1\text{O}_2$ formation pathway and enlarged diagrams of the rate-determining step for various catalysts.

4. **(4.1)** The section concerning the degradation pathway of BPA and the associated toxicity analysis appeared to be relatively superficial and lacked depth in mechanistic insight. The current analysis did not convincingly emphasize the unique reactivity or selectivity resulting from the high shell coordination defects in the $\text{FeN}_5\text{-SD}_2$ catalyst. The proposed degradation pathways appeared to follow the general trends observed in previous studies and did not clearly show how the engineered coordination environment affected the formation or inhibition of specific intermediates. **(4.2)** Furthermore, the identification of transformation products remained largely qualitative. The authors did not provide quantitative analyses of key intermediates, such as their concentrations or profiles over time. Without these quantitative data, subsequent toxicity assessments lost much of their persuasive power and did not accurately reflect the true environmental risks.

Response: We sincerely appreciate the reviewer's insightful comments regarding the degradation pathway and toxicity analysis. We fully agree that our original discussion lacked sufficient mechanistic detail and quantitative assessment. In response, we have significantly expanded this section to provide a more rigorous and comprehensive evaluation of the BPA degradation process and the associated environmental risks.

Response 4.1: Mechanistic insights into BPA degradation

Impact of high coordination shell defects on ROS formation and selectivity:

The key to understanding BPA degradation in our system lies in the unique reactive oxygen species (ROS), which is governed by the coordination environment of the $\text{FeN}_5\text{-SD}_x$ catalysts. As confirmed by EPR spectroscopy and scavenger experiments (Fig. 4a-e), $^1\text{O}_2$ is the dominant ROS in the $\text{FeN}_5\text{-SD}_x/\text{PMS}$ systems. Importantly, the production

of $^1\text{O}_2$ correlates directly with the observed BPA degradation efficiency across the series (**Fig. 4f-g**), indicating a structure-reactivity relationship modulated by the higher coordination shell.

Our system operates *via* a non-radical pathway, enabled by precise defect engineering in the higher coordination shell. DFT calculations reveal that increasing defect density alters Fe 3d orbital occupancy, facilitates charge transfer, and elongates the O-O bond in PMS, thereby lowering the activation barrier for $^1\text{O}_2$ generation (**Fig. 5**). This defect-driven modulation leads to selective $^1\text{O}_2$ production, offering a mechanistically distinct and energetically favorable pathway for PMS activation.

Evidence for selective substrate degradation *via* $^1\text{O}_2$:

Owing to the low-lying and empty π -antibonding orbital, $^1\text{O}_2$ has high ability to obtain a pair of electrons³⁴. As a result, $^1\text{O}_2$ can act as an electrophile agent to attack electron-rich organics³⁵. To further probe the selectivity imparted by $^1\text{O}_2$ -dominated oxidation, we evaluated the degradation of pollutants with varied electronic properties (**Fig. 3f**). Substrates containing electron-donating groups (e.g., 4-chlorophenol, 4-CP) showed significantly higher degradation rates, whereas electron-withdrawing group-bearing compounds (e.g., atrazine, ATZ) exhibited minimal reactivity. This clear preference highlights a non-radical, electrophilic oxidation mechanism. Unlike non-selective radicals, $^1\text{O}_2$ exhibits high reactivity toward electron-rich functional groups.

Mechanistic implications for BPA degradation pathway:

BPA degradation mechanisms differ substantially depending on the nature of the reactive species involved. Radical pathways ($\cdot\text{OH}$ or $\text{SO}_4^{\cdot-}$) are highly reactive and non-selective species that degrade BPA *via* electron transfer and aromatic ring cleavage pathways. These radicals tend to indiscriminately attack multiple sites on the molecule, leading to complex and potentially toxic intermediates³⁶. In electron transfer pathway (ETP)-dominated systems, BPA undergoes direct oxidation by transferring electrons to the catalyst surface. This process often induces polymerization, resulting in the formation of oligomeric byproducts^{37,38}. In contrast, $^1\text{O}_2$ acts as a selective electrophile, preferentially attacking electron-rich regions of BPA. The degradation occurs through three main routes^{39, 40, 41}: (i) electrophilic addition to electron-rich aromatic rings, especially at ortho- and para-positions relative to hydroxyl groups, forming epoxides or peroxides; (ii) oxidative cleavage of the aromatic rings, yielding quinones, carboxylic acids, and other oxygenated products; and (iii) oxidation of the aliphatic bridging group between the phenol rings, which are particularly promoted under conjugated structures. In some studies, p-benzoquinone is found to be the primary product, which can be

further degraded by $^1\text{O}_2$ into smaller molecules⁴². This selectivity highlights the distinct electrophilic nature of $^1\text{O}_2$ compared to non-selective radicals.

The proposed degradation mechanism is consistent with known $^1\text{O}_2$ -mediated oxidation processes. As illustrated in Fig. 38, two primary routes were proposed based on the identified intermediates. In Pathway 1, BPA undergoes hydroxylation to generate hydroxylated intermediates (e.g., P1), which are then further oxidized to P2-P4 bearing ketone and carboxyl groups, accompanied by aromatic ring cleavage. This may be due to that $^1\text{O}_2$ were predominantly present in the reaction media and have high reactivity toward hydroxylated intermediates. In Pathway 2, β -scission of the quaternary carbon linking the two phenyl rings yields isopropenylphenol (P6), which subsequently transforms into 2-phenyl-1-propene (P7) and p-hydroxyacetophenone (P8). These intermediates are then oxidized to form open-chain compounds such as hexa-1,5-diene-3,4-diol (P9), phenol (P10), and p-benzoquinone (P11). Eventually, the aromatic structures are mineralized into CO_2 and H_2O through sequential oxidative steps.

These findings highlight that $^1\text{O}_2$ -driven degradation is not only more selective but also yields simpler and potentially less toxic products, distinguishing it from both radical and ETP. To the best of our knowledge, this is the first study to demonstrate that deliberately engineered high coordination shell defects in FeN_5 SACs can also promote $^1\text{O}_2$ generation via PMS activation, thereby advancing both the mechanistic understanding and environmental applicability of PMS-based AOPs.

Fig. 4 Roles of ROS during Fenton-like reactions. (a) Comparison of degradation

Editorial Note: The crystal structures in Fig. 5c were visualized using VESTA software (Momma, K. & Izumi, F. VESTA 3 for three-dimensional visualization of crystal, volumetric and morphology data, J. Appl. Crystallogr. 44, 1272–1276 (2011).)

kinetics under various quenching conditions. (b) Decomposition rate of PMS in different systems. (c) Effect of premixing in FeN₅-SD₂/PMS system on BPA removal. (d) EPR spectra of ¹O₂ captured by TEMP. (e) DPA signals in different systems. (f) Trends in the k_{obs} and ¹O₂ contribution for various catalysts. (g) The contribution of ROS in different system.

Fig. 5 The mechanistic of high coordination shell defects in regulating FeN₅ active sites. (a) Density of states of FeN₅, FeN₅-SD₁, FeN₅-SD₂ and FeN₅-SD₃. (b) Correlation among the oxidation capacity, coupling strength to PMS, and d-band structure of FeN₅, FeN₅-SD₁, FeN₅-SD₂ and FeN₅-SD₃. (c) Electron density difference diagrams and Bader charges for PMS adsorbed on FeN₅, FeN₅-SD₁, FeN₅-SD₂ and FeN₅-SD₃, with blue and yellow indicating electron depletion and accumulation, respectively. (d) Calculated potential energy diagrams for various bond-cleavage pathways during PMS decomposition on FeN₅-SD₂. (e) Calculated potential energy diagrams for the ¹O₂ formation pathway. (f) Schematic diagram of ¹O₂ generation process. (g) Length of the O–O bonds after adsorption of PMS adsorbed on different catalysts.

Fig. 3 (f) Different pollutants removal in FeN₅-SD₂/PMS system. Routine conditions: [pollutants] = 80 μM, [catalyst] = 0.03 g L⁻¹, [PMS] = 0.05 mM, temperature = 25 °C, without pH adjustment.

Supplementary Fig. 38 Possible degradation pathway diagram of BPA.

Action 4.1: We have the added relevant statements in the revised manuscript and Supplementary Fig. 38 has been added in the supporting information.

“As shown in Supplementary Fig. 38, two primary pathways were identified based on the detected intermediates. In Pathway 1, BPA undergoes hydroxylation to form hydroxylated intermediates (e.g., P1), which are further oxidized to P2-P4, containing ketone and carboxyl groups, accompanied by aromatic ring cleavage. This pathway is

likely driven by the predominant presence of ¹O₂ in the reaction medium, which exhibits high reactivity towards hydroxylated intermediates. In Pathway 2, β-scission of the quaternary carbon connecting the two phenyl rings leads to the formation of isopropenylphenol (P6), which subsequently converts into 2-phenyl-1-propene (P7) and p-hydroxyacetophenone (P8). These intermediates are further oxidized to form open-chain compounds, including hexa-1,5-diene-3,4-diol (P9), phenol (P10), and p-benzoquinone (P11). Ultimately, the aromatic structures are mineralized into CO₂ and H₂O through sequential oxidation steps.”

Response 4.2: Quantitative profiling of intermediates and additional toxicity evaluation

Beyond mechanistic understanding, we performed quantitative profiling of reaction intermediates and systematically evaluated their associated toxicity.

Specifically, beyond the initial T.E.S.T. software predictions, we have now included ECOSAR-based assessments to estimate the acute and chronic toxicity of BPA and its degradation products. The results include predicted LC₅₀ (lethal concentration for 50%), EC₅₀ (effective concentration for 50%), and chronic toxicity (ChV) values for aquatic organisms such as fish, Daphnia, and green algae. These values were classified according to the Globally Harmonized System of Classification and Labelling of Chemicals (GHS) into four categories: very toxic (≤1 mg/L), toxic (1-10 mg/L), harmful (10-100 mg/L), and not harmful (>100 mg/L). As shown in **Supplementary Table 7**, both the acute and chronic toxicity values of the oxidized products significantly decreased as degradation progressed. Notably, advanced oxidation products such as P9, P10, and P11 were predicted to be “not harmful”, underscoring the detoxification capacity of our FeN₅-SD_x system.

To support these predictions, we further performed quantitative gas chromatography-mass spectrometry (GC-MS) analysis of key degradation intermediates throughout the reaction (**Supplementary Table 6, Supplementary Fig. 36 and Fig. 37**). For example, P3 was detected at 3.75 ng/mL, while P6, P7, and P8 were present at 0.14, 0.06, and 0.59 ng/mL, respectively. Importantly, downstream products P9 and P10 were found at relatively higher concentrations of 1.11 ng/mL and 293.13 ng/mL, respectively, but all remained at ng/mL levels, far below the toxicological thresholds classified by GHS (in mg/L range), and were consistently categorized as “not harmful”. These results provide strong quantitative evidence that our system not only degrades BPA effectively but also drives the reaction toward low-toxicity end products. Concentrations of additional intermediates are provided in **Supplementary Table 6**. We regret that due to the

unavailability of suitable standards, quantitative determination of P1, P2, P4, and P9 was not conducted.

Beyond computational predictions, we carried out experimental toxicity validation using zebrafish and microbial viability assays. Zebrafish exposed to BPA exhibited early mortality and behavioral abnormalities over a 120-hour period. In contrast, those cultured in the post-reaction BPA solution from the FeN₅-SD₂/PMS system showed no abnormalities or fatalities (**Supplementary Fig. 40**), providing compelling evidence for the biocompatibility of the treated water. Finally, we conducted flow cytometry to assess the cytotoxic effects of treated and untreated solutions on microbial populations (**Fig. 6c-d**). After 40 minutes of treatment, the proportion of live cells increased from 35.1% to 70.6%, while dead cells decreased from 3.92% to 0.73%. These results highlight the robust detoxification capability of the FeN₅-SD₂/PMS system.

Action 4.2: We have added the relevant statements in the revised manuscript (Fig. 6a-d), and Supplementary Text 7, Supplementary Figs. 35-37 and 39-40, as well as Supplementary Tables 5 and 6, have been included in the Supporting Information

“The proposed BPA degradation pathways in the FeN₅-SD₂/PMS system were deduced by combining liquid chromatography-mass spectrometry (LC-MS) results (Supplementary Fig. 35) with gas chromatography-mass spectrometry (GC-MS) spectrometry data (Supplementary Figs. 36-37 and Supplementary Table 6).”

The detailed quantitative methods have been provided in *Supporting Information*.

“In order to determine the degradation products of BPA, a 20 mL of the post-degradation solution (initial BPA concentrations 50 ppm) was acidified to pH 3.0 with HCl, mixed with 1 g of NaCl, and transferred to a separatory funnel. Subsequently, 5 mL of dichloromethane was added to the funnel, and the sample was shaken vigorously. After the solution was extracted for 5 min, the organic phase was transferred to a 15 mL centrifuge tube. The sample was extracted repeated once more and the organic phases combined. Finally, the extract was concentrated and quantified to 1 mL by acetone after drying with 1 g Na₂SO₄.

The degradation products of BPA were determined by GC/MS (QP2020 NX, Shimadzu, Japan) combined with a DB-17MS (20 × 0.18 mm, 0.18 μm) quartz capillary column and an electron impact (EI) detector (70 eV). The carrier gas was helium at a flow rate of 1 mL/min and the injection volume was 1 μL in splitless mode. The temperature program was as follows: the initial temperature was 40 °C and held for 3

min; then it increased at 15 °C/min to 320 °C where it was held for 6 min. The temperatures of the injection port, interface and the ion source were 300, 300 and 230 °C, respectively.”

Supplementary Fig. 35 ESI mass spectra of BPA during degradation in FeN₅-SD₂/PMS system.

Supplementary Fig. 36. Calibration curves of the intermediates during BPA degradation.

Supplementary Fig. 37. GC/MS chromatograms of the intermediates in the standard solution and the sample from BPA degradation.

Supplementary Table 6. Quantitative analysis of degradation intermediates by GC-MS.

Product	Chemical name	Selected Quantitative ions (m/z)	Retention time (min)	Peak area	Concentration (ng/mL)
P3	4-(4-hydroxybenzyl)cyclohexa-3,5-diene-1,2-dione	214	21.20	806	3.75
P5	4-(prop-1-en-2-yl)benzene-1,2-diol	150	12.95	ND	ND
P6	4-isopropylphenol	136	10.12	238	0.14
P7	prop-1-en-2-ylbenzene	116	6.87	396	0.06
P8	1-(4-hydroxyphenyl)ethan-1-one	136	13.12	740	0.59
P10	Phenol	94	7.30	11899	1.11
P11	benzoquinone	108	7.32	242230	293.13

“To assess the real-world applicability of the catalytic system, we first predicted the toxicity of BPA and its intermediates using the Ecological Structure-Activity Relationships (ECOSAR) and Toxicity Estimation Software Tool (T.E.S.T.) systems (Fig. 6a-b, Supplementary Fig. 39, and Supplementary Table 7). The half-lethal concentration (LC_{50}) values for BPA in Fathead minnow and *Daphnia magna* were within the “toxic” range. However, most degradation intermediates exhibited reduced toxicity. For instance, the 96-hour LC_{50} value for BPA in Fathead minnow was 3.24 mg L^{-1} , while intermediates like P8, P10, and P11 showed significantly higher LC_{50} values of 35.51 mg L^{-1} , 38.69 mg L^{-1} , and 31.96 mg L^{-1} , respectively, indicating a reduced toxicity classification (Fig. 6a). Similarly, the 48-hour LC_{50} value for BPA in *Daphnia magna* was 1.58 mg L^{-1} , while most intermediates showed a marked increase in LC_{50} (Fig. 6b). As illustrated in Supplementary Fig. 39, BPA is classified as a “developmental toxicant”, while P8 and P9 are classified as “developmental non-toxicants”. Supplementary Table 7 shows that both acute and chronic toxicity values of the oxidized products decreased as degradation progressed. Notably, advanced oxidation products such as P9, P10, and P11 were predicted to be “not harmful”, underscoring the detoxification capacity of the $\text{FeN}_5\text{-SD}_x$ system.

Beyond computational predictions, experimental toxicity validation was performed using zebrafish and microbial viability assays. Zebrafish exposed to BPA solution exhibited early mortality and behavioral abnormalities over a 120-hour period. In contrast, those cultured in the post-reaction BPA solution from the FeN₅-SD₂/PMS system showed no abnormalities or fatalities (Supplementary Fig. 40), providing strong evidence for the biocompatibility of the treated water. Finally, flow cytometry analysis was conducted to evaluate the cytotoxic effects of treated and untreated solutions on microbial populations (Fig. 6c-d). After 40 minutes of treatment, the proportion of live cells increased from 35.1% to 70.6%, while dead cells decreased from 3.92% to 0.73%. These results highlight the robust detoxification capability of the FeN₅-SD₂/PMS system.”

Fig. 6 Toxicity analysis and environmental application. (a) Fathead minnow LC₅₀-96 h and (b) Daphnia magna LC₅₀-48 h of BPA and its degradation intermediates. Flow cytometry analysis of E. coli cells during BPA degradation in FeN₅-SD₂/PMS system at different reaction time: (c) 0 min, (d) 40 min. The four sections in flow cytometry images represent as below: Q1 dead cells; Q2 late apoptotic cells; Q3 early apoptotic cells; Q4 live cells.

Supplementary Fig. 39 Developmental toxicity of BPA and its degradation intermediates.

Supplementary Table 7. Acute and chronic toxicity results of BPA and its degradation products predicted using the Ecological Structure-Activity Relationships system.

Degradation intermediates	Acute toxicity (mg/L)			Chronic toxicity (mg/L)		
	Fish LC ₅₀	Daphnid LC ₅₀	Green Algae EC ₅₀	Fish Chv	Daphnid Chv	Green Algae Chv
BPA	6.27	4.15	5.78	0.733	0.617	2.12
P1	18.1	11.5	13.3	2.01	1.51	4.42
P2	13.3	8.50	10.4	1.50	1.16	3.58
P3	71.1	42.0	37.0	7.29	4.59	10.6
P4	906	536	473	93.0	58.6	136
P5	45.5	27.0	24.2	4.69	2.98	7.00
P6	15.1	9.37	10.1	1.64	1.17	3.23
P7	4.92	3.19	4.12	0.563	0.451	1.45
P8	593	313	171	53.1	24.8	38.0
P9	2870	1400	555	235	89.3	103
P10	212	115	71.1	19.6	9.89	16.8
P11	3330	1610	614	269	100	112

Supplementary Fig. 40 Images of zebrafish embryos under different treatments: nutrient medium only (Blank), BPA solution (Control), and post-reaction BPA solution from the FeN₅-SD₂/PMS system (FeN₅-SD₂/PMS).

“Supplementary Text 7 Toxicity Assessment.

Three healthy adult zebrafish (one female and two males) were selected and cohabited in a tank containing nutrient-rich medium for 12 hours to induce spawning and maximize embryo yield. After incubation, the fertilized embryos were thoroughly rinsed with deionized water to eliminate surface contaminants. Embryos at the eight-cell stage were then identified under a microscope and collected for subsequent experiments.

Exposure solutions were then prepared for the different test groups. The experimental group received the post-reaction solution obtained from the FeN₅-SD₂ catalytic system. In comparison, the blank group used only the nutrient medium, while the control group was treated with a BPA solution. Embryos were transferred into 24-well culture plates, with ten embryos assigned per group. Each well contained one embryo immersed in a mixture of 1 mL nutrient solution and 1 mL of the designated exposure solution. Embryo development was monitored microscopically, and the exposure solutions were refreshed every 24 hours from the onset of exposure until hatching.

The toxicity changes during BPA degradation in FeN₅-SD₂/PMS system were assessed by evaluating LIVE/DEAD stains of Escherichia coli (E. coli) using flow cytometry. First, 10 µL of E. coli was inoculated in 10 mL medium containing 3 g/L of beef extract, 10 g/L of tryptone, and 5 g/L of NaCl at pH 7.2 on a rotary shaker for 10 h at 30 °C. Then the strains were harvested by centrifugation at 6000 rpm for 5 min, followed by three washes in phosphate buffered saline (PBS, 0.5 M, pH 7.4). 1 mL of the obtained strain was added into a 10 mL of reaction solution, which was taken at different time intervals (0, 40 min). After culturing for 9 h, the mixed solution was centrifuged at 6000 rpm for 5 min, and the obtained product was washed three times with a PBS buffer. Then, 50 µL of propidium iodide (PI) and 50 µL of SYTOX were stained in the dark at 25 °C for 15 min to observe the changes in cytotoxicity over time by flow cytometry.”

5. The overall writing style of the manuscript needed improvement. In particular, the frequent use of excessively long sentences with multiple subordinate clauses tended to obscure the original meaning and reduced the clarity of key scientific points. A more concise and direct sentence structure was recommended to improve readability and ensure effective communication of central ideas. In addition, some of the wording in the manuscript appeared imprecise or overly generalized. Given the complexity and technical depth of the study, authors were advised to carefully reconsider the use of terminology.

Response: Thank you for your valuable comments on the overall writing style of the manuscript. We have revised the text to improve clarity and readability. Long and complex sentences have been shortened. We now use direct and concise language to highlight key scientific points. Imprecise and generalized wording has been corrected. Technical terms have been refined to accurately reflect the study's content. We also reviewed and standardized terminology related to coordination structure and reactive species. These changes improve the precision and consistency of the manuscript.

Once again, we sincerely thank you for your insightful comments and suggestions. We hope that the additional experiments and our responses help to further support the significance of our study.

References

1. Li, X. & Xiang, Z. Identifying the impact of the covalent-bonded carbon matrix to FeN₄ sites for acidic oxygen reduction. *Nat. Commun.* **13**, 57 (2022).
2. Zhang, K. et al. Molecular modulation of sequestered copper sites for efficient electroreduction of carbon dioxide to methane. *Adv. Funct. Mater.* **33**, 2214062 (2023).
3. Wu, H. et al. π -Conjugated polymeric phthalocyanine for the oxidative coupling of amines. *Chem. Commun.* **56**, 3637-3640 (2020).
4. Jin, Z. et al. Boosting electrocatalytic carbon dioxide reduction via self-relaxation of asymmetric coordination in Fe-based single atom catalyst. *Angew. Chem. Int. Ed.* **63**, e202318246 (2024).
5. Mo, F. et al. The optimized Fenton-like activity of Fe single-atom sites by Fe atomic clusters-mediated electronic configuration modulation. *Pro. Natl. Acad. Sci.* **120**, e2300281120 (2023).
6. Li, L. et al. Recent developments of microenvironment engineering of single-atom catalysts for oxygen reduction toward desired activity and selectivity. *Adv. Funct. Mater.* **31**, 2103857 (2021).
7. Tseng, Y. Y. & Tuan, H. Y. Coherent single-atom dipole-dipole coupling mediates holistic regulation of K⁺ migration for superior energy storage and dendrite-free metal deposition. *Adv. Funct. Mater.* 2423387. (2025).
8. Qian, K., Chen, H., Li, W., Ao, Z., Wu, Y. n. & Guan, X. Single-atom Fe catalyst outperforms its homogeneous counterpart for activating peroxydisulfate to

- achieve effective degradation of organic contaminants. *Environ. Sci. Technol.* **55**, 7034-7043 (2021).
9. Song, W. et al. Review of carbon support coordination environments for single metal atom electrocatalysts (SACS). *Adv. Mater.* **36**, 2301477 (2024).
 10. Yang, H. et al. Manganese vacancy-confined single-atom Ag in cryptomelane nanorods for efficient Wacker oxidation of styrene derivatives. *Chem. Sci.* **12**, 6099-6106 (2021).
 11. Sun, T. et al. Design of local atomic environments in single-atom electrocatalysts for renewable energy conversions. *Adv. Mater.* **33**, 2003075 (2021).
 12. Zhong, H. et al. Selective introduction of pentagon defects into Co-N₄ sites for boosting Fenton-like activity. *Adv. Funct. Mater.* 2501208. (2025).
 13. Yang, P. et al. Regulating the local electronic structure of copper single atoms with unsaturated B,O-coordination for selective ¹O₂ generation. *ACS Catal.* **13**, 12414-12424 (2023).
 14. Zhang, L. et al. Constructing hollow multishelled microreactors with a nanoconfined microenvironment for ofloxacin degradation through peroxymonosulfate activation: Evolution of high-valence cobalt-oxo species. *Environ. Sci. Technol.* **57**, 16141-16151 (2023).
 15. Zhu, C. et al. Heterogeneous Fe-Co dual-atom catalyst outdistances the homogeneous counterpart for peroxymonosulfate-assisted water decontamination: New surface collision oxidation path and diatomic synergy.

- Water Res.* **241**, 120164 (2023).
16. Wu, Z. et al. Facilely tuning the first-shell coordination microenvironment in iron single-atom for Fenton-like chemistry toward highly efficient wastewater purification. *Environ. Sci. Technol.* **57**, 14046-14057 (2023).
 17. Wu, Q. Y., Yang, Z. W., Wang, Z. W. & Wang, W. L. Oxygen doping of cobalt-single-atom coordination enhances peroxymonosulfate activation and high-valent cobalt-oxo species formation. *Pro. Natl. Acad. Sci.* **120**, e2219923120 (2023).
 18. Cheng, C., Ren, W., Miao, F., Chen, X., Chen, X. & Zhang, H. Generation of Fe^{IV}=O and its contribution to Fenton-Like reactions on a single-atom iron–N–C catalyst. *Angew. Chem.* **135**, e202218510 (2023).
 19. Xie, M. et al. Single-atom Co-N₅ catalytic sites on carbon nanotubes as peroxymonosulfate activator for sulfamerazine degradation via enhanced electron transfer pathway. *Sep. Purif. Technol.* **304**, 122398 (2023).
 20. Zhu, C., Nie, Y., Cun, F., Wang, Y., Tian, Z. & Liu, F. Two-step pyrolysis to anchor ultrahigh-density single-atom FeN₅ sites on carbon nitride for efficient Fenton-like catalysis near 0 °C. *Appl. Catal. B: Environ.* **319**, 121900 (2022).
 21. Zhang, L. S. et al. Carbon nitride supported high-loading Fe single-atom catalyst for activation of peroxymonosulfate to generate ¹O₂ with 100 % selectivity. *Angew. Chem. Int. Ed.* **60**, 21751-21755 (2021).
 22. Yang, T., Fan, S., Li, Y. & Zhou, Q. Fe-N/C single-atom catalysts with high density of Fe-N_x sites toward peroxymonosulfate activation for high-efficient

- oxidation of bisphenol A: Electron-transfer mechanism. *Chem. Eng. J* **419**, 129590 (2021).
23. Chen, T., Zhu, Z., Shen, X., Zhang, H., Qiu, Y. & Yin, D. Boosting peroxymonosulfate activation by porous single-atom catalysts with FeN₄O₁ configuration for efficient organic pollutants degradation. *Chem. Eng. J* **450**, 138469 (2022).
24. Yin, K. et al. Microenvironment modulation of cobalt single-atom catalysts for boosting both radical oxidation and electron-transfer process in Fenton-like system. *Appl. Catal. B* **329**, 122558 (2023).
25. Wang, Z. et al. Cobalt single atoms anchored on oxygen-doped tubular carbon nitride for efficient peroxymonosulfate activation: Simultaneous coordination structure and morphology modulation. *Angew. Chem.* **134**, e202202338 (2022).
26. Yin, K. et al. High-loading of well dispersed single-atom catalysts derived from Fe-rich marine algae for boosting Fenton-like reaction: Role identification of iron center and catalytic mechanisms. *Appl. Catal. B* **336**, 122951 (2023).
27. Liu, C., Li, J., He, X., Yue, J., Chen, M. & Chen, J. P. The “4 + 1” strategy fabrication of iron single-atom catalysts with selective high-valent iron-oxo species generation. *Pro. Natl. Acad. Sci.* **121**, e2322283121 (2024).
28. Zhen, J. et al. M–N₃ configuration on boron nitride boosts singlet oxygen generation via peroxymonosulfate activation for selective oxidation. *Angew. Chem. Int. Ed.* **63**, e202402669 (2024).
29. Yang, Z., Yang, X., Zhang, W., Ai, J. & Wang, D. Local electric field driving

- selective formation of high-valent Co(IV)=O species by regulating coordination microenvironment of Co single atom. *ACS ES&T Water* **4**, 3297-3308 (2024).
30. Zhu, P. et al. Tailoring the selectivity and activity of oxygen reduction by regulating the coordination environments of carbon-supported atomically dispersed metal sites. *J. Mater. Chem. A* **10**, 17948-17967 (2022).
 31. Kong, S. et al. Delocalization state-induced selective bond breaking for efficient methanol electrosynthesis from CO₂. *Nat. Cat.* **6**, 6-15 (2023).
 32. Cui, X., Gao, L., Lu, C. H., Ma, R., Yang, Y. & Lin, Z. Rational coordination regulation in carbon-based single-metal-atom catalysts for electrocatalytic oxygen reduction reaction. *Nano Converg.* **9**, 34 (2022).
 33. Li, Q. et al. Modulation of the second-beyond coordination structure in single-atom electrocatalysts for confirmed promotion of ammonia synthesis. *J. Am. Chem. Soc.* **147**, 1884-1892 (2025).
 34. Al-Nu'airat, J., Oluwoye, I., Zeinali, N., Altarawneh, M. & Dlugogorski, B. Z. Review of chemical reactivity of singlet oxygen with organic fuels and contaminants. *Chem. Rec.* **21**, 315-342 (2021).
 35. Lee, J., von Gunten, U. & Kim, J. H. Persulfate-based advanced oxidation: Critical assessment of opportunities and roadblocks. *Environ. Sci. Technol.* **54**, 3064-3081 (2020).
 36. Sun, Y. et al. Mn(III)-mediated bisphenol a degradation: Mechanisms and products. *Water Res.* **235**, 119787 (2023).

37. Miao, J., Geng, W., Alvarez, P. J. J. & Long, M. 2D N-doped porous carbon derived from polydopamine-coated graphitic carbon nitride for efficient nonradical activation of peroxymonosulfate. *Environ. Sci. Technol.* **54**, 8473-8481 (2020).
38. Deng, Y., Zhou, Y., Song, Z. & Yang, X. Long-range electronically polarized Fe-N₅ catalysts redirect polymerization-driven phenolic pollutant removal toward sustainable carbon sequestration. *Angew. Chem.* e202509493.
39. Bu, Y. et al. Peroxydisulfate activation and singlet oxygen generation by oxygen vacancy for degradation of contaminants. *Environ. Sci. Technol.* **55**, 2110-2120 (2021).
40. Wei, W. et al. Construction of anthracene-based metal–organic framework exhibiting enhanced singlet oxygen storage and release capabilities for efficient photodegradation of phenolic pollutants. *Small* **21**, 2411328 (2025).
41. Wang, D. et al. Selective oxygen activation by integrating dual photosensitizers in conjugated porous polymers and microenvironments manipulation for high-performance water purification and H₂O₂ production. *Adv. Funct. Mater.* **35**, 2419010 (2025).
42. Wang, L., Xiao, K. & Zhao, H. The debatable role of singlet oxygen in persulfate-based advanced oxidation processes. *Water Res.* **235**, 119925 (2023).

Responses to Reviewer's Comments

The revised manuscript entitled “**Remote Tuning of Single-Atom Fe-N₅ Sites via High-Coordination Defects for Enhanced Fenton-Like Water Decontamination**” (Manuscript ID: NCOMMS-25-25049B)

Reviewer #1 (Remarks to the Author):

While the authors have clearly invested significant effort and conducted extensive work, as previously noted, the level of innovation does not meet the threshold required for publication in this journal.

Response: We sincerely thank the reviewer for the thoughtful suggestions provided during the review process, which have helped us further refine and strengthen the manuscript. We believe our systematic regulation of FeN₅-SD_x coordination shells provides valuable mechanistic insights into PMS activation.

Reviewer #2 (Remarks to the Author):

In the manuscript, the multiple coordination shells of FeN₅-SD_x single-atom catalysts (SACs) were precisely regulated for peroxymonosulfate (PMS) activation in Fenton-like reactions. A point-by-point response to each point raised has been made, the revisions now meet the reviewers' requirements. In my opinion, the manuscript can be considered for publication at current state.

Response: Thank you for your positive feedback. We appreciate your approval and are glad the revisions meet the publication requirements.

Reviewer #3 (Remarks to the Author):

Accept

Response: We are grateful for the reviewer's positive assessment and acceptance recommendation.